# PromptRestorer: A Prompting Image Restoration Method with Degradation Perception

**Cong Wang**[1]*, **Jinshan Pan**[2]*, **Wei Wang**[3]*, **Jiangxin Dong**[2], **Mengzhu Wang**[4],
**Yakun Ju**[1], **Junyang Chen**[5]

[1]The Hong Kong Polytechnic University, [2]Nanjing University of Science and Technology,
[3]Dalian University of Technology, [4]Hebei University of Technology, [5]Shenzhen University

## Abstract

We show that raw degradation features can effectively guide deep restoration models, providing accurate degradation priors to facilitate better restoration. While networks that do not consider them for restoration forget gradually degradation during the learning process, model capacity is severely hindered. To address this, we propose a **Prompt**ing image **Restorer**, termed as **PromptRestorer**. Specifically, PromptRestorer contains two branches: a restoration branch and a prompting branch. The former is used to restore images, while the latter perceives degradation priors to prompt the restoration branch with reliable perceived content to guide the restoration process for better recovery. To better perceive the degradation which is extracted by a pre-trained model from given degradation observations, we propose a prompting degradation perception modulator, which adequately considers the characters of the self-attention mechanism and pixel-wise modulation, to better perceive the degradation priors from global and local perspectives. To control the propagation of the perceived content for the restoration branch, we propose gated degradation perception propagation, enabling the restoration branch to adaptively learn more useful features for better recovery. Extensive experimental results show that our PromptRestorer achieves state-of-the-art results on 4 image restoration tasks, including image deraining, deblurring, dehazing, and desnowing.

## 1 Introduction

Image restoration aims to recover clear high-quality images from given degraded ones. It is highly ill-posed since only degraded images can be exploited, statistical observations are thus required to well-pose the problems [37, 67, 66, 41, 42]. Although conventional approaches can recover images to some extent, they typically involve solving optimization algorithms that are difficult due to the non-convexity and non-smooth problems. Additionally, the observations may not always hold, which can cause algorithms to fail.

With the emergence of convolutional neural networks (CNNs) [48] and Transformers [22, 45], which perform well at implicitly learning the priors from large-scale data, learning-based methods have dominated recent image restoration tasks and achieved impressive performance [81, 51, 103, 72, 61, 93, 33, 13]. However, these methods are usually built without explicitly considering the specific degradation information, which accordingly limits model capacity (**Case 1** in Fig. 1). An alternative approach is to design a conditional branch to learn additional information to provide the restoration network with useful content for modulation [30, 17, 34, 35] (**Case 2** in Fig. 1). However, we note that while conditional branches in these models are learnable, they may not effectively provide degradation information, as the optimizable parameters result in gradually clearer features during the learning process, leading to the degradation vanishing which accordingly limits model performance.

---

*These authors equally contribute to this work.

37th Conference on Neural Information Processing Systems (NeurIPS 2023).

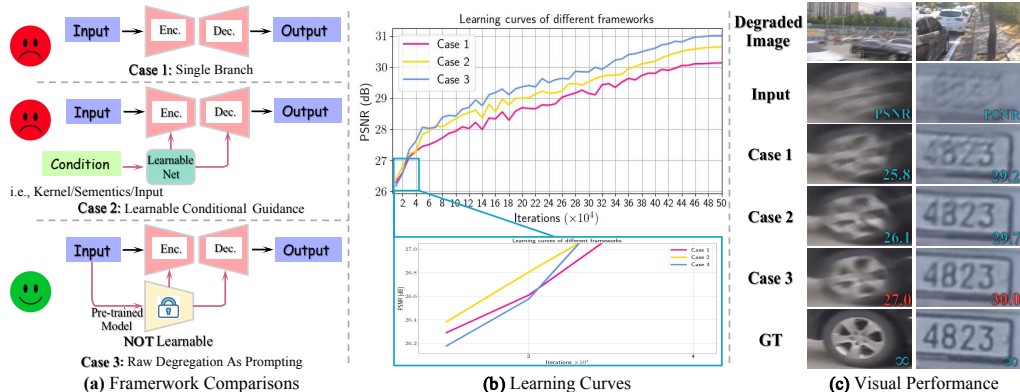

Figure 1: **(a)** compares different restoration frameworks. Unlike existing approaches that are built within the architectures such as **Cases 1-2**, which are unable to memorize the degradation well during the learning process, we propose a prompting method (**Case 3**) that directly exploits raw degradation features extracted by a pre-trained model from the given degradation observations to guide restoration. In **(b)**, we observe that both **Cases 1-2** outperform our method in early iterations, as they effectively memorize degraded information. However, both **Cases 1-2** experience degradation vanishing with further iterations (better demonstrated in Sec. 4.3), while our prompting method persists in guiding the restoration network with accurate degradation priors, accordingly producing better restoration quality. In **(c)**, visual performance demonstrates that our prompting method recovers sharper images. Quantitative results are reported in Tab. 5.

Recently, prompt learning has been shown an effective tool to improve model performance by designing various prompts [115, 98, 104, 23, 114, 46, 53, 43]. The prompt usually serves as the guidance tool to correct the networks toward better results [28]. However, prompt learning still keeps a margin for image restoration, and existing prompts may not be suitable for image restoration since they cannot effectively model degradation priors well. Hence, we ask: *Is there a reasonable prompting manner to correct degraded image restoration networks to facilitate better recovery?*

*The answer is in the riddle*. This paper proposes the **PromptRestorer**, a **Prompt**ing image **Restorer**, to overcome degradation vanishing in image restoration via promoting by exploring degradation input itself for better restoration (**Case 3** in Fig. 1). Our idea is simple: we directly exploit the raw degraded features extracted by a pre-trained model from the degraded inputs to generate more reliable prompting content to guide image restoration. Raw degraded features preserve accurately degraded information, which can consistently prompt the restoration network with accurate degraded priors, enabling the restoration network to perceive the degradation for better restoration. Hence, we design the PromptRestorer, which consists of two branches: (a) the restoration branch and (b) the prompting branch. The former is used to restore images and the latter is used to generate reliable prompting features to guide the restoration network for better restoration. To better perceive the degradation, we propose a **Prompt**ing **D**egradation **P**erception **M**odulator (**PromptDPM**), which consists of **G**lobal **P**rompting **P**erceptor (**G2P**) and **L**ocal **P**rompting **P**erceptor (**L2P**). The G2P adequately exploits the self-attention mechanism to form global prompting attention, while the L2P considers the pixel-level perception to build local prompting content. To control the propagation of perceived features in the restoration branch, we propose **G**ated **D**egradation Perception **P**ropagation (**GDP**), enabling the restoration network to adaptively learn more useful features to facilitate better restoration.

The main contributions of this work are summarized below:

- We propose PromptRestorer, which is the first approach to our knowledge that takes advantage of the prompting learning for general image restoration by considering raw degradation features in restoration, enabling the restoration model to overcome degradation vanishing while consistently retaining the degradation priors to facilitate better restoration.
- We propose a prompting degradation perception modulator that is used to perceive degradation from global and local perspectives, which is able to provide the restoration network

with more reliable perceived content learned from the degradation priors, enabling it to better guide the restoration process.

- We propose gated degradation perception propagation that exploits a gating mechanism to control the propagation of the perceived features, enabling the model to adaptively learn more useful features for better image restoration.

Fig. 1 summarises framework comparisons, and their learning curves and visual performance. Deeper analysis and discussion about them are provided in Sec. 4.2.

## 2 Related Work

In this section, we review image restoration, conditional modulation, and prompt learning.

**Image Restoration.** Recently, CNN-based architectures [110, 112, 103, 4, 24, 102, 89, 91, 87, 19, 88, 117] and Transformer-based models [96, 56, 49, 12, 93, 93] have been shown to outperform conventional restoration approaches [37, 83, 64, 47, 7, 79]. These learning-based methods usually adopt U-Net architectures [50, 18, 103, 100, 1, 93, 108, 101], which have been demonstrated the effectiveness because of hierarchical multi-scale representation and effective learning between shallow and deeper layers by skip connection [111, 59, 102, 31]. We refer the readers to recent excellent literature reviews on image restoration [5, 54, 82], which summarise the main designs in deep image restoration models.

Although these models have achieved promising performance, they do not explicitly take degradation into consideration for model design which is vital for restoration, limiting the model capacity.

**Conditional Modulation.** Conditional modulation usually involves implicitly modulating the additional content to guide the restoration [30, 10, 16, 17, 36, 35, 34, 60, 57, 92]. These approaches usually contain two branches: a basic network and a conditional network. The conditional network provides additional information to guide the basic network for restoration via spatial feature transform (SFT) [92]. Among these methods, blur kernel [30], semantics [92], and degraded input [57, 16] which serve as the additional conditions are broadly known.

The learnable nature of the conditional network in these models does not effectively provide degradation information for the basic network. As parameters are optimized in the learning process, features become gradually clear.

**Prompt Learning.** Prompt learning methods have been studied broadly in natural language processing (NLP) [75, 78, 8]. Due to high effectiveness, prompt learning is recently used in vision-related tasks [115, 98, 104, 23, 114, 46, 53, 43, 71, 29, 40, 86, 28]. In vision prompt learning, many works seek useful prompts to correct task networks toward better performance [28].

Although prompt learning has shown promise in various vision tasks, it still keeps a margin in general image restoration. This paper proposes an effective prompting method, enabling the restoration model to overcome the degradation vanishing in the learning process for better restoration.

## 3 PromptRestorer

Our goal aims to overcome degradation vanishing and better perceive degradation in deep restoration models to improve image recovery quality. To achieve this, we introduce a prompting strategy that helps the model consistently memorize degradation information, enabling it to prompt restoration with better degradation for better restoration. To better perceive degradation, we propose the **Prompt**ing **D**egradation **P**erception **M**odulator (**PromptDPM**), which can provide more reliable perceived content to guide the restoration network. To control the propagation of the perceived content, we propose the **G**ated **D**egradation Perception **P**ropagation (**GDP**), enabling the restoration network to adaptively learn more useful features for better restoration.

### 3.1 Overall Pipeline

Fig. 2 shows the framework of our PromptRestorer, which contains two branches: (a) the restoration branch and (b) the prompting branch. The restoration branch is used to restore images, where each block is prompted by the prompting branch. The prompting branch first generates the accurate

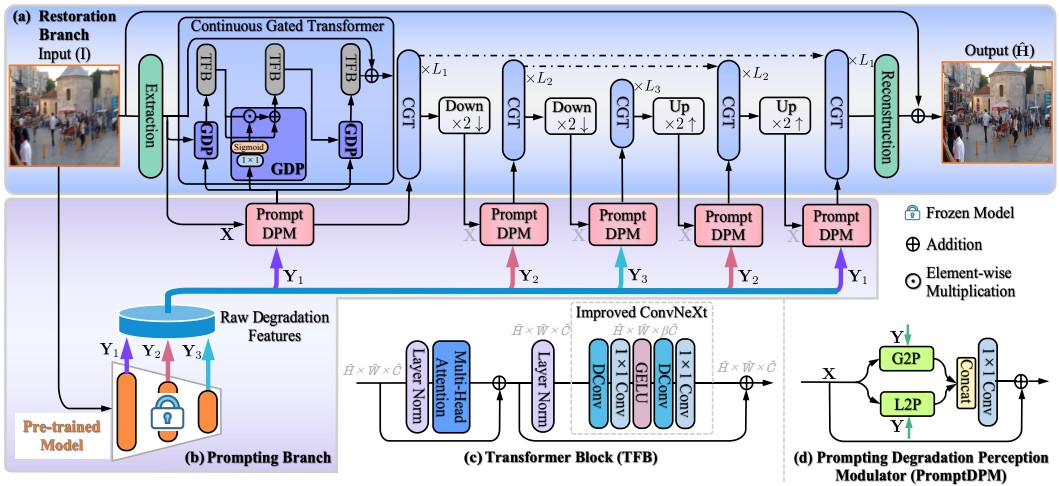

Figure 2: Overall pipeline of our **PromptRestorer**. PromptRestorer contains two branches: **(a)** the restoration branch and **(b)** the prompting branch. The restoration branch is used to restore images, where each block **(c)** in CGT is prompted by the prompting branch. The prompting branch first generates precise degradation features extracted by a pre-trained model from degradation observations, then these features prompt the restoration branch to facilitate better restoration via PromptDPM **(d)**.

degradation feature extracted by a pre-trained model, and then the feature is to prompt the restoration branch, enabling the restoration branch to better perceive the degradation prior for better recovery.

**Restoration Branch.** Given a degraded input image $\mathbf{I} \in \mathbb{R}^{H \times W \times 3}$, we first applies a $3 \times 3$ convolution as the feature extraction to obtain low-level embeddings $\mathbf{X}_0 \in \mathbb{R}^{H \times W \times C}$; where $H \times W$ denotes the spatial dimension and $C$ is the number of channels. Next, the shallow features $\mathbf{X}_0$ gradually are hierarchically encoded into deep features $\mathbf{X}_l \in \mathbb{R}^{\frac{H}{l} \times \frac{W}{l} \times lC}$. After encoding the degraded input into low-resolution latent features $\mathbf{X}_3 \in \mathbb{R}^{\frac{H}{3} \times \frac{W}{3} \times 3C}$, the decoder progressively recovers the high-resolution representations. Finally, a reconstruction layer which contains 4 Transformer blocks as the refinement followed by a $3 \times 3$ convolution is applied to decoded features to generate residual image $\mathbf{S} \in \mathbb{R}^{H \times W \times 3}$ to which degraded image is added to obtain the restored output image: $\hat{\mathbf{H}} = \mathbf{I} + \mathbf{S}$. Both encoder and decoder at $l$- level consist of multiple **C**ontinuous **G**ated **T**ransformers (**CGT**) with expanding channel capacity. To help better recovery, the encoder features are concatenated with the decoder features via skip connections [74] by $1 \times 1$ convolutions.

**Prompting Branch.** The prompting branch, as shown in Fig. 2(b), aims to generate and perceive degradation features and then provide useful guidance content for the restoration branch. We note VQGAN [25] has been demonstrated that it can generate high-quality images while representing the features of input images. However, it tends to damage image structure after vector quantization [116, 11, 32]. To avoid this problem, we only exploit the encoder of pre-trained VQGAN to represent the deep features of the degraded inputs. We first use the pretrained encoder to extract degraded features $\mathbf{Y}_l \in \mathbb{R}^{\frac{H}{l} \times \frac{W}{l} \times lC}$; where $l$ denotes the $l$- level layer in the pre-trained encoder. Then, the degraded features are exploited to generate reliable prompting content to prompt the restoration branch by PromptDPM (see Sec. 3.2). The generated prompting content is transmitted to each CGT to guide the restoration branch.

**Continuous Gated Transformers.** CGT exploits the perceived features from PromptDPM to provide the Transformer block with more reliable content to overcome degradation vanishing to facilitate better restoration. Each CGT consists of three Transformer blocks (Fig. 2(c)) with residual connections [38] and the input in each block is gated by GDP (see Sec. 3.3) to control the propagation of perceived features. Let $\mathcal{P}$, $\mathcal{G}$, and $\mathcal{T}$ respectively denote the operations of PromptDPM (expressed in (2)), GDP (expressed in (7)), and Transformer, the features flow in $k^{th}$ block in one CGT at $l$- level encoder/decoder, which can be expressed as:

$$\mathbf{X}_k = \mathcal{T}(\mathbf{G}_{k-1}); \mathbf{G}_{k-1} = \mathcal{G}(\mathbf{X}_{k-1}, \mathbf{P}_l); \mathbf{P}_l = \mathcal{P}(\mathbf{X}_{k-1}, \mathbf{Y}_l), \tag{1}$$

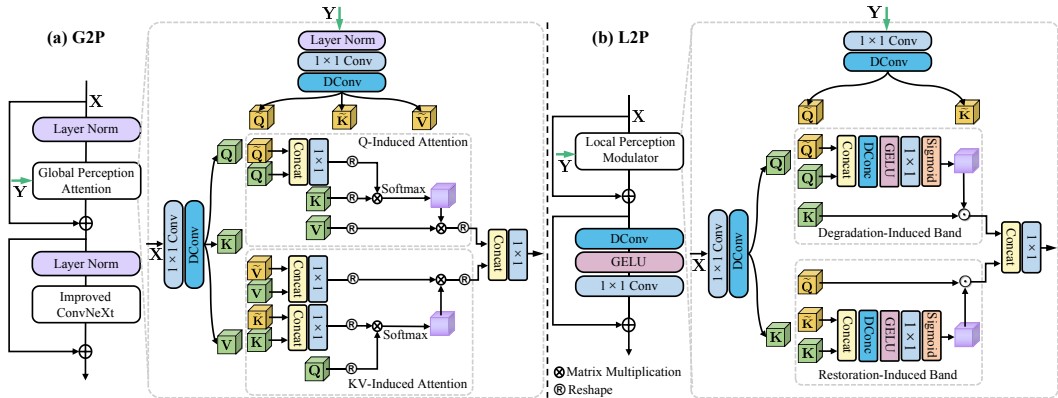

Figure 3: **(a)** Global Prompting Perceptor (**G2P**); **(b)** Local Prompting Perceptor (**L2P**).

where $\mathbf{X}_k$ means the output of $k^{th}$ Transformer block in one CGT, especially $\mathbf{X}_0$ is the downsampled/upsampled features at $(l-1)$- level encoder/decoder; $\mathbf{P}_l$ refers to the generated features of PromptDPM at $l$-level layer; $\mathbf{G}_{k-1}$ means the gated features between $\mathbf{X}_{k-1}$ and $\mathbf{P}_l$, which serves as the input of $k^{th}$ Transformer block. Each Transformer block consists of multi-head attention [101] followed by an improved ConvNeXt [62] as the feed-forward network (see Fig. 2(c)).

## 3.2 Prompting Degradation Perception Modulator

To better perceive the degradation to prompt the restoration network with more reliable perceived content from the degradation priors, we propose the PromptDPM (see Fig. 2(d)). The PromptDPM consists of 1) **G**lobal **P**rompting **P**erceptor (**G2P**, introduced in Sec. 3.2.1) and 2) **L**ocal **P**rompting **P**erceptor (**L2P**, introduced in Sec. 3.2.2) to respectively perceive the degradation from global and local perspectives, enabling to generate more useful content to guide the restoration branch. From a restoration tensor $\mathbf{X} \in \mathbb{R}^{\hat{H} \times \hat{W} \times \hat{C}}$ and a degradation tensor $\mathbf{Y} \in \mathbb{R}^{\hat{H} \times \hat{W} \times \hat{C}}$, we prompt $\mathbf{X}$ with $\mathbf{Y}$:

$$\mathcal{P}(\mathbf{X}, \mathbf{Y}) = W_p\Big(\mathcal{C}\big[\Psi^{\mathbf{global}}(\mathbf{X}, \mathbf{Y}), \Psi^{\mathbf{local}}(\mathbf{X}, \mathbf{Y})\big]\Big) + \mathbf{X}, \tag{2}$$

where $\Psi^{\mathbf{global}}(\cdot, \cdot)$ and $\Psi^{\mathbf{local}}(\cdot, \cdot)$ respectively denote the operations of G2P and L2P; $\mathcal{C}[\cdot, \cdot]$ means the concatenation at channel dimension; $W_p(\cdot)$ refers to the $1 \times 1$ point-wise convolution.

### 3.2.1 Global Prompting Perceptor

The G2P, shown in Fig. 3(a), fully exploits the self-attention mechanism to form the global prompting attention induced by the degraded features. The G2P contains the global perception attention followed by an improved ConvNeXt [62]. Our global perception attention consists of 1) **Q**uery-**In**duced **Att**ention (**Q-InAtt**) and 2) **K**ey-**V**alue-**In**duced **Att**ention (**KV-InAtt**). The Q-InAtt considers re-forming the query vector induced by degradation features to build a representative query to perform attention, while the KV-InAtt re-considers key and value vectors induced by other degradation counterparts to search for more similar content with the restoration query. From a layer normalized restoration tensor $\mathbf{X} \in \mathbb{R}^{\hat{H} \times \hat{W} \times \hat{C}}$, our G2P first generates *restoration query* (**Q**), *key* (**K**), and *value* (**V**) projections from the restoration features. It is achieved by applying $1 \times 1$ convolutions to aggregate pixel-wise cross-channel context followed by $3 \times 3$ depth-wise convolutions $W_d(\cdot)$ to encode channel-wise spatial context, yielding $\mathbf{Q} = W_d W_p \mathbf{X}$, $\mathbf{K} = W_d W_p \mathbf{X}$, and $\mathbf{V} = W_d W_p \mathbf{X}$. Meanwhile, we similarly convert the degradation tensor $\mathbf{Y} \in \mathbb{R}^{\hat{H} \times \hat{W} \times \hat{C}}$ into *degradation query* ($\widetilde{\mathbf{Q}}$), *key* ($\widetilde{\mathbf{K}}$), and *value* ($\widetilde{\mathbf{V}}$) projections: $\widetilde{\mathbf{Q}} = W_d W_p \mathbf{Y}$, $\widetilde{\mathbf{K}} = W_d W_p \mathbf{Y}$, and $\widetilde{\mathbf{V}} = W_d W_p \mathbf{Y}$. Then, we respectively conduct Q-InAtt and KV-InAtt:

$$\mathbf{A}_{\text{Q-InAtt}} = \mathcal{A}_{\text{Q-InAtt}}\Big(W_p\big(\mathcal{C}[\mathbf{Q}, \widetilde{\mathbf{Q}}]\big), \mathbf{K}, \mathbf{V}\Big); \mathbf{A}_{\text{KV-InAtt}} = \mathcal{A}_{\text{KV-InAtt}}\Big(\mathbf{Q}, W_p\big(\mathcal{C}[\mathbf{K}, \widetilde{\mathbf{K}}]\big), W_p(\mathcal{C}[\mathbf{V}, \widetilde{\mathbf{V}}])\Big),$$
$$\tag{3}$$

where $\mathcal{A}_{(\cdot)}\left(\hat{\mathbf{Q}}, \hat{\mathbf{K}}, \hat{\mathbf{V}}\right) = \hat{\mathbf{V}} \cdot \text{Softmax}\left(\hat{\mathbf{K}} \cdot \hat{\mathbf{Q}} / \alpha\right)$; Here, $\alpha$ is a learnable scaling parameter to control

the magnitude of the dot product of $\hat{\mathbf{K}}$ and $\hat{\mathbf{Q}}$ before applying the softmax function. Similar to the conventional multi-head SA [22], we divide the number of channels into 'heads' and learn separate attention maps. Then two induced attentions are fused and followed by an improved ConvNeXt:

$$\mathbf{A}^{'} = W_p\big(\mathcal{C}[\mathbf{A}_{\text{Q-InAtt}}, \mathbf{A}_{\text{KV-InAtt}}]\big) + \mathbf{X}; \mathbf{A} = W_p W_d \phi W_p W_d\big(LN(\mathbf{A}^{'})\big) + \mathbf{A}^{'}, \tag{4}$$

where the $W_p W_d \phi W_p W_d(\cdot)$ means the improved ConvNeXt shown in the latter of Fig. 2(c); $LN(\cdot)$ means the operation of layer normalization [6].

### 3.2.2 Local Prompting Perceptor

The L2P, as shown in Fig. 3(b), adequately considers the pixel-level degradation perception, enabling to better perceive degradation from spatially neighboring pixel positions. The L2P consists of a local perception modulator followed by a separable depth-level convolution. The local perception modulator contains two core components: 1) **Deg**radation-**In**duced **Ban**d (**Deg-InBan**) and 2) **Res**toration-**In**duced **Ban**d (**Res-InBan**). The former is achieved by exploiting the degradation features to induce spatially useful content from restoration content to guide restoration gating fusion, while the latter utilizes the deep restoration features to induce more useful features from another degradation counterpart to form the degradation gating. Given the degradation tensor $\mathbf{Y} \in \mathbb{R}^{\hat{H} \times \hat{W} \times \hat{C}}$, we first exploit the point-wise convolution and $3 \times 3$ depth-wise convolution to encode two *degradation* projections, yielding $\widetilde{\mathbf{Q}} = W_d^Q W_p^Q \mathbf{Y}$ and $\widetilde{\mathbf{K}} = W_d^K W_p^K \mathbf{Y}$. Meanwhile, the restoration tensor $\mathbf{X} \in \mathbb{R}^{\hat{H} \times \hat{W} \times \hat{C}}$ are also encoded into two *restoration* projections: $\mathbf{Q} = W_d^Q W_p^Q \mathbf{X}$ and $\mathbf{K} = W_d^K W_p^K \mathbf{X}$. Then, we respectively conduct Deg-InBan and Res-InBan:

$$\mathbf{Z}_{\text{Deg-InBan}} = \sigma\Big(W_p \phi W_d(\mathcal{C}[\widetilde{\mathbf{Q}}, \mathbf{Q}])\Big) \odot \mathbf{K}; \mathbf{Z}_{\text{Res-InBan}} = \widetilde{\mathbf{Q}} \odot \sigma\Big(W_\phi W_d(\mathcal{C}[\widetilde{\mathbf{K}}, \mathbf{K}])\Big), \tag{5}$$

where $\sigma(\cdot)$ denotes the sigmoid function that controls the gating level. Then, the perceived features in the two bands are fused via concatenation and $1 \times 1$ convolution and followed by a depth-level separable convolution $W_p \phi W_d(\cdot)$:

$$\mathbf{Z}^{'} = W_p\big(\mathcal{C}[\mathbf{Z}_{\text{Deg-InBan}}, \mathbf{Z}_{\text{Res-InBan}}]\big) + \mathbf{X}; \mathbf{Z} = W_p \phi W_d(\mathbf{Z}^{'}) + \mathbf{Z}^{'}. \tag{6}$$

### 3.3 Gated Degradation Perception Propagation

The GDP aims to control the propagation of the perceived degradation, enabling to adaptively learn more useful features in Transformer blocks to facilitate better restoration. Given the output restoration tensor $\mathbf{X}_{k-1} \in \mathbb{R}^{\hat{H} \times \hat{W} \times \hat{C}}$ of $(k-1)^{th}$ Transformer block in one CGT and the perceived tensor $\mathbf{P}_l \in \mathbb{R}^{\hat{H} \times \hat{W} \times \hat{C}}$ which is the output feature of one PromptDPM at $l$- level, the input of $k^{th}$ Transformer block can be obtained by gating the $\mathbf{X}_{k-1}$ with $\mathbf{P}_l$ by $1 \times 1$ convolution and gated control function sigmoid $\sigma(\cdot)$ with residual learning [39]:

$$\mathcal{G}\big(\mathbf{X}_{k-1}, \mathbf{P}_l\big) = \sigma(W_p \mathbf{P}_l) \odot \mathbf{X}_{k-1} + \mathbf{X}_{k-1}. \tag{7}$$

### 3.4 Learning Strategy

To train the network, two objective loss functions are adopted, including image reconstruction loss ($\mathcal{L}_i$) for pixel recovery and frequency loss ($\mathcal{L}_f$) for detail enhancement [18]:

$$\mathcal{L} = \mathcal{L}_i + \lambda \mathcal{L}_f, \text{where } \mathcal{L}_i = \|\hat{\mathbf{H}} - \mathbf{H}\|_1; \mathcal{L}_f = \|\mathcal{F}(\hat{\mathbf{H}}) - \mathcal{F}(\mathbf{H})\|_1, \tag{8}$$

where $\mathbf{H}$ denotes the ground truth image; $\mathcal{F}$ denotes the Fast Fourier transform; $\lambda$ is a weight that is empirically set to be 0.1.

## 4 Experiment

We evaluate PromptRestorer on benchmarks for 4 image restoration tasks: **(a)** deraining, **(b)** deblurring, **(c)** desnowing, and **(d)** dehazing. We train separate models for different image restoration tasks. Our PromptRestorer employs a 3-level encoder-decoder. From level-1 to level-3, the number of CGT is $[2, 3, 6]$, attention heads are $[2, 4, 8]$, and number of channels is $[48, 96, 192]$. The expanding channel capacity factor $\beta$ is 4. For downsampling and upsampling, we adopt pixel-unshuffle and pixel-shuffle [77], respectively. We train models with AdamW optimizer with the initial learning rate $3e^{-4}$ gradually reduced to $1e^{-6}$ with the cosine annealing [63]. The patch size is set as $256 \times 256$.

Table 1: **Image deraining** results. Our **PromptRestorer** advances recent 14 state-of-the-arts on average.

| Method | Test100 [107] PSNR ↑ | SSIM ↑ | Rain100H [97] PSNR ↑ | SSIM ↑ | Rain100L [97] PSNR ↑ | SSIM ↑ | Test2800 [27] PSNR ↑ | SSIM ↑ | Test1200 [106] PSNR ↑ | SSIM ↑ | Average PSNR ↑ | SSIM ↑ |
|---|---|---|---|---|---|---|---|---|---|---|---|---|
| DerainNet [26] | 22.77 | 0.810 | 14.92 | 0.592 | 27.03 | 0.884 | 24.31 | 0.861 | 23.38 | 0.835 | 22.48 | 0.796 |
| SEMI [95] | 22.35 | 0.788 | 16.56 | 0.486 | 25.03 | 0.842 | 24.43 | 0.782 | 26.05 | 0.822 | 22.88 | 0.744 |
| DIDMDN [106] | 22.56 | 0.818 | 17.35 | 0.524 | 25.23 | 0.741 | 28.13 | 0.867 | 29.65 | 0.901 | 24.58 | 0.770 |
| UMRL [99] | 24.41 | 0.829 | 26.01 | 0.832 | 29.18 | 0.923 | 29.97 | 0.905 | 30.55 | 0.910 | 28.02 | 0.880 |
| RESCAN [55] | 25.00 | 0.835 | 26.36 | 0.786 | 29.80 | 0.881 | 31.29 | 0.904 | 30.51 | 0.882 | 28.59 | 0.857 |
| PreNet [72] | 24.81 | 0.851 | 26.77 | 0.858 | 32.44 | 0.950 | 31.75 | 0.916 | 31.36 | 0.911 | 29.42 | 0.897 |
| MSPFN [44] | 27.50 | 0.876 | 28.66 | 0.860 | 32.40 | 0.933 | 32.82 | 0.930 | 32.39 | 0.916 | 30.75 | 0.903 |
| DCSFN [90] | 27.46 | 0.887 | 28.98 | 0.887 | 34.70 | 0.961 | 30.96 | 0.903 | 32.92 | 0.937 | 31.00 | 0.915 |
| MPRNet [103] | 30.27 | 0.897 | 30.41 | 0.890 | 36.40 | 0.965 | 33.64 | 0.938 | 32.91 | 0.916 | 32.73 | 0.921 |
| SPAIR [69] | 30.35 | 0.909 | 30.95 | 0.892 | 36.93 | 0.969 | 33.34 | 0.936 | 33.04 | 0.922 | 32.91 | 0.926 |
| Uformer [93] | 29.17 | 0.880 | 30.06 | 0.884 | 36.34 | 0.966 | 33.36 | 0.935 | 31.98 | 0.909 | 32.18 | 0.915 |
| MAXIM-2S [84] | 31.17 | **0.922** | 30.81 | 0.903 | 38.06 | 0.977 | 33.80 | 0.943 | 32.37 | 0.922 | 33.24 | 0.933 |
| Restormer [101] | **32.00** | 0.923 | 31.46 | 0.904 | 38.99 | **0.978** | **34.18** | **0.944** | 33.19 | 0.926 | 33.96 | 0.935 |
| SFNet [20] | 31.47 | 0.919 | **31.90** | 0.908 | 38.21 | 0.974 | 33.69 | 0.937 | 32.55 | 0.911 | 33.56 | 0.929 |
| **PromptRestorer** | **31.84** | 0.920 | **31.72** | **0.908** | **39.04** | **0.977** | **34.40** | **0.947** | **33.27** | **0.928** | **34.05** | **0.936** |

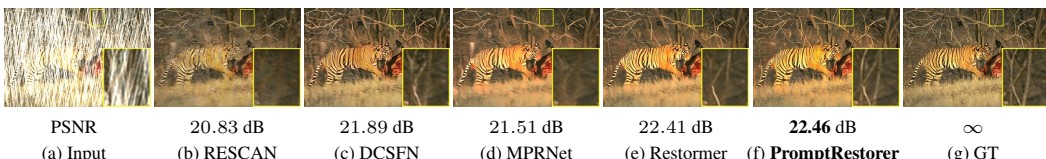

| PSNR | 20.83 dB | 21.89 dB | 21.51 dB | 22.41 dB | **22.46 dB** | ∞ |
|---|---|---|---|---|---|---|
| (a) Input | (b) RESCAN | (c) DCSFN | (d) MPRNet | (e) Restormer | (f) **PromptRestorer** | (g) GT |

Figure 4: **Image deraining** example on Rain100H [97].

Table 2: **Image deblurring** results. Our **PromptRestorer** is trained only on the GoPro dataset [65] and directly applied to the HIDE [76] and RealBlur [73] benchmark datasets.

| Benchmark | Metrics | Nah et al. [65] | SRN [81] | DBGAN [109] | MT-RNN [68] | DMPHN [105] | Suin et al. [80] | SPAIR [69] | MIMO-UNet+ [18] | MPRNet [103] | Restormer [101] | PromptRestorer |
|---|---|---|---|---|---|---|---|---|---|---|---|---|
| **GoPro** [65] | PSNR ↑ | 21.00 | 30.26 | 31.10 | 31.15 | 31.20 | 31.85 | 32.06 | 32.45 | 32.66 | **32.92** | **33.06** |
|  | SSIM ↑ | 0.914 | 0.934 | 0.942 | 0.945 | 0.940 | 0.948 | 0.953 | 0.957 | 0.959 | **0.961** | **0.962** |
| **HIDE** [76] | PSNR ↑ | 25.73 | 28.36 | 28.94 | 29.15 | 29.09 | 29.98 | 30.29 | 29.99 | 30.96 | **31.22** | **31.36** |
|  | SSIM ↑ | 0.874 | 0.915 | 0.915 | 0.918 | 0.924 | 0.930 | 0.931 | 0.930 | 0.939 | **0.942** | **0.944** |
| **RealBlur-R** [73] | PSNR ↑ | 32.51 | 35.66 | 33.78 | 35.79 | 35.70 | - | - | 35.54 | 35.99 | **36.19** | **36.06** |
|  | SSIM ↑ | 0.841 | 0.947 | 0.909 | 0.951 | 0.948 | - | - | 0.947 | 0.952 | **0.957** | 0.954 |
| **RealBlur-J** [73] | PSNR ↑ | 27.87 | 28.56 | 24.93 | 28.44 | 28.42 | - | 28.81 | 27.63 | 28.70 | **28.96** | 28.82 |
|  | SSIM ↑ | 0.827 | 0.867 | 0.745 | 0.862 | 0.860 | - | **0.875** | 0.837 | 0.873 | **0.879** | 0.873 |

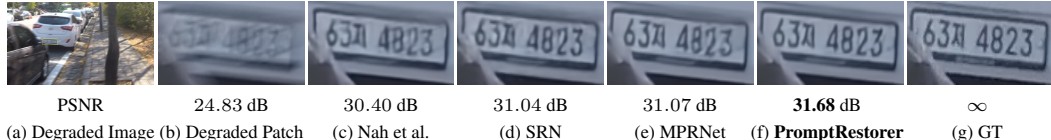

| PSNR | 24.83 dB | 30.40 dB | 31.04 dB | 31.07 dB | **31.68 dB** | ∞ |
|---|---|---|---|---|---|---|
| (a) Degraded Image | (b) Degraded Patch | (c) Nah et al. | (d) SRN | (e) MPRNet | (f) **PromptRestorer** | (g) GT |

Figure 5: **Image deblurring** example on GoPro [65].

## 4.1 Main Results

**Image Deraining Results.** Similar to existing methods [44, 103, 69], we report PSNR/SSIM scores using Y channel in YCbCr color. Tab. 1 shows that our PromptRestorer outperforms current state-of-the-art approaches when averaged across all five datasets. Compared to the recent best method Restormer [101], PromptRestorer achieves 0.09 dB improvement on average. On individual datasets, the gain can be as large as 0.22 dB, e.g., Test2800 [27]. In Fig. 4, we present a challenging visual deraining example, where our PromptRestorer is able to generate a clearer result with finer details.

**Image Deblurring Results.** We evaluate deblurring results on both synthetic datasets (GoPro [65], HIDE [76]) and real-world datasets (RealBlur-R [73], RealBlur-J [73]). Tab. 2 summarises the results, where our PromptRestorer advances current state-of-the-art approaches on GoPro [65] and HIDE [76]. Compared with MPRNet [103], our PromptRestorer obtains a performance 0.12 dB gains. Fig. 5 provides a visual deblurring example. Our PromptRestorer produces a sharper result with fewer artifacts.

Table 3: **Image dehazing** results on SOTS-Indoor [52] and real-world benchmarks Dense-Haze [2] and NH-Haze [3]. Our **PromptRestorer** significantly advances state-of-the-arts on SOTS-Indoor [52].

| Benchmark | Metrics | DCP [37] | DehazeNet [9] | AODNet [51] | GridNet [58] | FFANet [70] | MSBDN [21] | UHD [113] | MAXIM [84] | DeHamer [33] | PromptRestorer |
|---|---|---|---|---|---|---|---|---|---|---|---|
| **SOTS-Indoor** [52] | PSNR ↑ | 16.61 | 19.82 | 20.51 | 32.16 | 36.39 | 32.77 | 21.75 | **38.11** | 36.63 | 42.54 |
| | SSIM ↑ | 0.8546 | 0.8209 | 0.8162 | 0.9836 | 0.9886 | 0.9812 | 0.8786 | **0.9910** | 0.9881 | 0.9945 |
| **Dense-Haze** [2] | PSNR ↑ | 11.01 | 9.48 | 12.82 | 14.96 | 12.22 | 15.13 | 12.16 | - | **16.62** | 15.86 |
| | SSIM ↑ | 0.4165 | 0.4383 | 0.4683 | 0.5326 | 0.4440 | 0.5551 | 0.4594 | - | **0.5602** | 0.5680 |
| **NH-Haze** [3] | PSNR ↑ | 12.72 | 11.76 | 15.69 | 18.33 | 18.13 | 17.97 | 16.05 | - | **20.66** | 20.36 |
| | SSIM ↑ | 0.4419 | 0.3988 | 0.5728 | 0.6667 | 0.6473 | 0.6591 | 0.4612 | - | **0.6844** | 0.7203 |

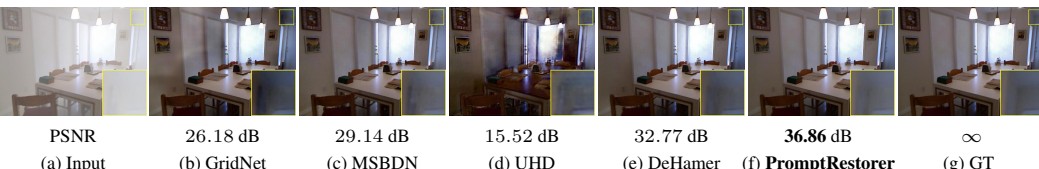

| PSNR | 26.18 dB | 29.14 dB | 15.52 dB | 32.77 dB | **36.86** dB | ∞ |
|---|---|---|---|---|---|---|
| (a) Input | (b) GridNet | (c) MSBDN | (d) UHD | (e) DeHamer | (f) **PromptRestorer** | (g) GT |

Figure 6: **Image dehazing** example on SOTS-Indoor [52].

Table 4: **Image desnowing** results on CSD (2000) [65], SRRS (2000) [76], and Snow100K (2000) [73]. Our **PromptRestorer** achieves the best metrics on all datasets on the image desnowing problem.

| Benchmark | Metrics | DesnowNet [61] | JSTASR [14] | HDCW-Net [15] | TransWeather [85] | MSP-Former [13] | Uformer [94] | Restormer [101] | PromptRestorer |
|---|---|---|---|---|---|---|---|---|---|
| **CSD (2000)** [15] | PSNR ↑ | 20.13 | 27.96 | 29.06 | 31.76 | 33.75 | 33.80 | **35.43** | 37.48 |
| | SSIM ↑ | 0.81 | 0.88 | 0.91 | 0.93 | 0.96 | 0.96 | **0.97** | 0.99 |
| **SRRS (2000)** [14] | PSNR ↑ | 20.38 | 25.82 | 27.78 | 28.29 | 30.76 | 30.12 | **32.24** | 33.99 |
| | SSIM ↑ | 0.84 | 0.89 | 0.92 | 0.92 | 0.95 | 0.96 | **0.96** | 0.99 |
| **Snow100K (2000)** [61] | PSNR ↑ | 30.50 | 23.12 | 31.54 | 31.82 | 33.43 | 33.81 | **34.67** | 36.02 |
| | SSIM ↑ | 0.94 | 0.86 | 0.95 | 0.95 | **0.96** | 0.94 | 0.95 | 0.97 |

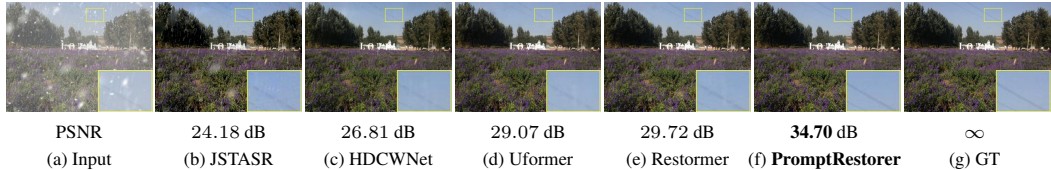

| PSNR | 24.18 dB | 26.81 dB | 29.07 dB | 29.72 dB | **34.70** dB | ∞ |
|---|---|---|---|---|---|---|
| (a) Input | (b) JSTASR | (c) HDCWNet | (d) Uformer | (e) Restormer | (f) **PromptRestorer** | (g) GT |

Figure 7: **Image desnowing** example on CSD (2000) [15].

**Image Dehazing Results.** We perform the image dehazing experiments on both synthetic benchmark RESIDE SOTS-Indoor [52], and real-world hazy benchmarks Dense-Haze [2] and NH-Haze [3]. Tab. 3 summarise the quantitative results. Compared to the recent works DeHamer [33] and MAXIM [84], our method receives 4.33 dB and 5.91 dB PSNR gains on the SOTS-Indoor, respectively. On the real-world benchmark NH-Haze [3], our PromptRestorer can achieve 0.7203 of the SSIM result, which is a new record and significantly outperforms current state-of-the-art approaches DeHamer [33]. The results on both synthetic and real-world benchmarks have demonstrated the effectiveness of our PromptRestorer on the image dehazing task. Fig. 6 shows the visual results, where our PromptRestorer is more effective in removing haze than other methods.

**Image Desnowing Results.** For the image desnowing task, we compare our PromptRestorer on the CSD [15], SRRS [14], and Snow100K [61] datasets with existing state-of-the-art methods [61, 14, 15, 13, 85]. We also compare recent Transformer-based general image restoration approaches Restormer [101] and Uformer [93]. As shown in Tab. 4, our PromptRestorer yields a 2.05 dB PSNR improvement over the state-of-the-art approach [101] on the CSD benchmark [15]. The visual results in Fig. 7 show that our PromptRestorer is able to remove spatially varying snowflakes than competitors.

Table 6: **Ablation experiments on PromptDPM**. Each component in L2P and G2P is effective.

(a) **Effect on L2P**. Both Res-InBan and Deg-InBan play positive roles for image restoration.

| Experiment | PSNR | FLOPs (G) | Params (M) |
|---|---|---|---|
| w/o L2P | 30.819 | 148.34 | 12.60 |
| w/o Res-InBan | 30.952 | 153.39 | 12.97 |
| w/o Deg-InBan | 30.964 | 153.39 | 12.97 |
| Full (*Ours*) | **31.015** | 157.04 | 13.24 |

(b) **Effect on G2P**. Both Q-InAtt and KV-InAtt play a positive effect on high-quality image restoration.

| Experiment | PSNR | FLOPs (G) | Params (M) |
|---|---|---|---|
| w/o G2P | 30.697 | 123.44 | 10.69 |
| w/o Q-InAtt | 30.914 | 148.93 | 12.65 |
| w/o KV-InAtt | 30.906 | 147.26 | 12.52 |
| Full (*Ours*) | **31.015** | 157.04 | 13.24 |

## 4.2 Analysis and Discussion

For ablation experiments, following [84, 20], we train the image deblurring model on GoPro dataset [65] for 1000 epochs only and set the number of Transformer in each CGT is 1. Params mean the number of learnable parameters. Testing is performed on the GoPro testing dataset [65]. FLOPs are computed on image size $256\times256$. Next, we describe the influence of each component individually.

**Effect on Prompting.** The core design of our PromptRestorer is the 'prompting', which exploits a pre-trained model to extract raw degradation features from the degraded observations and then generate perceived content to guide the restoration branch (i.e., **Case 3** in Fig. 1). Compared to existing frameworks such as **Cases 1-2** in Fig. 1, our proposed prompting strategy shows superior performance, as demonstrated in Tab. 5. Our method achieved 0.877 dB gains compared to **Case 1**, and 0.369 dB higher than **Case 2**. Interestingly, the learnable condition branch in **Case 2**[2], despite consuming more FLOPs and Params, results in worse performance than ours. Our approach directly exploits raw degradation features to prompt restoration with persistent degradation priors to facilitate better recovery. Fig. 1(c) shows two examples, where our model that exploits raw degradation features as prompting generates sharper and clearer images.

Table 5: **Effect on prompting**. Our method that directly exploits the raw degradation to prompt restoration performs better.

| Case in Fig. 1 | PSNR | FLOPs (G) | Params (M) |
|---|---|---|---|
| 1 | 30.138 | 105.58 | 10.16 |
| 2 | 30.646 | 157.04 | 16.77 |
| 3 (*Ours*) | **31.015** | 157.04 | 13.24 |

**Effect on PromptDPM.** We analyze the impact of PromptDPM on restoration quality in Tab. 6 by disabling one component at a time. Each model in L2P and G2P consumes similar Params and FLOPs, while our full model achieves the best performance. Disabling L2P or G2P results in a decrease in performance by 0.196 dB and 0.318 dB, respectively. These experiments conclusively demonstrate the effectiveness of each component in L2P and G2P for restoration.

**Effect on GDP.** To understand the impact of GDP, we disable it to compare with full model in Tab. 7. Note that the computational cost of the GDP is negligible compared to disabling it as it only involves a $1\times1$ convolution and sigmoid function for the gating mechanism, while it leads to a gain of 0.091 dB. This finding highlights the significance of controlling the propagation of the perceived degradation features.

Table 7: **Effect on GDP**. Our GDP which controls the degradation propagation is effective.

| Experiment | PSNR | FLOPs (G) | Params (M) |
|---|---|---|---|
| w/o GDP | 30.924 | 154.60 | 12.95 |
| w/ GDP (*Ours*) | **31.015** | 157.04 | 13.24 |

## 4.3 Visualization Understanding for Degradation Vanishing

To emphasize the understanding of degradation vanishing, we visualize the features learned in the condition/prompting branches to better comprehend the learned status of these branches in Fig. 8. Notably, both **Cases 1-2** exhibit sharper results in later iterations compared to earlier ones, which fail to provide the restoration branch with sufficient degraded information and cause the restoration

---

[2]We ensure fairness in the comparison by using the same network architecture in the condition branch of **Case 2** and the encoder of the pre-trained model in our network.

models to not perceive the degradation well, thereby hindering the model capacity. In contrast, as the restoration branch needs to adapt perceived features from the PromptDPM which is to perceive the raw degradation features from inputs, our model (**Case 3**) initially exhibits inferior performance (around 20K iterations) as shown in Fig. 1(b). However, with better adaptation to the degradation information after more iterations, the prompting branch can better prompt the restoration branch consistently with more reliable perceived content learned from the raw degradation, enabling our restoration branch to overcome degradation vanishing and improve restoration quality, as shown in Fig. 1(c).

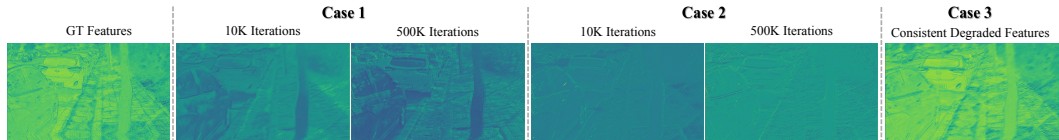

Figure 8: **Visualization**. We show the average features over the channel dimension in the condition/prompting branches for the second example in Fig. 1(c). We obtain GT/degraded features by inputting GT/degraded images into the pre-trained VQGAN. As single-branch models (**Case 1** in Fig. 1) do not have condition branches, we visualize the last layer in the 1-level encoder for reference.

## 5 Concluding Remarks

In this paper, we investigate the degradation vanishing in the learning process for image restoration. To solve this problem, we have proposed the PromptRestorer which explores the raw degradation features extracted by a pre-trained model from the given degraded observations to guide the restoration process to facilitate better recovery. Extensive experiments have demonstrated that our PromptRestorer favors against state-of-the-art approaches on 4 restoration tasks, including image deraining, deblurring, dehazing, and desnowing.

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
