# PromptRestorer: A Prompting Image Restoration Method with Degradation Perception —Supplementary Materials—

**Cong Wang**[1]*, **Jinshan Pan**[2]*, **Wei Wang**[3]*, **Jiangxin Dong**[2], **Mengzhu Wang**[4],
**Yakun Ju**[1], **Junyang Chen**[5]
[1]The Hong Kong Polytechnic University, [2]Nanjing University of Science and Technology,
[3]Dalian University of Technology, [4]Hebei University of Technology, [5]Shenzhen University

## Abstract

In this supplementary material, we first describe the datasets used in various restoration tasks and experimental details in Sec. 1. Then, we demonstrate the superiority of the proposed PromptDMP compared with commonly used feature fusion module-Spatial Feature Transform (SFT) [27] in Sec. 2. And then, we validate the effectiveness of the used improved ConvNeXt compared with ConvNeXt [21] in Sec. 3. Next, we present the detailed structure of the pre-trained encoder in Sec. 4. Last, we provide additional visual results compared to state-of-the-art approaches on various image restoration tasks in Sec. 5.

## 1 Datasets Description and Experimental Details

For different restoration tasks, the training and testing datasets are described in Tab. 1.

**Image Deraining:** In accordance with [15, 31, 8], we utilize a composite training dataset consisting of 13,712 image pairs collected from various training datasets, namely Rain100H [30], Rain100L [30], Test100 [34], Test1200 [33], and Test2800 [10]. PromptRestorer is evaluated on Rain100H [30], Rain100L [30], Test100 [34], Test1200 [33], and Test2800 [10]. Our PromptRestorer model is trained for 600 epochs.

**Image Deblurring:** Consistent with recent methods [29, 31], we train PromptRestorer using the GoPro dataset [22]. The GoPro dataset comprises 2,103 blurry/sharp image pairs for training and 1,111 pairs for evaluation. To assess the generalization capacity of our method, we directly apply the GoPro-trained model to the HIDE dataset [24] and the real-world benchmark RSBlur [23]. Our network is trained for 3,000 epochs on the GoPro dataset.

**Image Dehazing:** For dehazing, we train PromptRestorer on the commonly used RESIDE dataset [16]. We employ the ITS dataset for training and use SOTS-indoor as the testing dataset. PromptRestorer is trained for 600 epochs on ITS. Additionally, following [12], we include the real-world datasets Dense-Haze [1] and NH-Haze [2] as experimental datasets, containing 45 training images and 5 testing images, respectively. The network is trained on Dense-Haze and NH-Haze for 3,000 epochs.

**Image Desnowing:** For desnowing, we utilize the CSD [6], SRRS [5], and Snow100K [20] datasets. Following previous works [4], we randomly sample 2,500 image pairs from the training set for training and 2,000 images from the testing set for evaluation. The model is trained for 1,000 epochs on each dataset.

---

*These authors equally contribute to this work.

37th Conference on Neural Information Processing Systems (NeurIPS 2023).

Table 1: **Dataset description** for various image restoration tasks.

| Tasks | Datasets | Train Samples | Test Samples | Testset Rename |
|---|---|---|---|---|
| **Deraining** | Rain14000 [10] | 11200 | 2800 | Test2800 |
| | Rain1800 [30] | 1800 | 0 | - |
| | Rain800 [34] | 700 | 100 | Test100 |
| | Rain100H [30] | 0 | 100 | Rain100H |
| | Rain100L [30] | 0 | 100 | Rain100L |
| | Rain1200 [33] | 0 | 1200 | Test1200 |
| | Rain12 [18] | 12 | 0 | - |
| **Deblurring** | GoPro [22] | 2103 | 1111 | - |
| | HIDE [24] | 0 | 2025 | - |
| | RSBlur [23] | 0 | 1960 | - |
| **Dehazing** | RESIDE/ITS [16] | 13990 | 500 | SOTS-Indoor |
| | Dense-Haze [1] | 45 | 5 | - |
| | NH-Haze [2] | 45 | 5 | - |
| **Desnowing** | CSD [6] | 2500 | 2000 | CSD (2000) |
| | SRRS [5] | 2500 | 2000 | SRRS (2000) |
| | Snow100K [20] | 2500 | 2000 | Snow100K (2000) |

## 2  PromptDMP vs. Spatial Feature Transform (SFT) [27]

Spatial Feature Transform (SFT) [27] which usually generates affine transformation parameters for spatial-wise feature modulation is a commonly used feature fusion module to bridge the feature transformation between conditional features and restoration features [28, 13, 11, 7]. Hence, one may wonder to know whether our proposed PromptDMP is more effective than SFT or not. To validate this, we replace our PromptDMP with SFT to conduct ablation experiments. Tab. 2 shows that our PromptDMP improves the restoration quality by $0.316$ dB compared with the results by SFT. It clearly demonstrates that our PromptDMP is more effective than SFT thanks to the designs of global and local prompting perception.

Table 2: **PromptDMP vs. SFT.** Our PromptDMP is more effective than the commonly used SFT [27].

| Experiment | PSNR | FLOPs (G) | Params (M) |
|---|---|---|---|
| w/ SFT [27] | 30.699 | 108.14 | 10.36 |
| w/ PromptDMP (***Ours***) | **31.015** | 146.36 | 13.24 |

## 3  Improved ConvNeXt vs. ConvNeXt [21]

We in this paper employ the improved ConvNeXt by using two depth-wise convolutions with $3 \times 3$ kernel sizes instead of ConvNeXt [21] which uses a depth-wise convolution with $7 \times 7$ kernel sizes. To validate the effectiveness of improved ConvNeXt, we use the ConvNeXt [21] to replace our used improved ConvNeXt, and the results are reported in Tab. 3. It shows that our improved ConvNeXt is more effective. This also reveals that stacking small kernel convolutions can learn better representations than a large kernel convolution with the equivalent receptive field for image restoration.

Table 3: **Improved ConvNeXt vs. ConvNeXt.** Our improved ConvNeXt can help produce high-quality restoration results compared with ConvNeXt [27].

| Experiment | PSNR | FLOPs (G) | Params (M) |
|---|---|---|---|
| w/ ConvNeXt [21] | 30.354 | 144.28 | 12.97 |
| w/ Improved ConvNeXt (***Ours***) | **31.015** | 146.36 | 13.24 |

# 4 Network Architecture of Pre-trained Encoder

We further specify the architecture of the encoder in Tab. 4. Note that the learnable conditions have the same encoder architecture as our pre-trained encoder for fair comparisons. The training process of our pre-trained VQGAN method follows [3].

Table 4: **Encoder architecture.** The encoder mainly consists of convolution layers, residual blocks [14], and downsample operation. The encoder gradually encodes input images to spatially-reduced features.

| Layer | Encoder1 | Downsample | Encoder2 | Downsample | Encoder3 |
|---|---|---|---|---|---|
| Shape | $48 \times H \times W$ | $48 \times \frac{H}{2} \times \frac{W}{2}$ | $96 \times \frac{H}{2} \times \frac{W}{2}$ | $96 \times \frac{H}{4} \times \frac{W}{4}$ | $192 \times \frac{H}{4} \times \frac{W}{4}$ |
| Blocks | 1 Conv & 2 ResBlocks | - | 1 Conv & 2 ResBlocks | - | 1 Conv & 2 ResBlocks |
| Kernel sizes | $3 \times 3$ | - | $3 \times 3$ | - | $3 \times 3$ |
| Stride | 1 | - | 1 | - | 1 |

# 5 Addtional Visual Results

We provide more visual results on various image restoration tasks in Fig. 1-Fig. 38.

**Image Deraining Results:** Fig. 1-Fig. 4.

**Image Deblurring Results:** Fig. 5-Fig. 7.

**Image Dehazing Results:** Fig. 8-Fig. 24.

**Image Desnowing Results:** Fig. 25-Fig. 38.

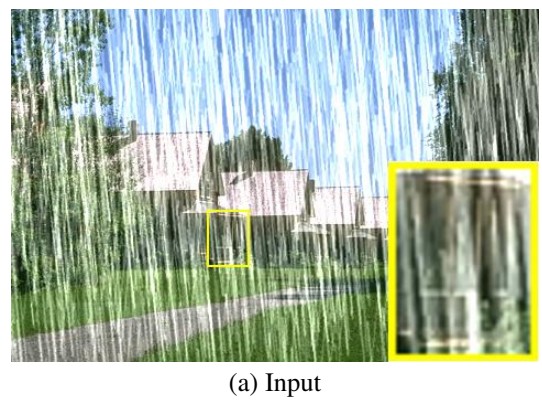

(a) Input

(b) RESCAN [17]

(c) DCSFN [26]

(d) MPRNet [32]

(e) Restormer [31]

(f) PromptRestorer

(g) GT

Figure 1: **Image deraining** example on Rain100H [30].

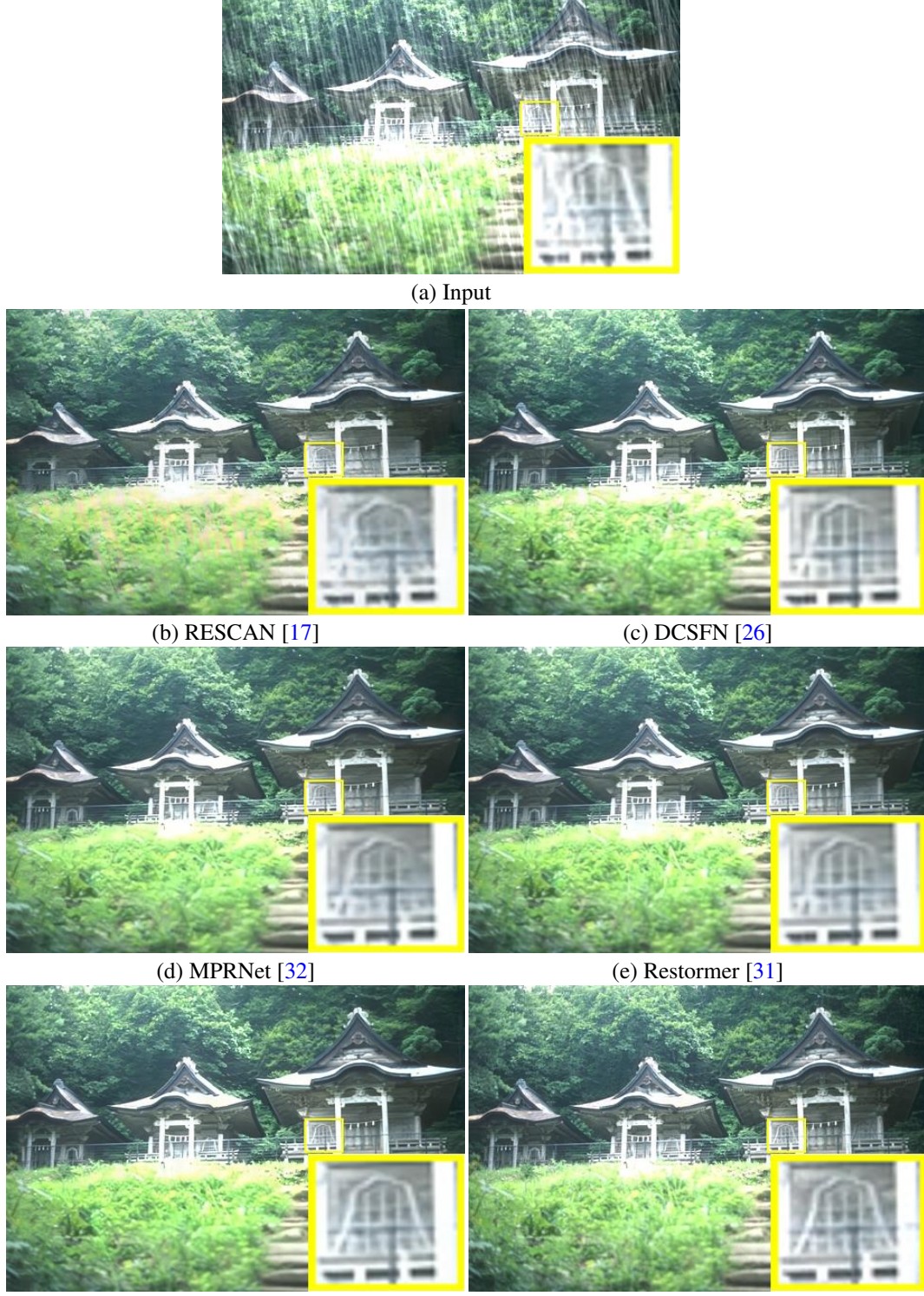

(a) Input

(b) RESCAN [17]

(c) DCSFN [26]

(d) MPRNet [32]

(e) Restormer [31]

(f) PromptRestorer

(g) GT

Figure 2: **Image deraining** example on Rain100H [30].

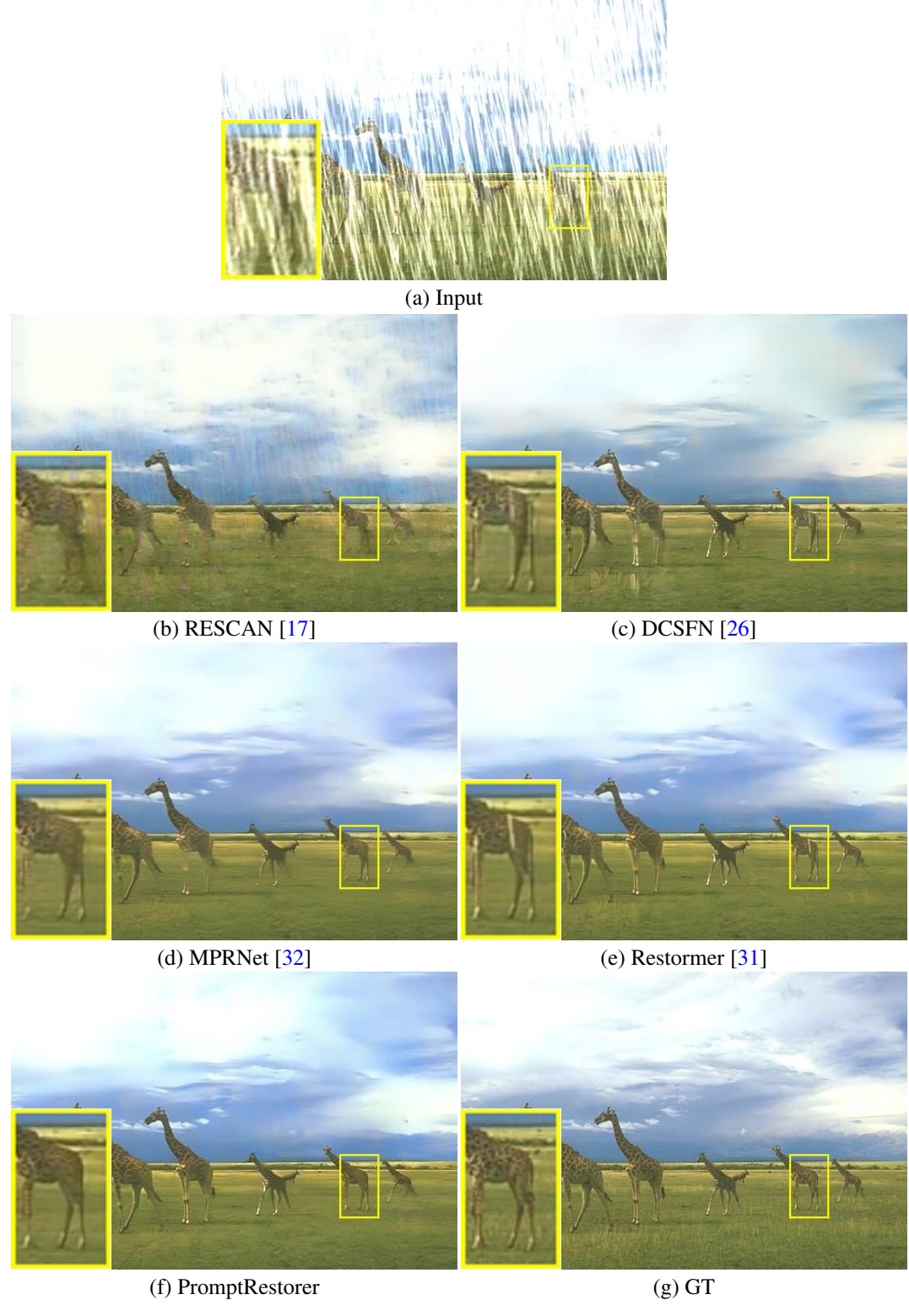

(a) Input

(b) RESCAN [17]

(c) DCSFN [26]

(d) MPRNet [32]

(e) Restormer [31]

(f) PromptRestorer

(g) GT

Figure 3: **Image deraining** example on Rain100H [30].

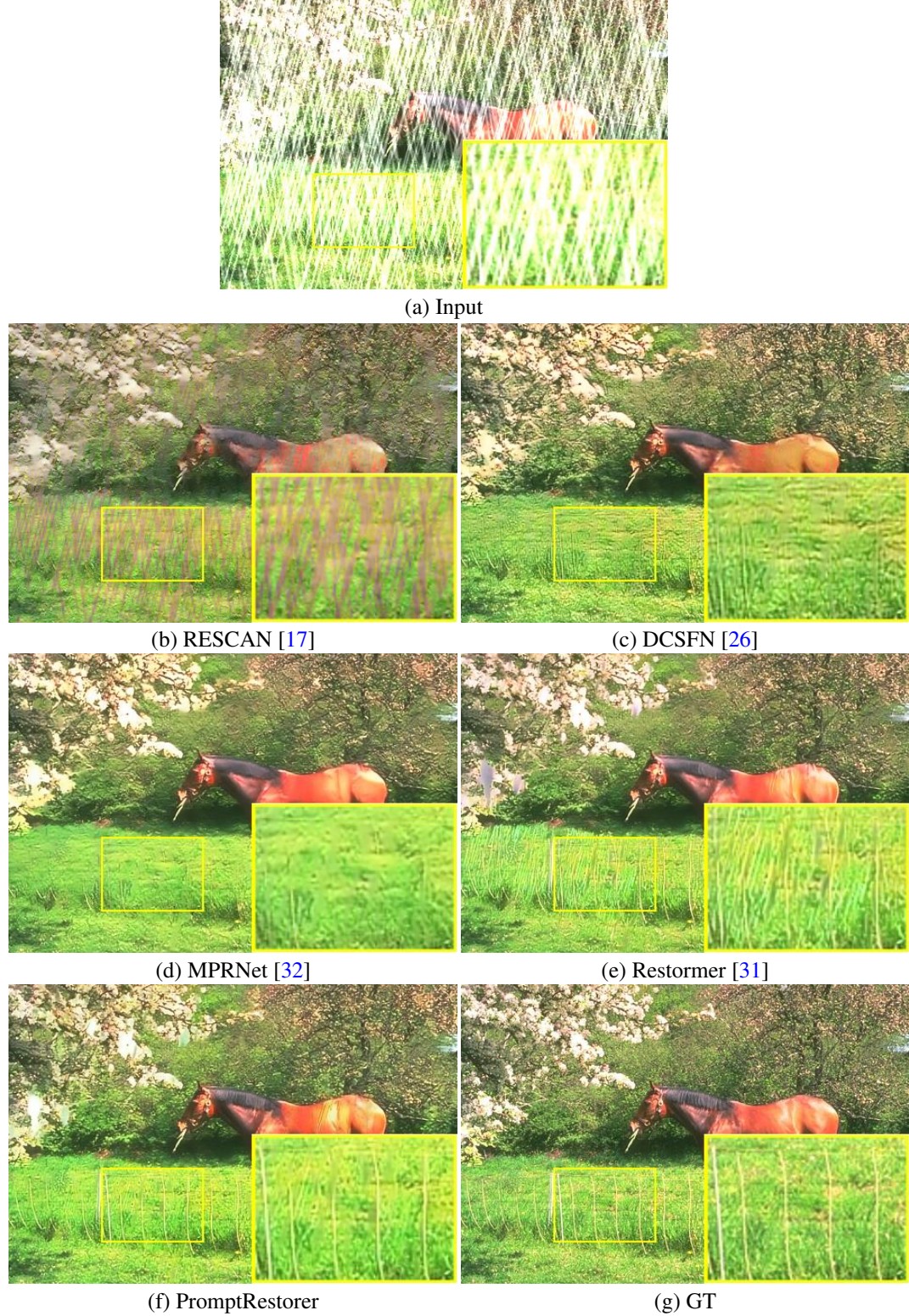

(a) Input

(b) RESCAN [17]

(c) DCSFN [26]

(d) MPRNet [32]

(e) Restormer [31]

(f) PromptRestorer

(g) GT

Figure 4: **Image deraining** example on Rain100H [30].

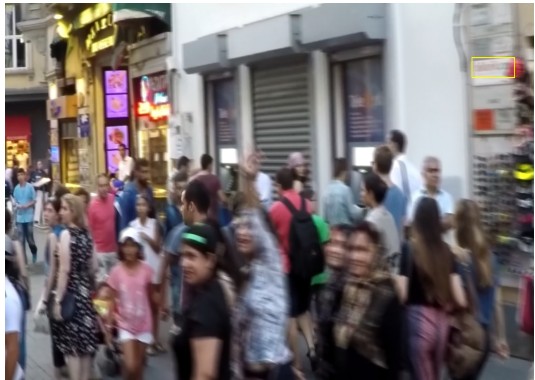

(a) Degraded Image

(b) Degraded Patch

(c) Nah et al. [22]

(d) SRN [25]

(f) MPRNet [32]

(g) PromptRestorer

(h) GT

Figure 5: **Image blurring** example on GoPro [22].

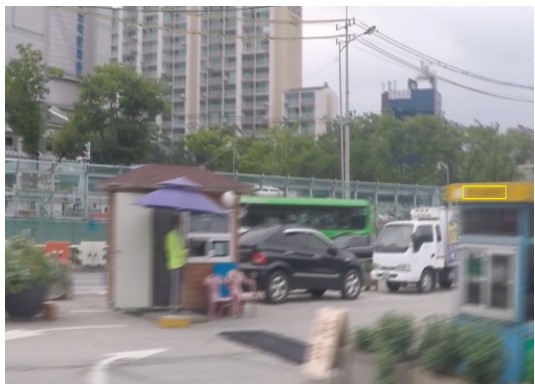

(a) Degraded Image

(b) Degraded Patch                    (c) Nah et al. [22]

(d) SRN [25]                          (f) MPRNet [32]

(g) PromptRestorer                    (h) GT

Figure 6: **Image blurring** example on GoPro [22].

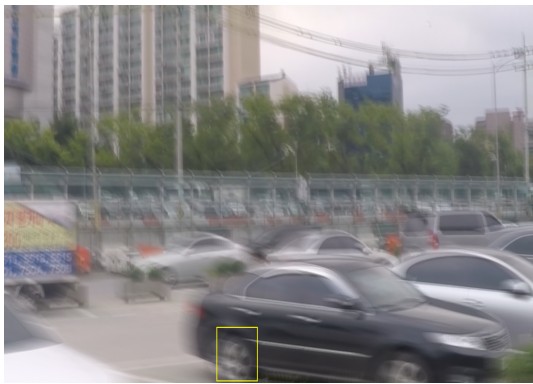
(a) Degraded Image

(b) Degraded Patch                    (c) Nah et al. [22]

(d) SRN [25]                          (f) MPRNet [32]

(g) PromptRestorer                    (h) GT

Figure 7: **Image blurring** example on GoPro [22].

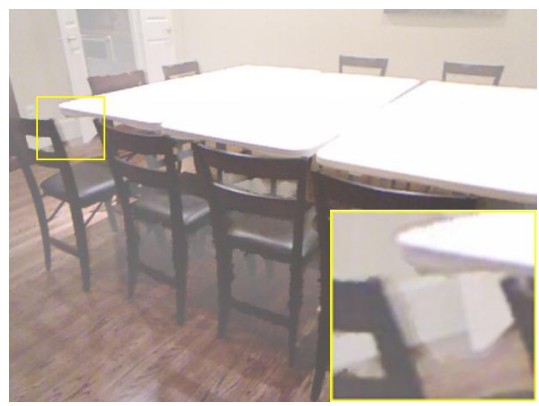

(a) Input

(b) GridNet [19]

(c) MSBDN [9]

(d) UHD [35]

(e) DeHamer [12]

(f) PromptRestorer

(g) GT

Figure 8: **Image dehazing** example on SOTS-Indoor [16].

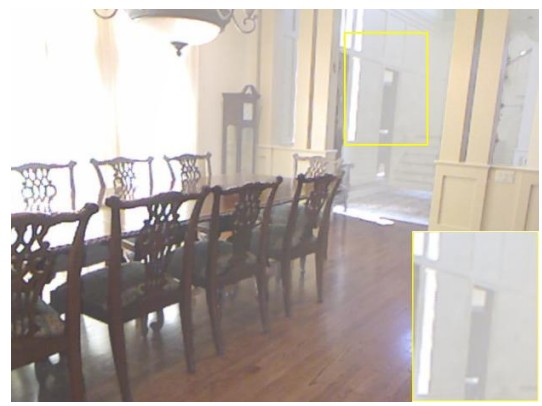

(a) Input

(b) GridNet [19]

(c) MSBDN [9]

(d) UHD [35]

(e) DeHamer [12]

(f) PromptRestorer

(g) GT

Figure 9: **Image dehazing** example on SOTS-Indoor [16].

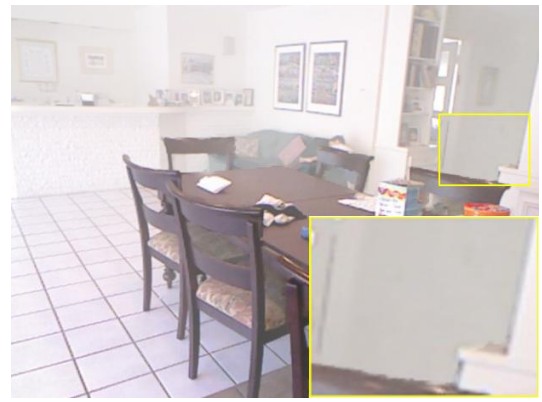

(a) Input

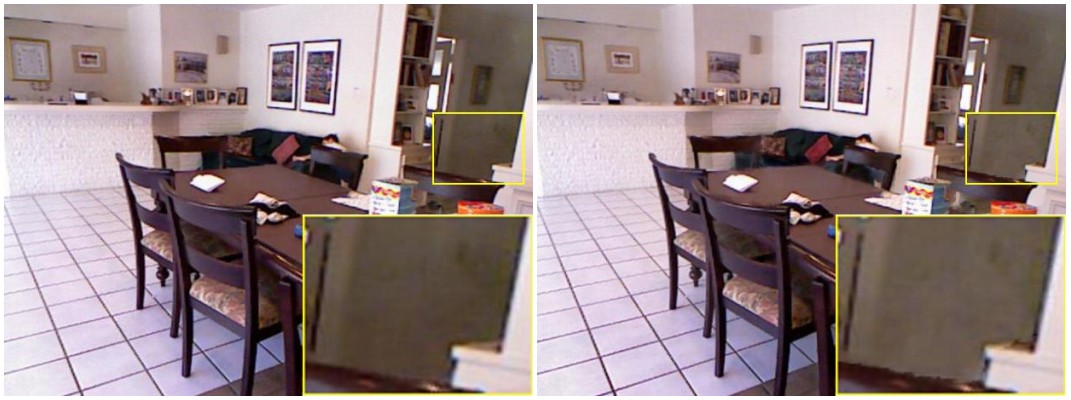

(b) GridNet [19]           (c) MSBDN [9]

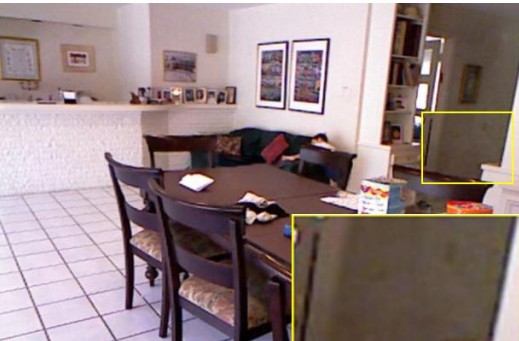

(d) UHD [35]           (e) DeHamer [12]

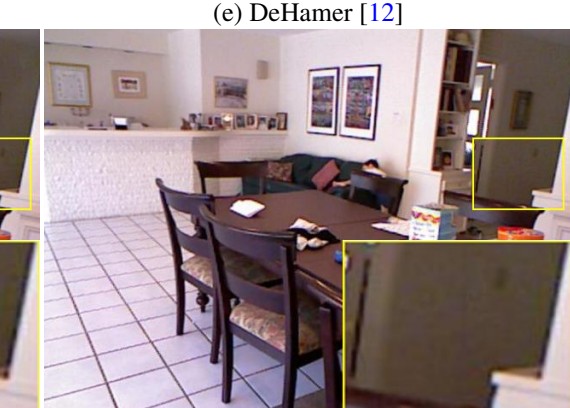

(f) PromptRestorer           (g) GT

Figure 10: **Image dehazing** example on SOTS-Indoor [16].

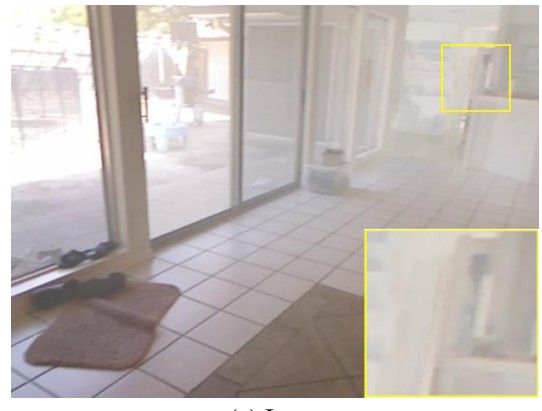

(a) Input

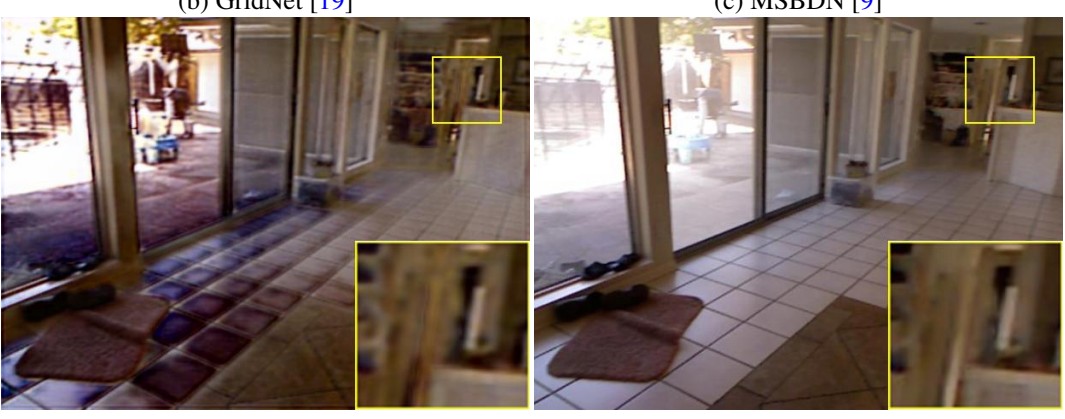

(b) GridNet [19]             (c) MSBDN [9]

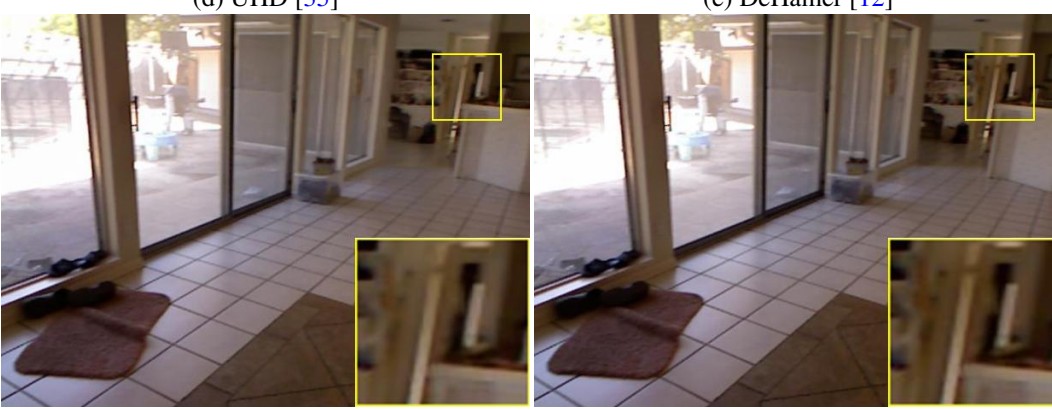

(d) UHD [35]             (e) DeHamer [12]

(f) PromptRestorer            (g) GT

Figure 11: **Image dehazing** example on SOTS-Indoor [16].

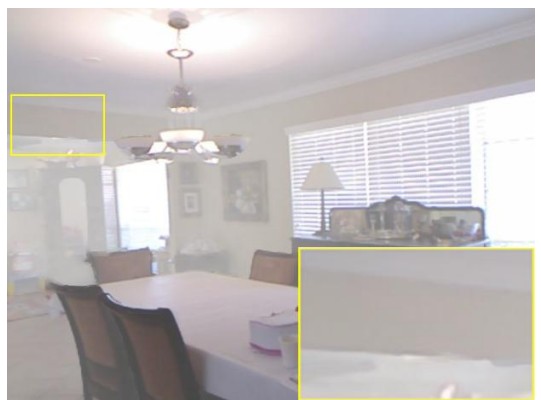

(a) Input

(b) GridNet [19]

(c) MSBDN [9]

(d) UHD [35]

(e) DeHamer [12]

(f) PromptRestorer

(g) GT

Figure 12: **Image dehazing** example on SOTS-Indoor [16].

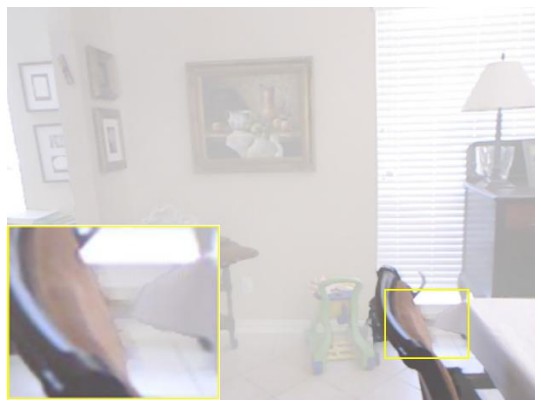

(a) Input

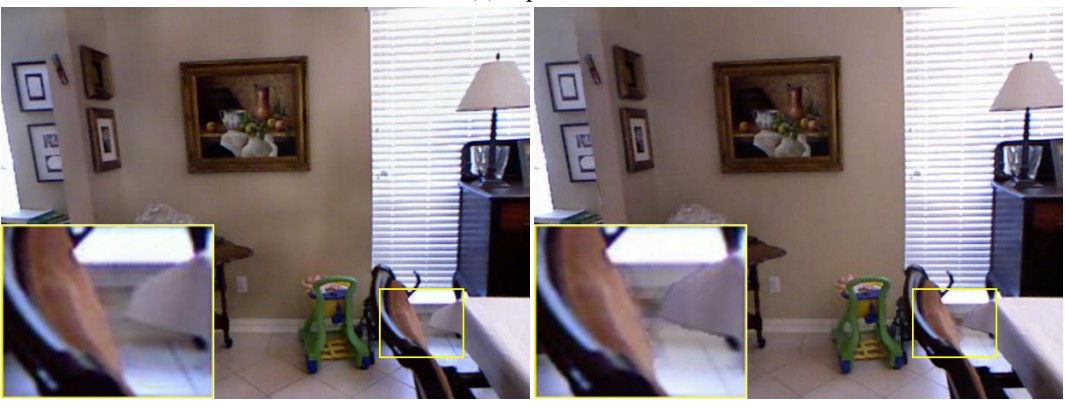

(b) GridNet [19]                                           (c) MSBDN [9]

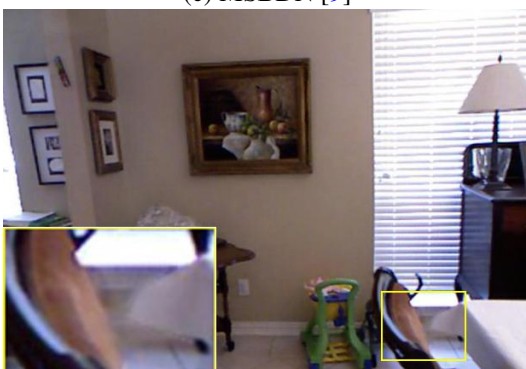

(d) UHD [35]                                               (e) DeHamer [12]

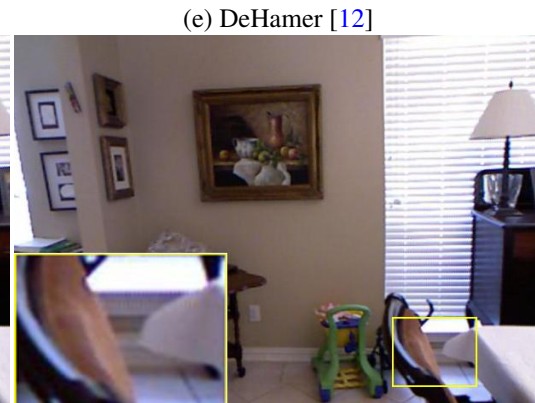

(f) PromptRestorer                                         (g) GT

Figure 13: **Image dehazing** example on SOTS-Indoor [16].

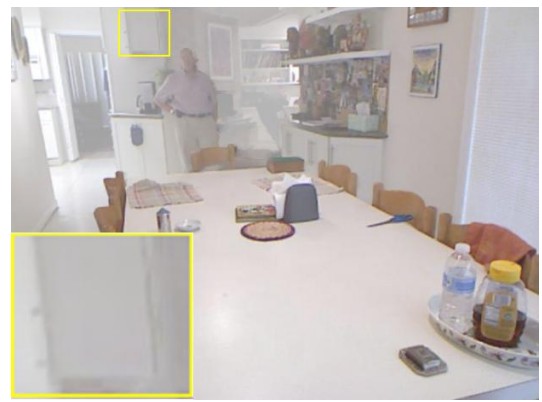

(a) Input

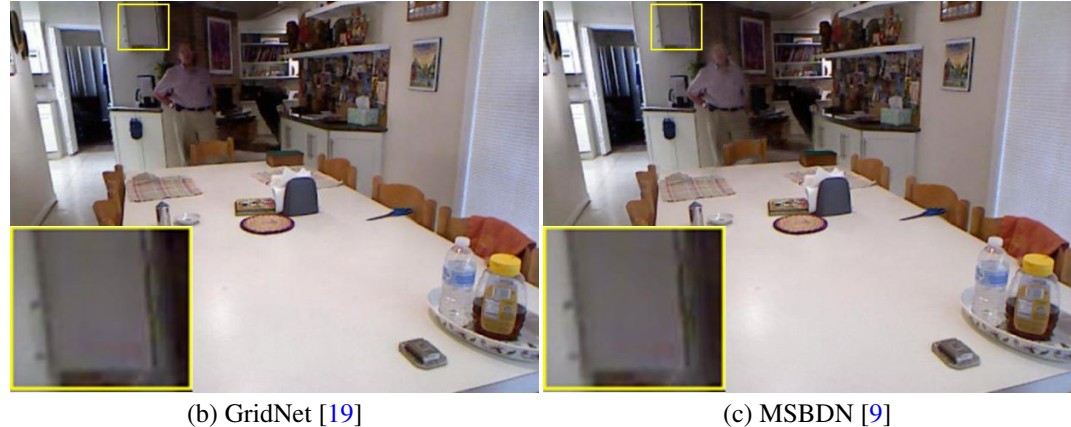

(b) GridNet [19]           (c) MSBDN [9]

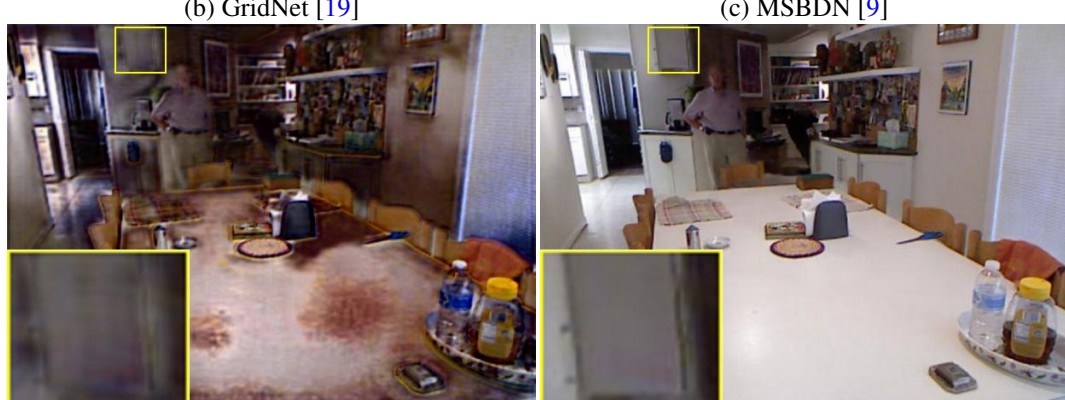

(d) UHD [35]           (e) DeHamer [12]

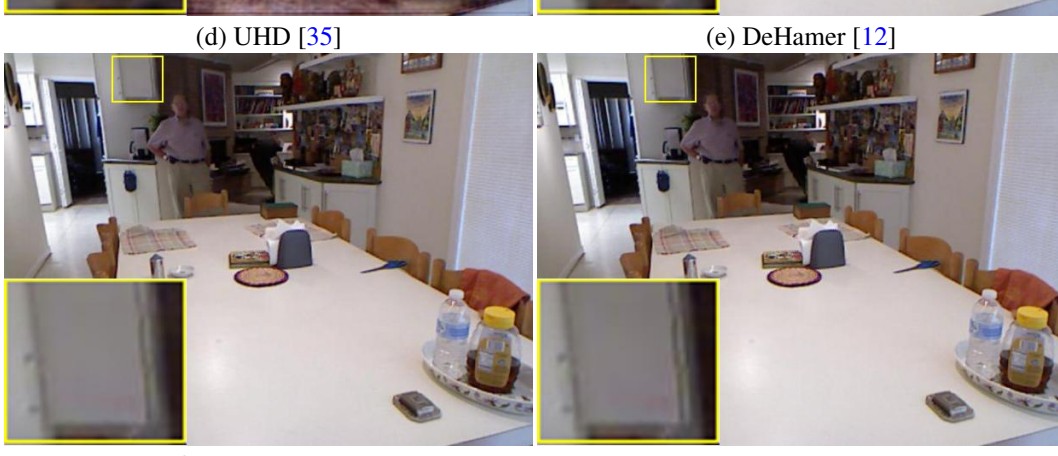

(f) PromptRestorer           (g) GT

Figure 14: **Image dehazing** example on SOTS-Indoor [16].

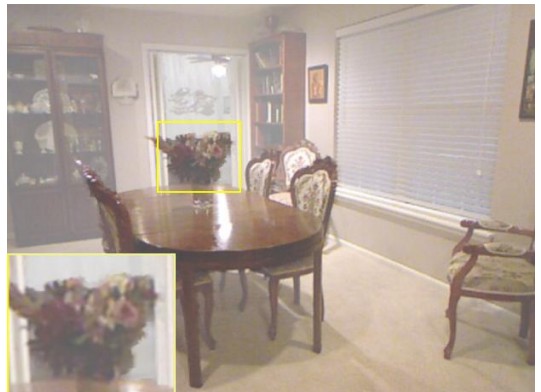

(a) Input

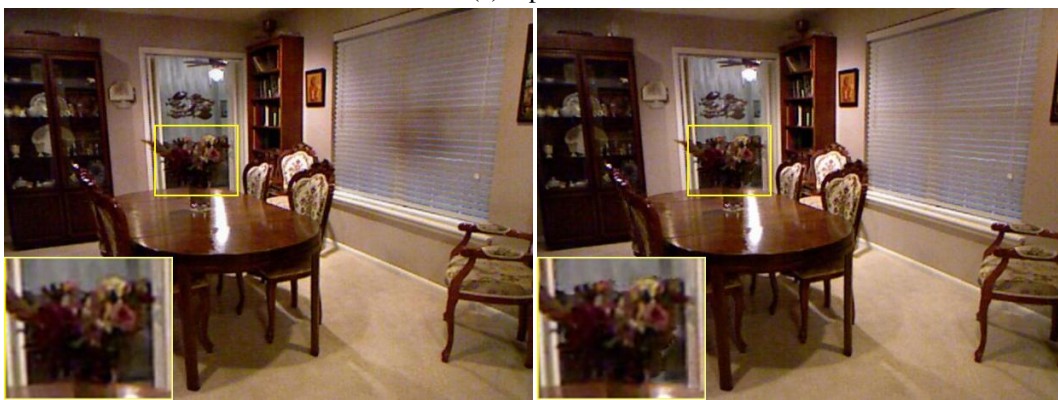

(b) GridNet [19]                   (c) MSBDN [9]

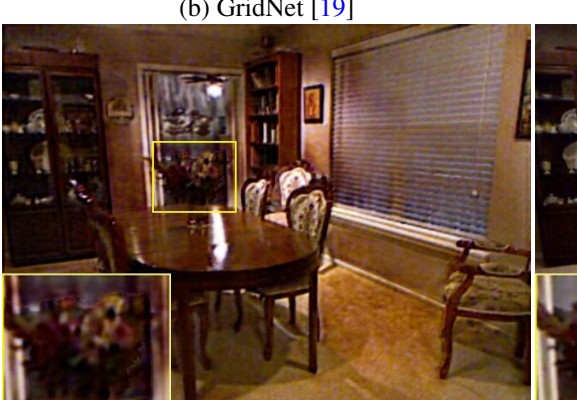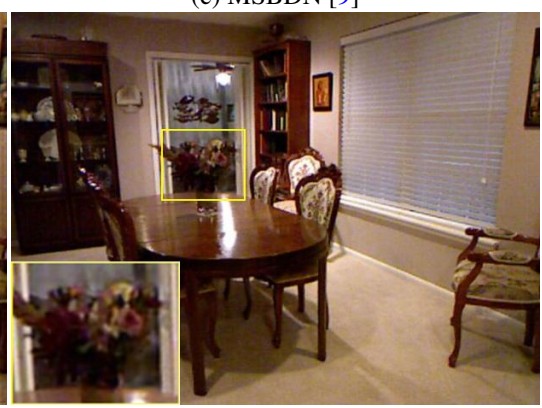

(d) UHD [35]                   (e) DeHamer [12]

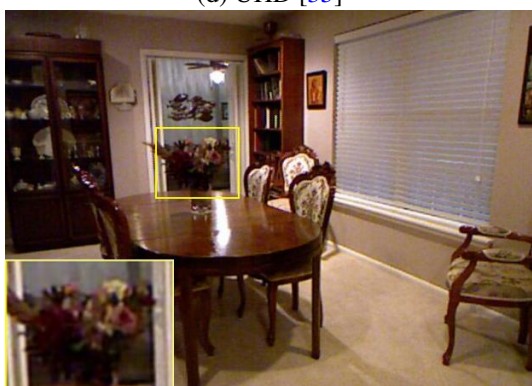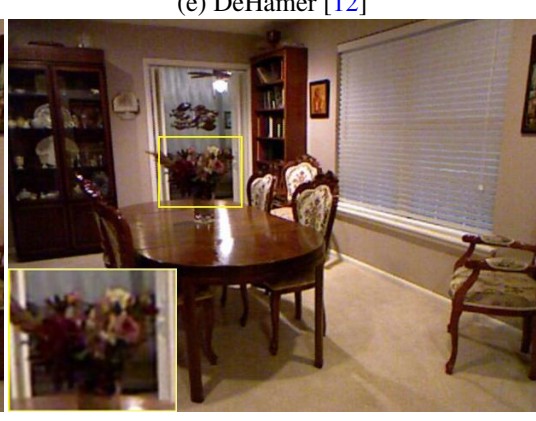

(f) PromptRestorer                   (g) GT

Figure 15: **Image dehazing** example on SOTS-Indoor [16].

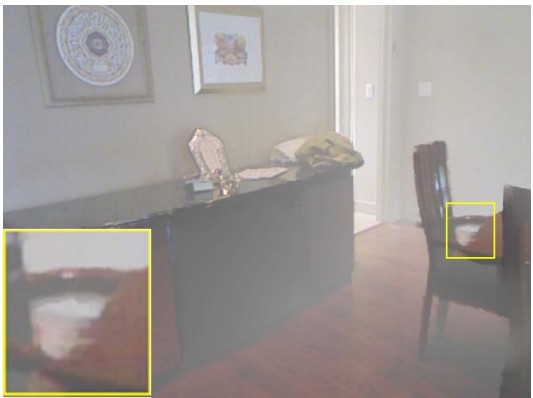

(a) Input

(b) GridNet [19]

(c) MSBDN [9]

(d) UHD [35]

(e) DeHamer [12]

(f) PromptRestorer

(g) GT

Figure 16: **Image dehazing** example on SOTS-Indoor [16].

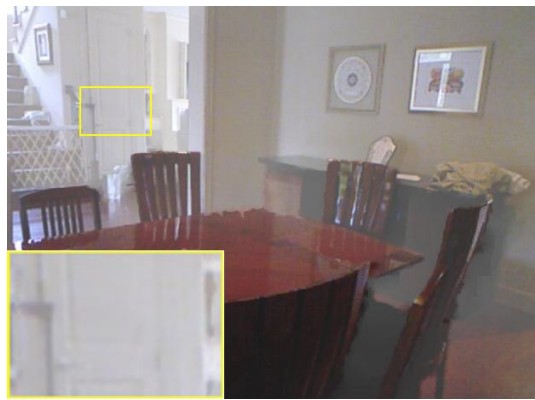

(a) Input

(b) GridNet [19]

(c) MSBDN [9]

(d) UHD [35]

(e) DeHamer [12]

(f) PromptRestorer

(g) GT

Figure 17: **Image dehazing** example on SOTS-Indoor [16].

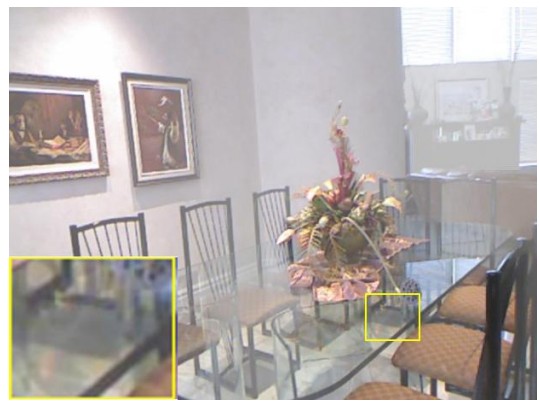
(a) Input

(b) GridNet [19]

(c) MSBDN [9]

(d) UHD [35]

(e) DeHamer [12]

(f) PromptRestorer

(g) GT

Figure 18: **Image dehazing** example on SOTS-Indoor [16].

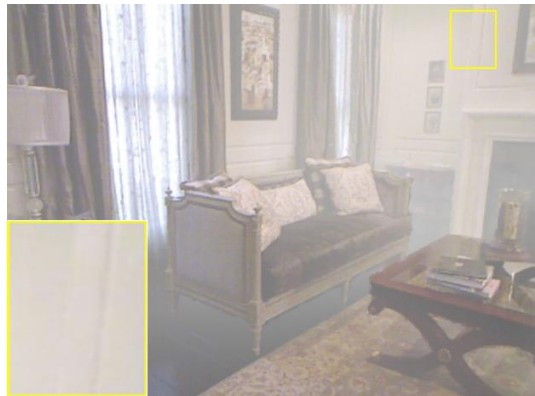

(a) Input

(b) GridNet [19]

(c) MSBDN [9]

(d) UHD [35]

(e) DeHamer [12]

(f) PromptRestorer

(g) GT

Figure 19: **Image dehazing** example on SOTS-Indoor [16].

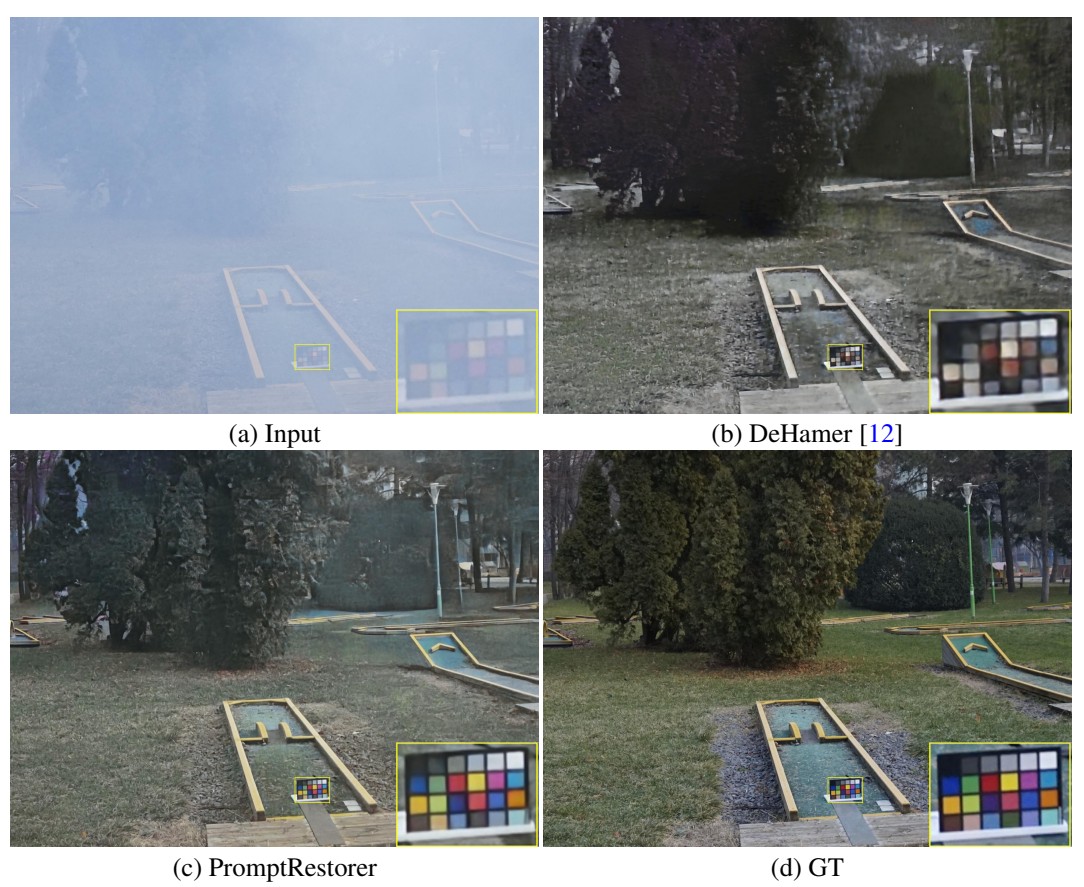

(a) Input

(b) DeHamer [12]

(c) PromptRestorer

(d) GT

Figure 20: **Image dehazing** example on Dense-Haze [1].

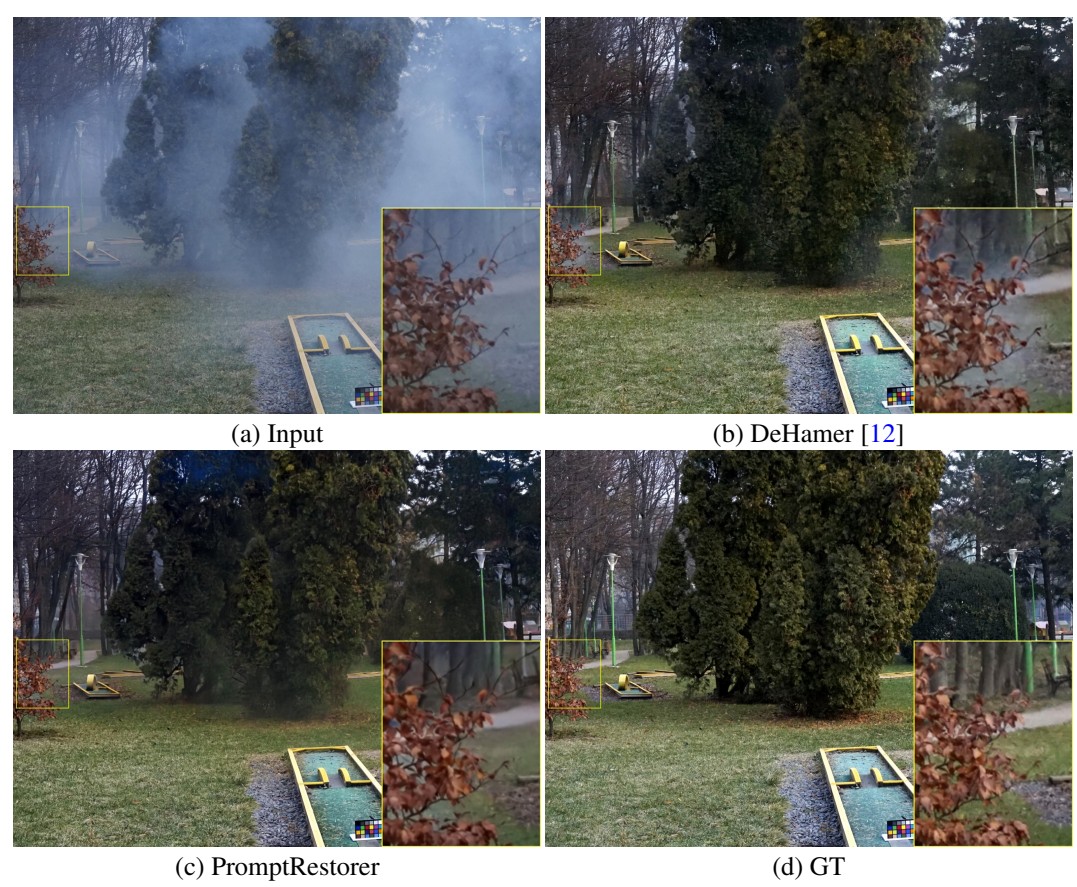

(a) Input

(b) DeHamer [12]

(c) PromptRestorer

(d) GT

Figure 21: **Image dehazing** example on NH-Haze [2].

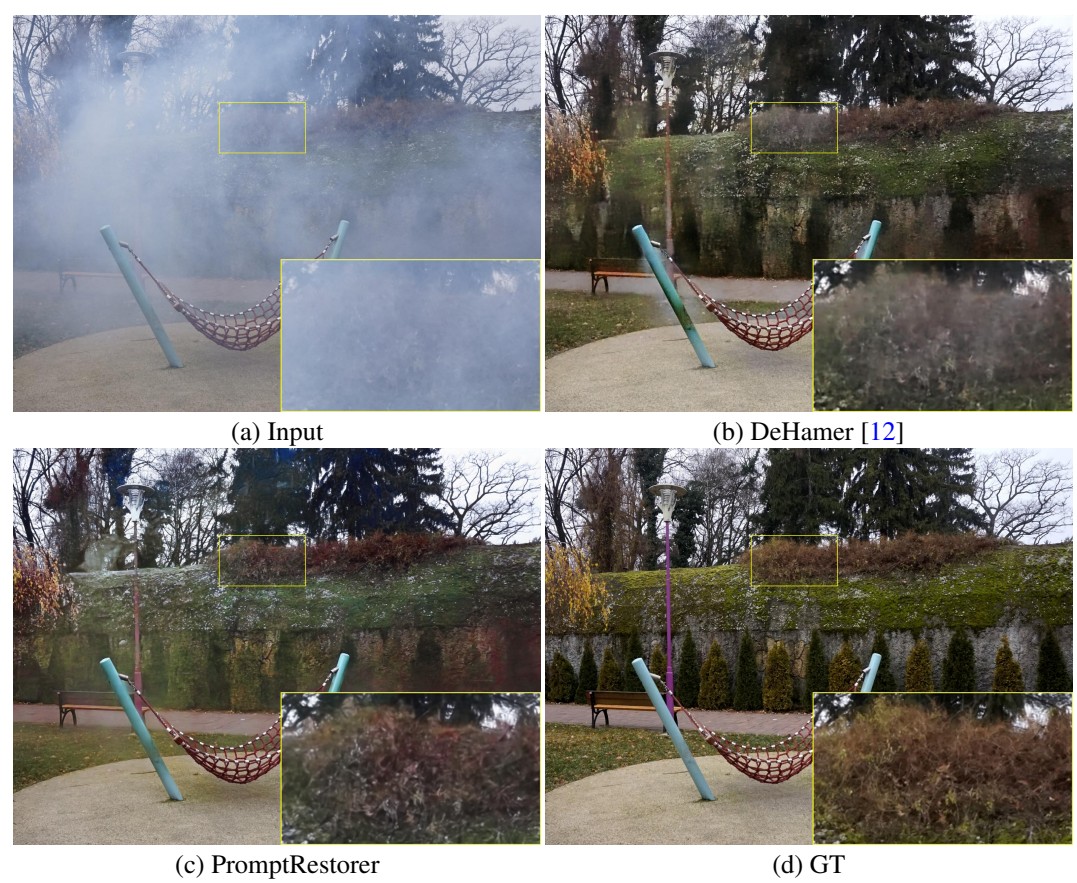

(a) Input

(b) DeHamer [12]

(c) PromptRestorer

(d) GT

Figure 22: **Image dehazing** example on NH-Haze [2].

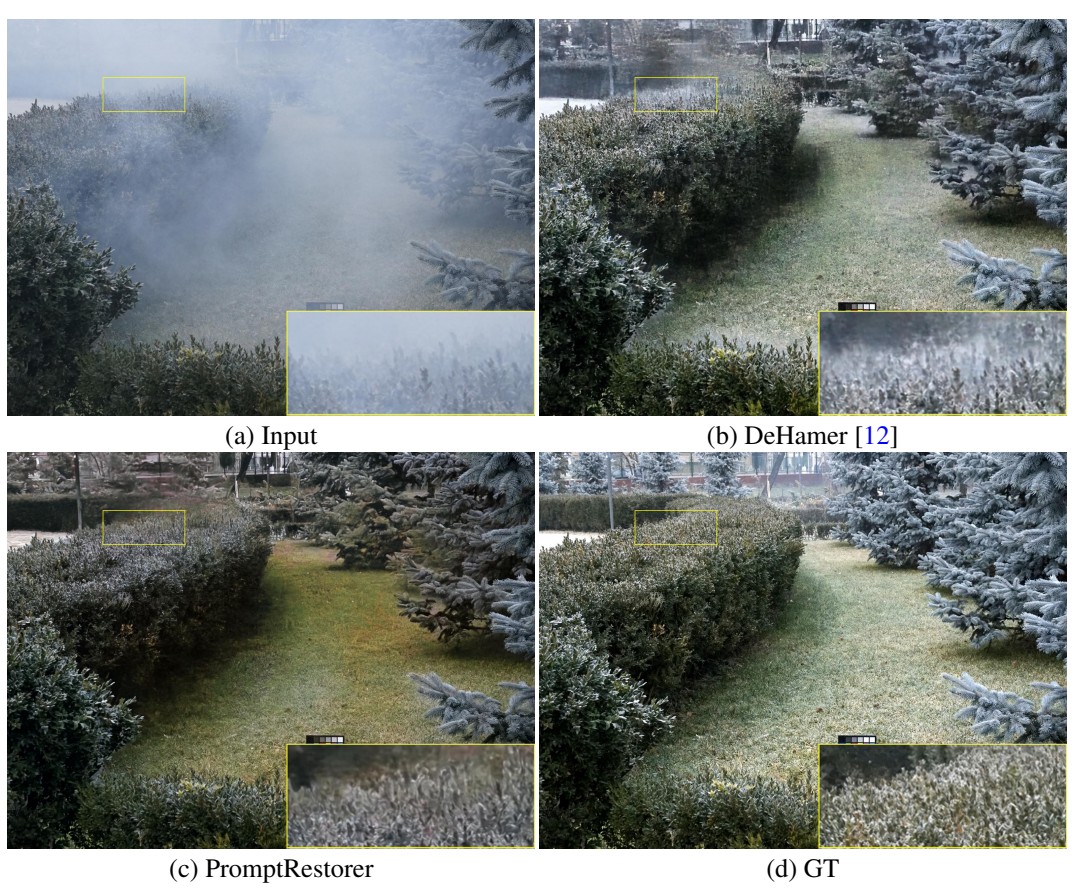

(a) Input

(b) DeHamer [12]

(c) PromptRestorer

(d) GT

Figure 23: **Image dehazing** example on NH-Haze [2].

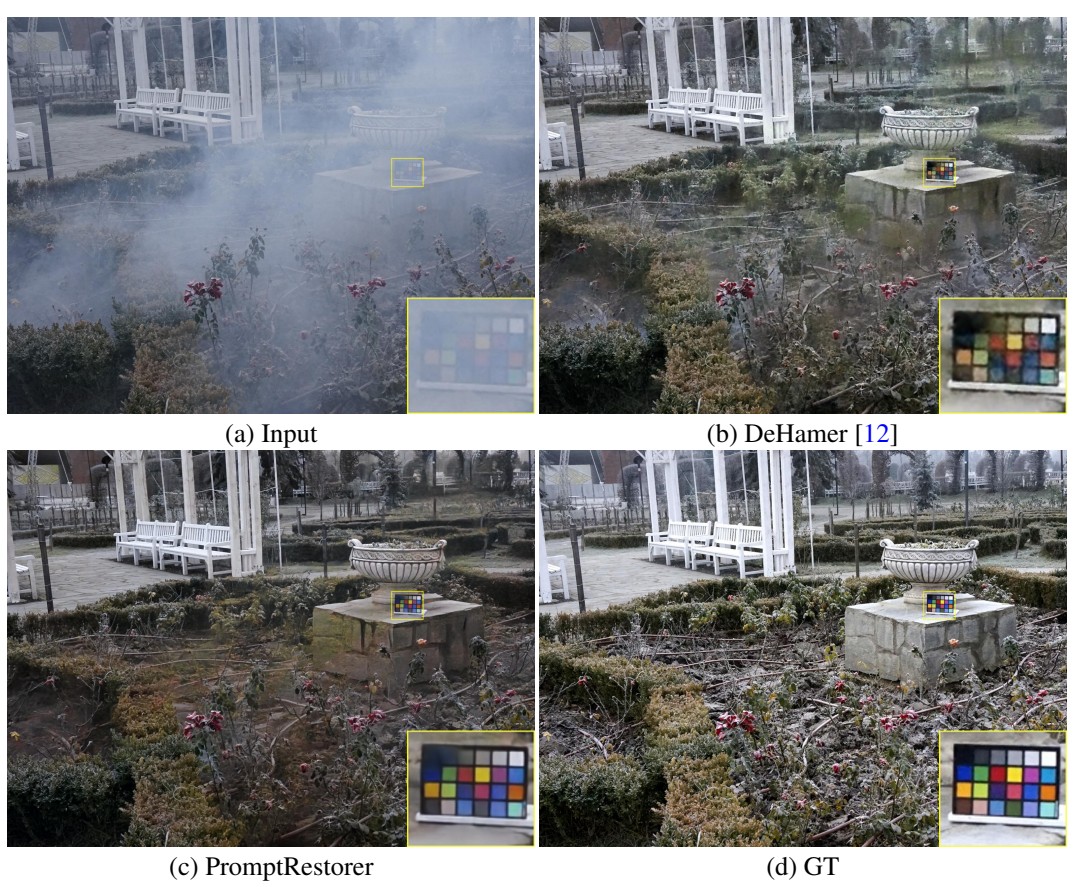

(a) Input

(b) DeHamer [12]

(c) PromptRestorer

(d) GT

Figure 24: **Image dehazing** example on NH-Haze [2].

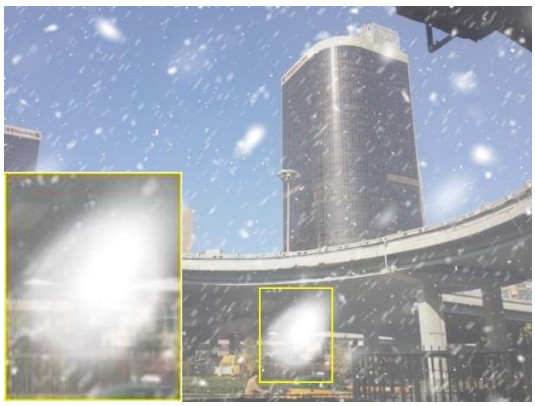

(a) Input

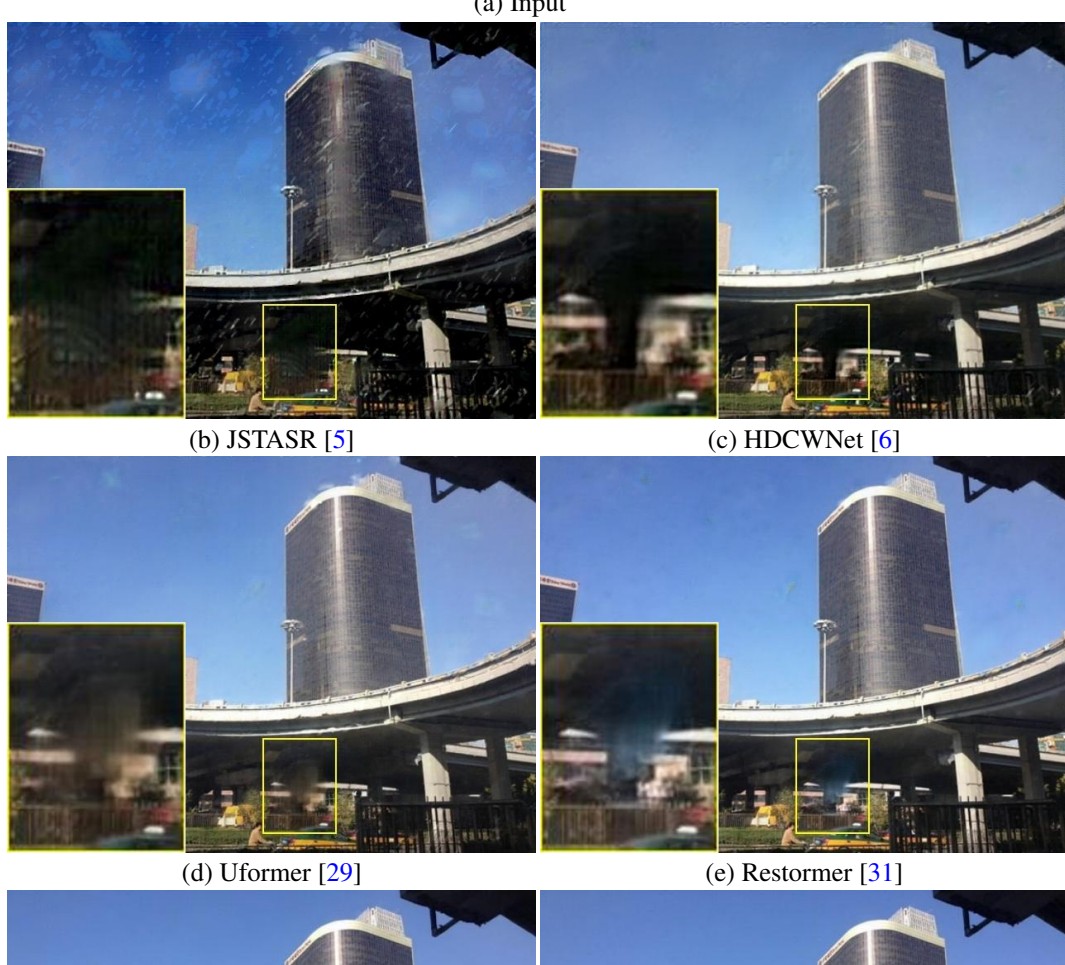

(b) JSTASR [5]

(c) HDCWNet [6]

(d) Uformer [29]

(e) Restormer [31]

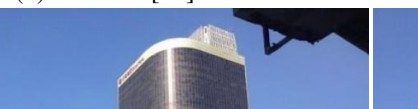

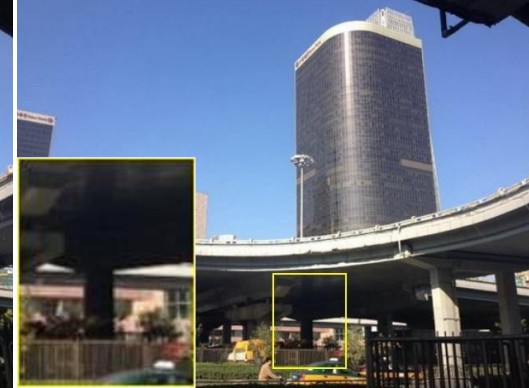

(f) PromptRestorer

(g) GT

Figure 25: **Image desnowing** example on CSD (2000) [6].

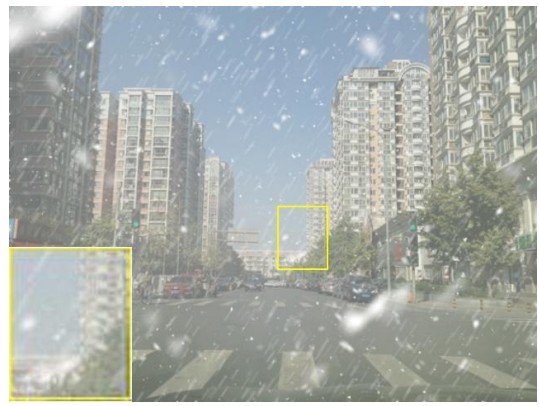

(a) Input

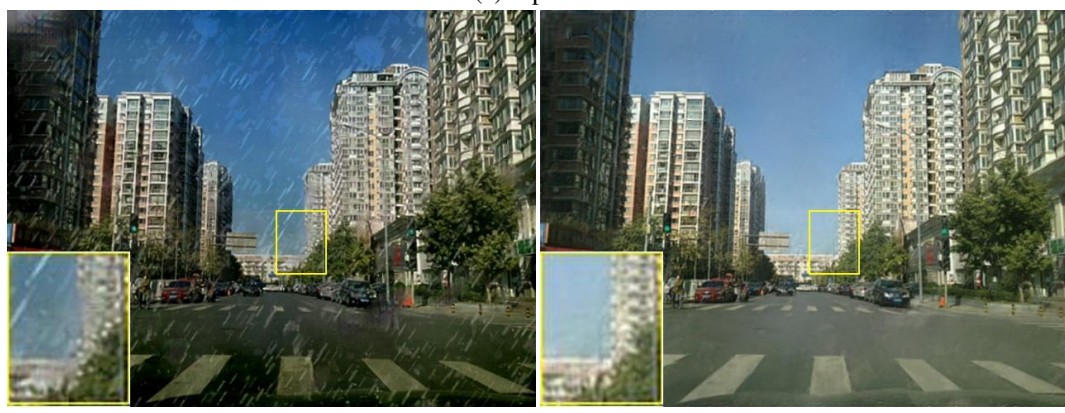

(b) JSTASR [5]                                           (c) HDCWNet [6]

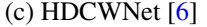
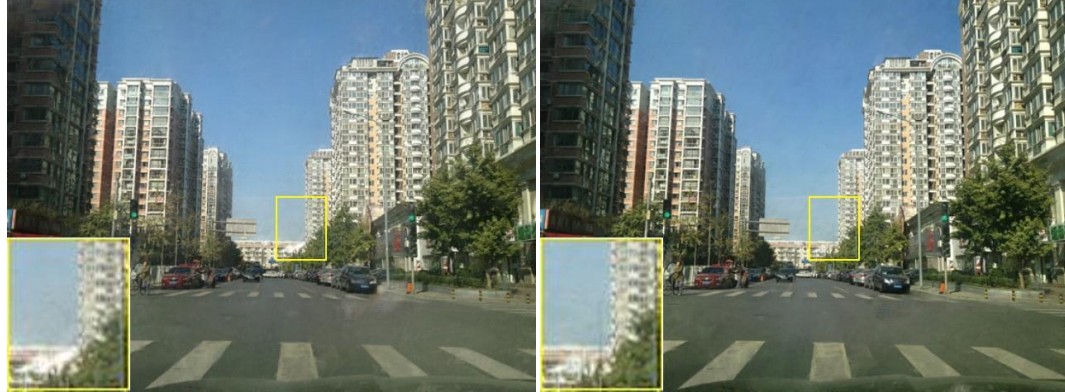

(d) Uformer [29]                                         (e) Restormer [31]

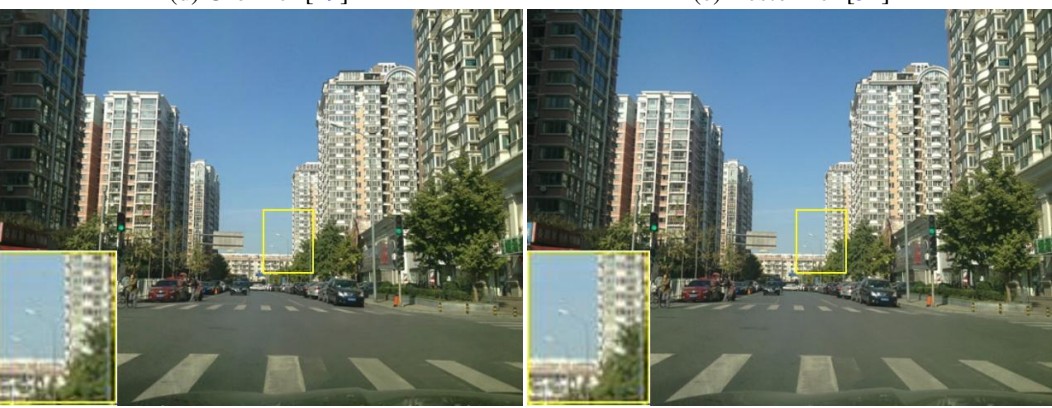

(f) PromptRestorer                                       (g) GT

Figure 26: **Image desnowing** example on CSD (2000) [6].

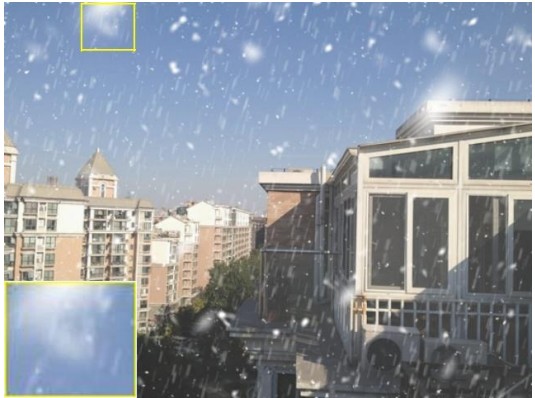

(a) Input

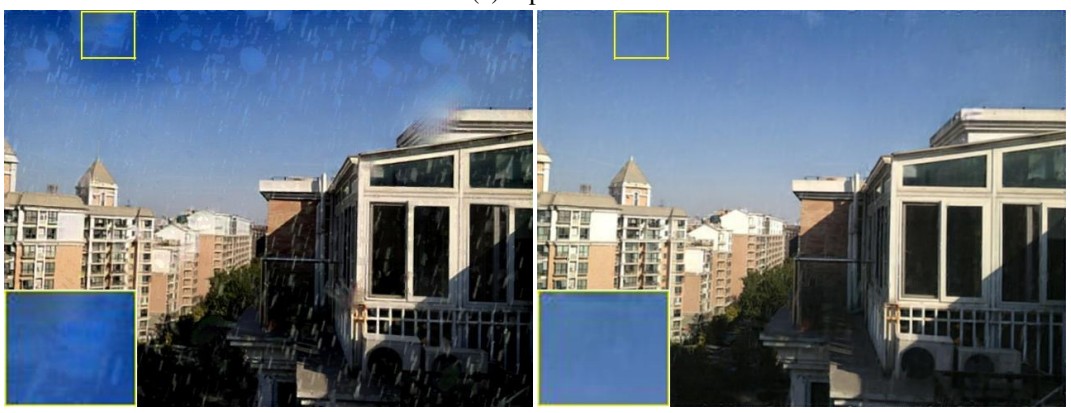

(b) JSTASR [5]  (c) HDCWNet [6]

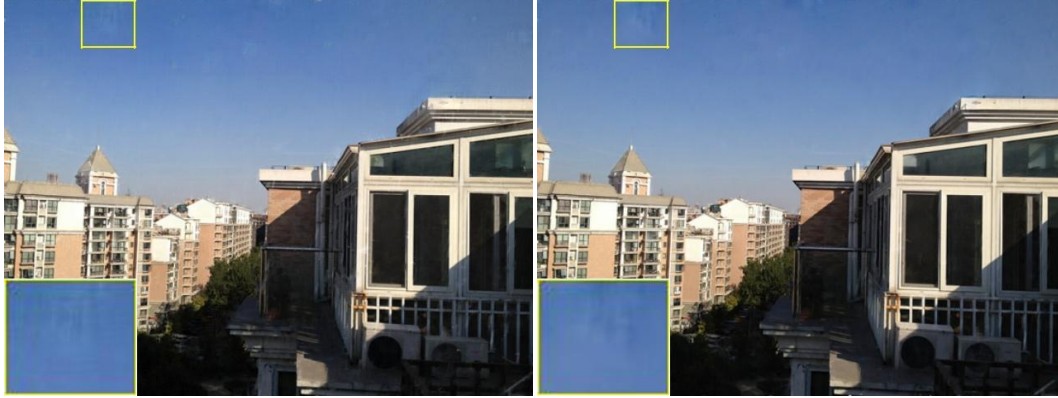

(d) Uformer [29]  (e) Restormer [31]

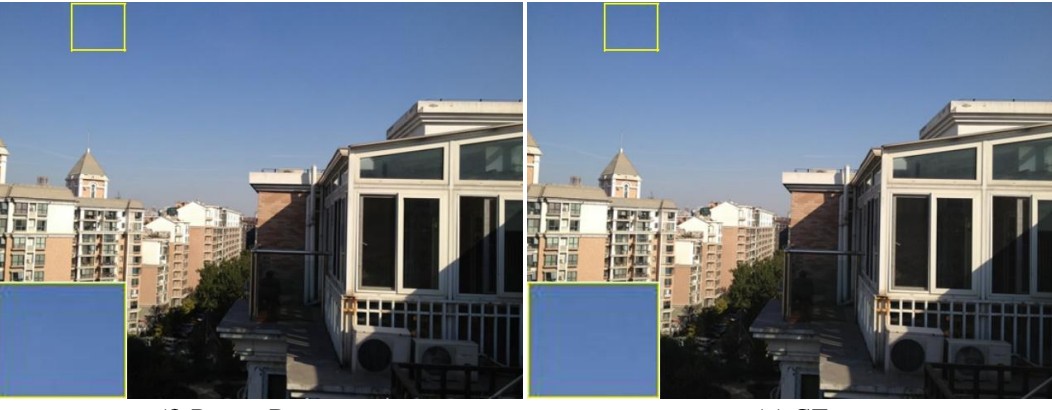

(f) PromptRestorer  (g) GT

Figure 27: **Image desnowing** example on CSD (2000) [6].

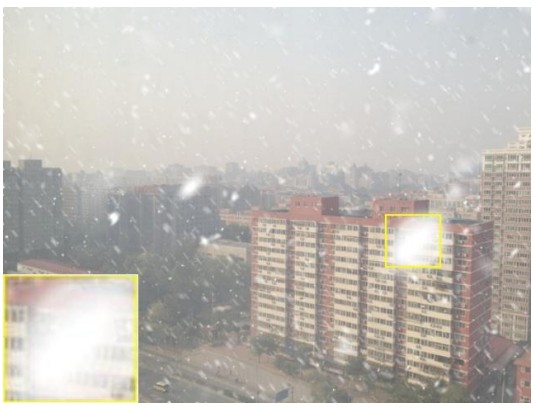

(a) Input

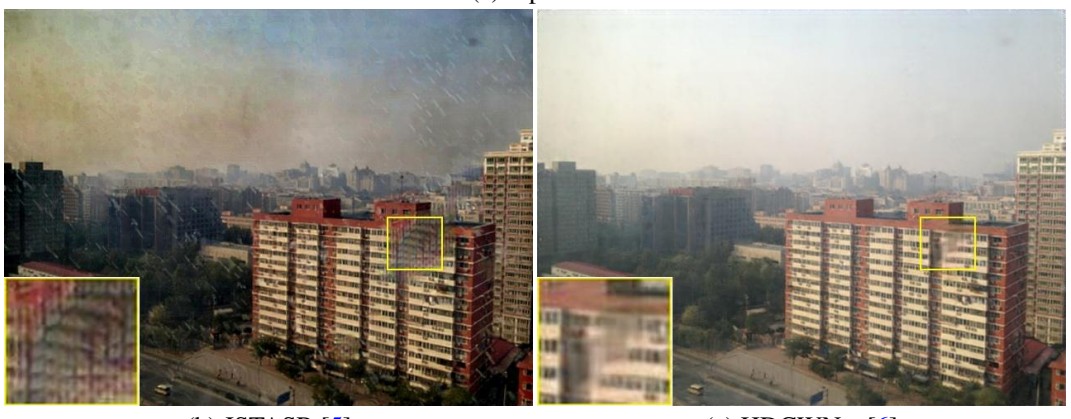

(b) JSTASR [5]  (c) HDCWNet [6]

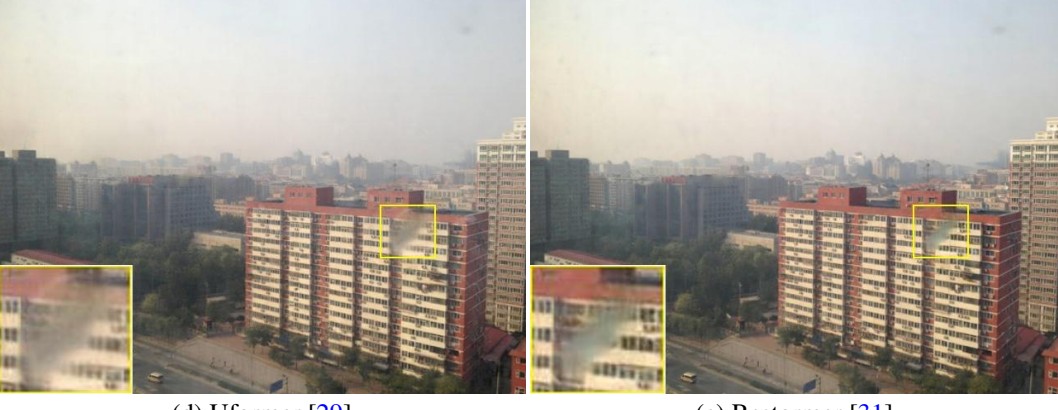

(d) Uformer [29]  (e) Restormer [31]

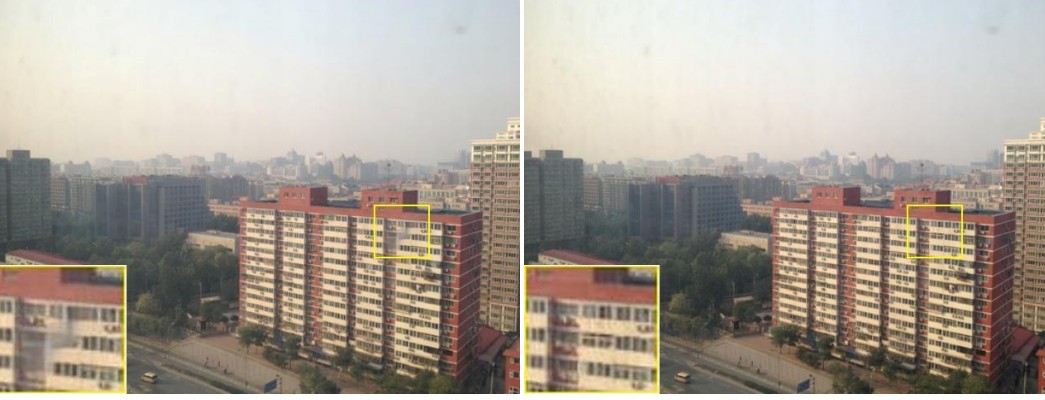

(f) PromptRestorer  (g) GT

Figure 28: **Image desnowing** example on CSD (2000) [6].

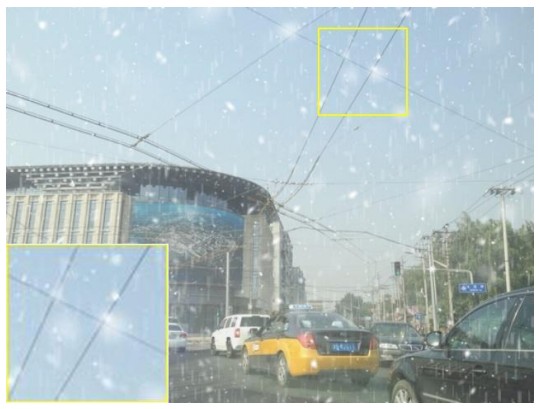

(a) Input

(b) JSTASR [5]

(c) HDCWNet [6]

(d) Uformer [29]

(e) Restormer [31]

(f) PromptRestorer

(g) GT

Figure 29: **Image desnowing** example on CSD (2000) [6].

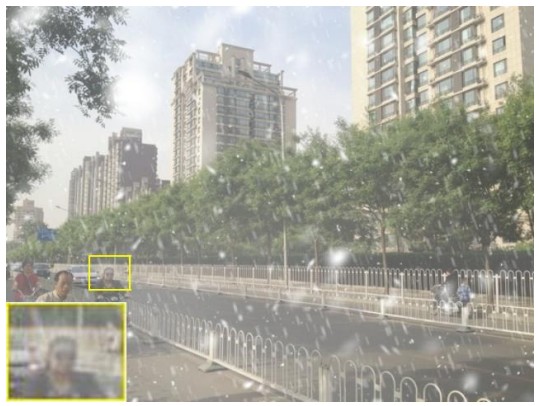

(a) Input

(b) JSTASR [5]

(c) HDCWNet [6]

(d) Uformer [29]

(e) Restormer [31]

(f) PromptRestorer

(g) GT

Figure 30: **Image desnowing** example on CSD (2000) [6].

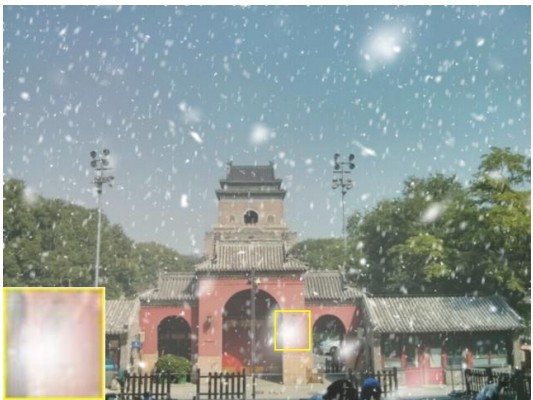

(a) Input

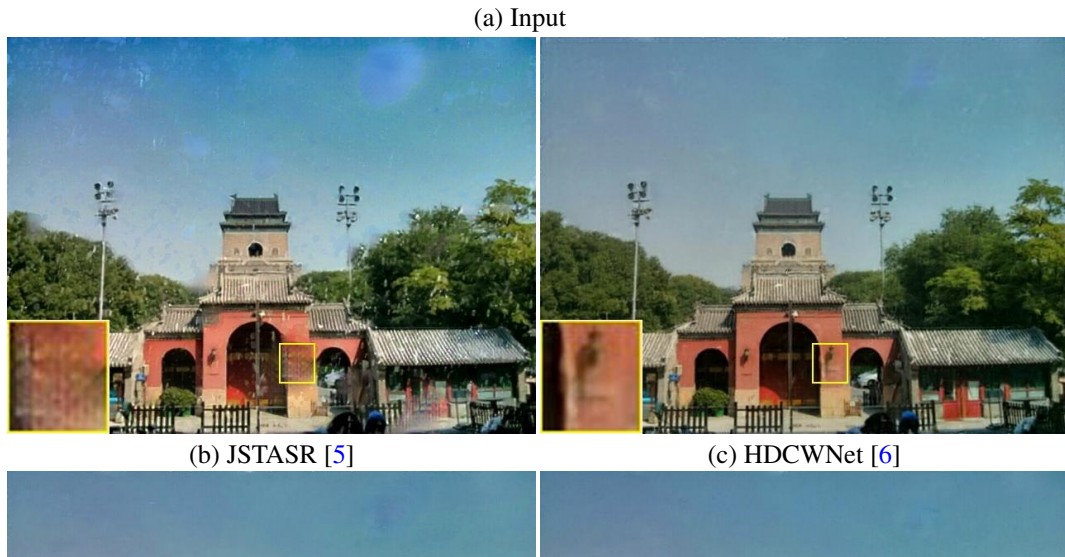

(b) JSTASR [5]                    (c) HDCWNet [6]

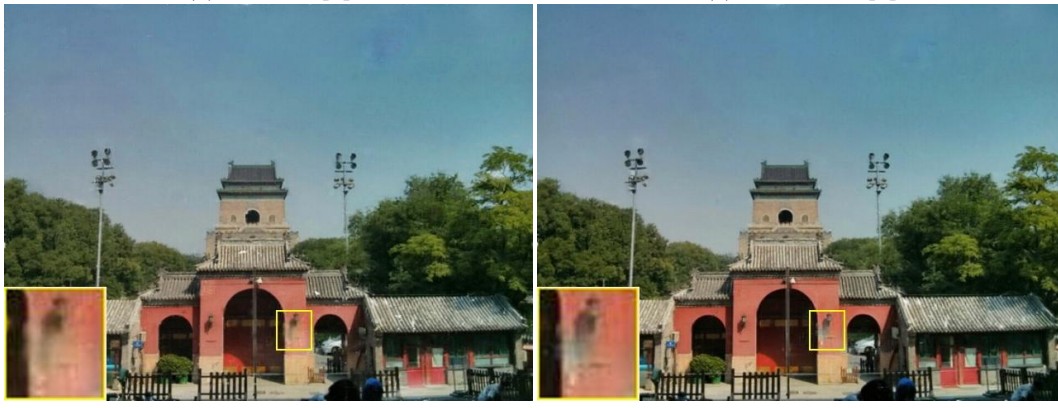

(d) Uformer [29]                    (e) Restormer [31]

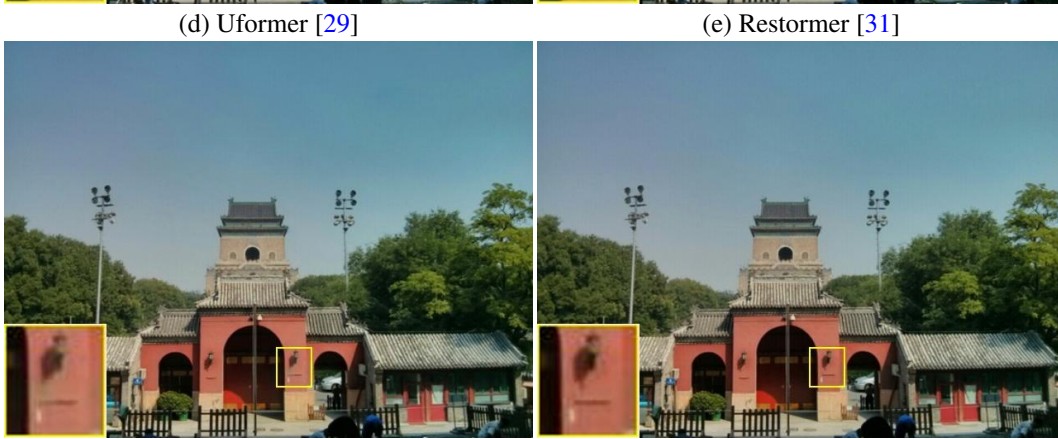

(f) PromptRestorer                    (g) GT

Figure 31: **Image desnowing** example on CSD (2000) [6].

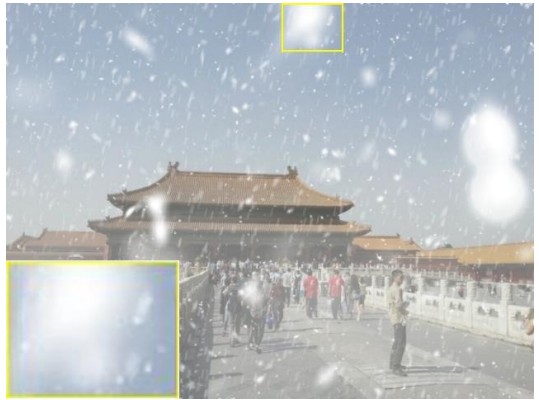

(a) Input

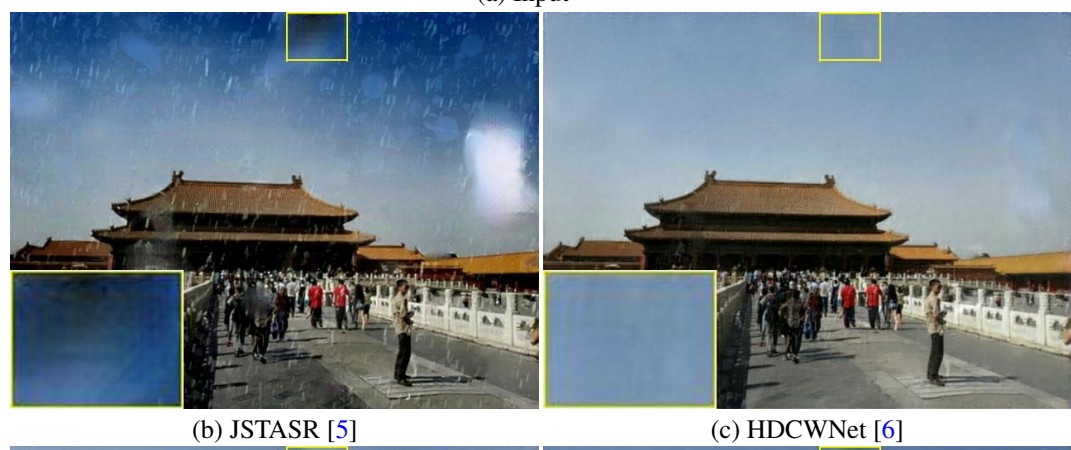

(b) JSTASR [5]                 (c) HDCWNet [6]

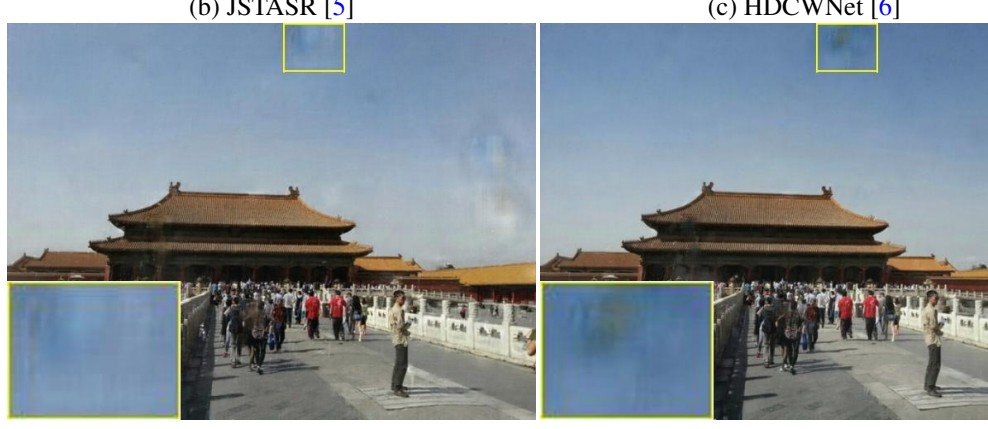

(d) Uformer [29]               (e) Restormer [31]

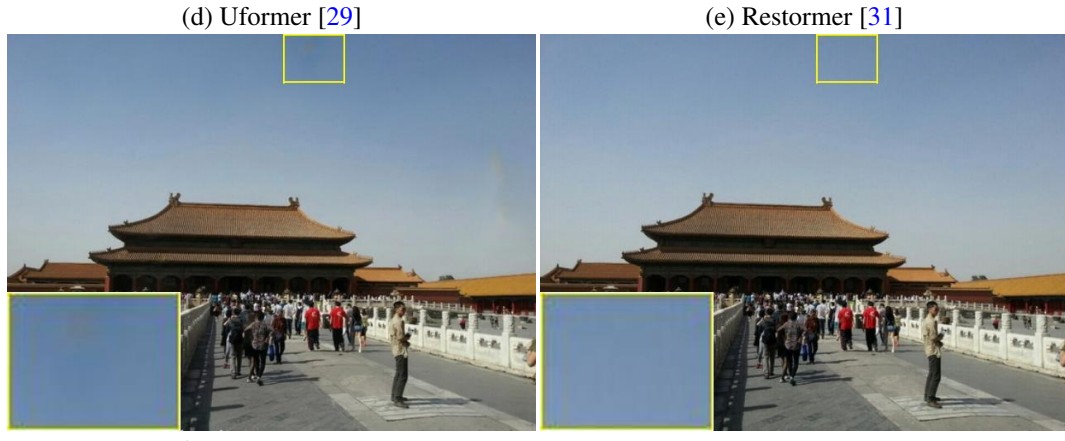

(f) PromptRestorer                 (g) GT

Figure 32: **Image desnowing** example on CSD (2000) [6].

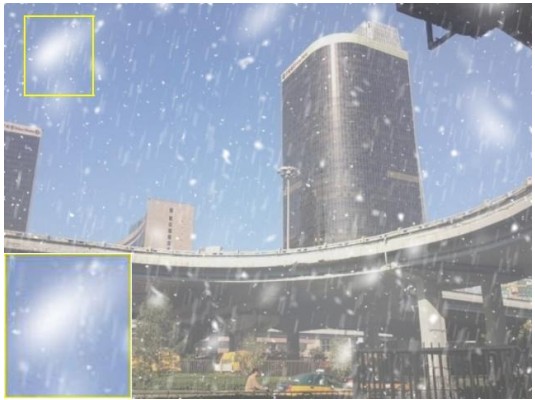

(a) Input

(b) JSTASR [5]

(c) HDCWNet [6]

(d) Uformer [29]

(e) Restormer [31]

(f) PromptRestorer

(g) GT

Figure 33: **Image desnowing** example on CSD (2000) [6].

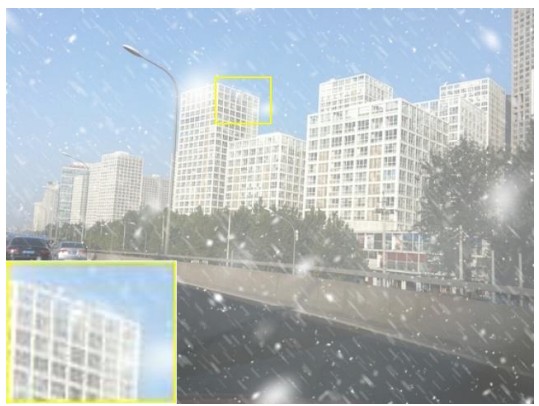

(a) Input

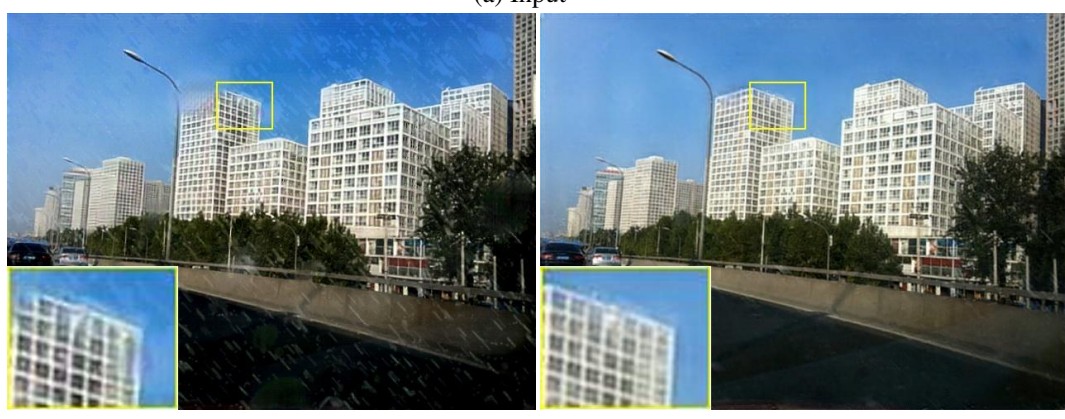

(b) JSTASR [5]                                        (c) HDCWNet [6]

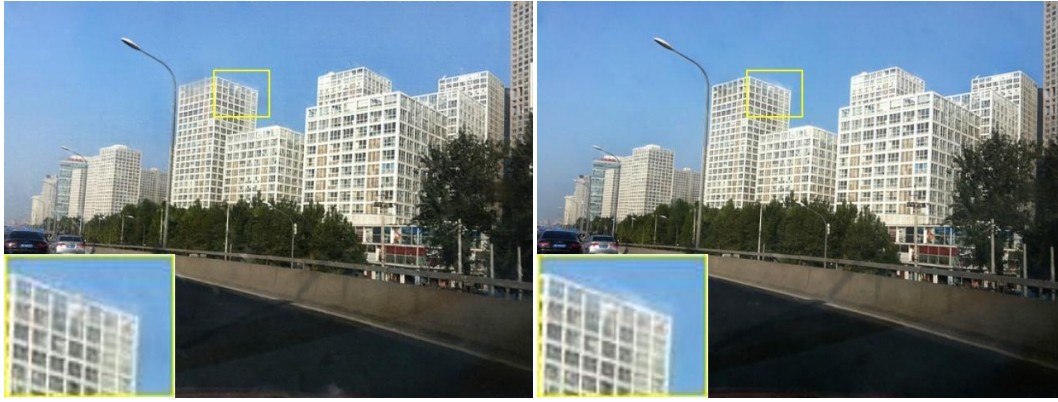

(d) Uformer [29]                                      (e) Restormer [31]

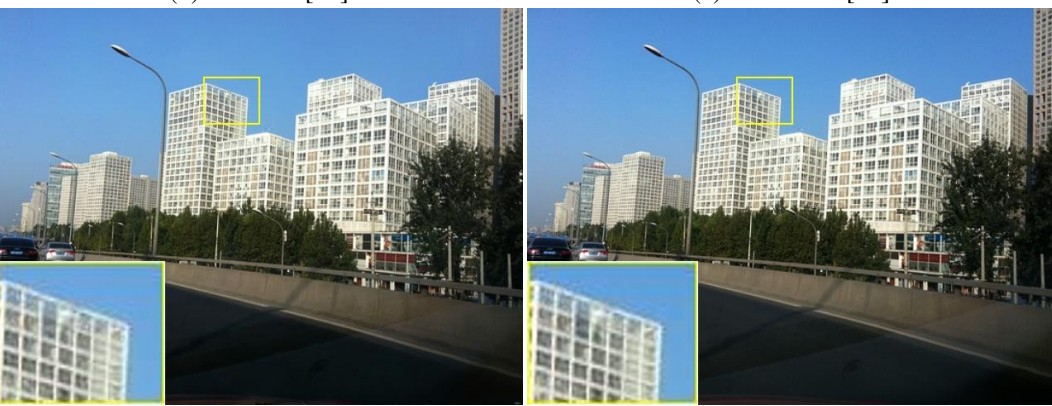

(f) PromptRestorer                                    (g) GT

Figure 34: **Image desnowing** example on CSD (2000) [6].

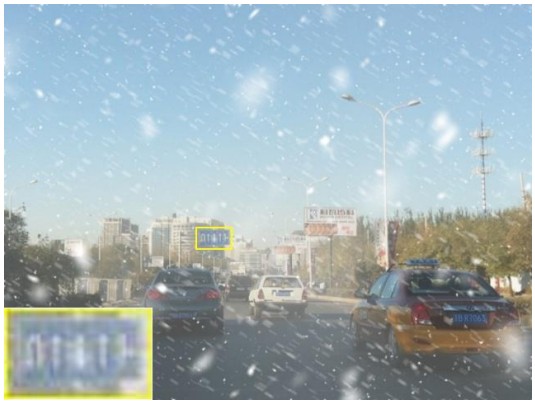

(a) Input

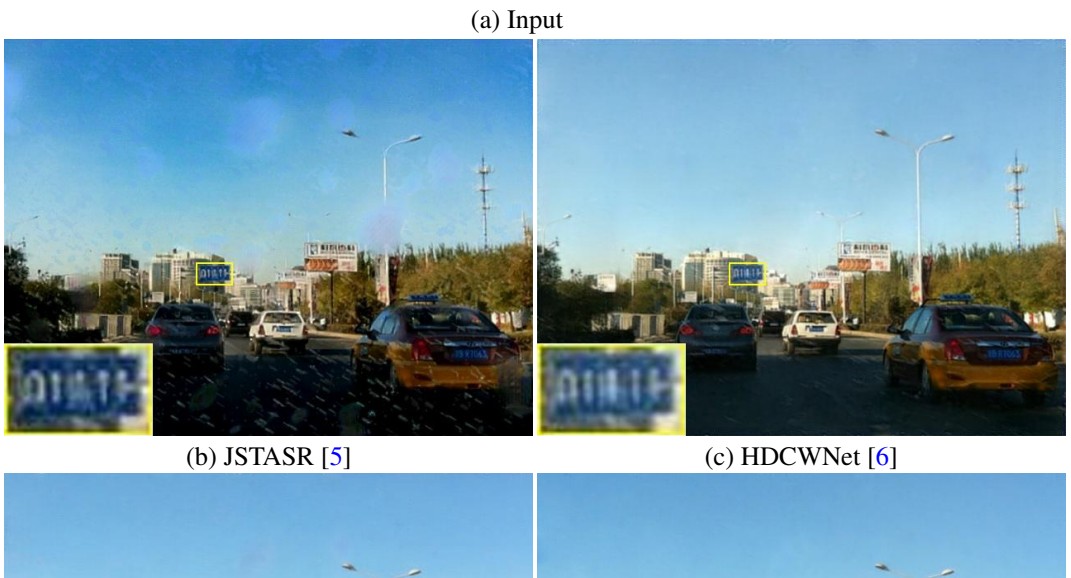

(b) JSTASR [5]        (c) HDCWNet [6]

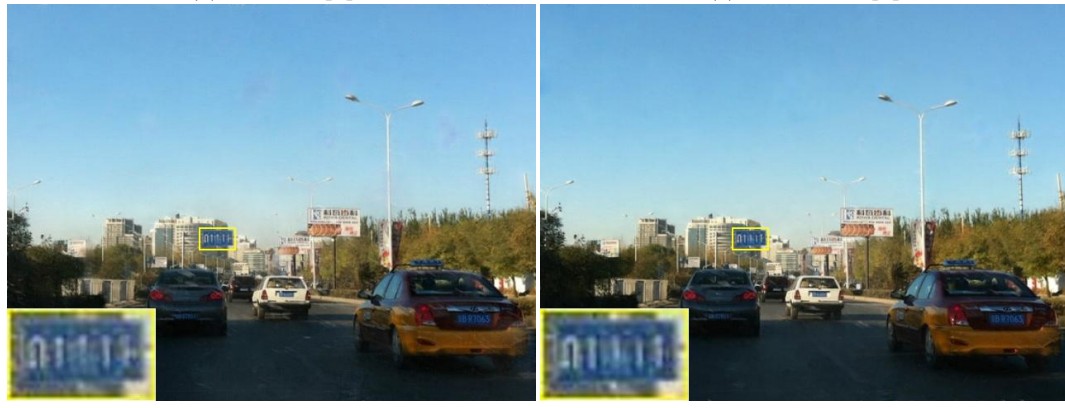

(d) Uformer [29]        (e) Restormer [31]

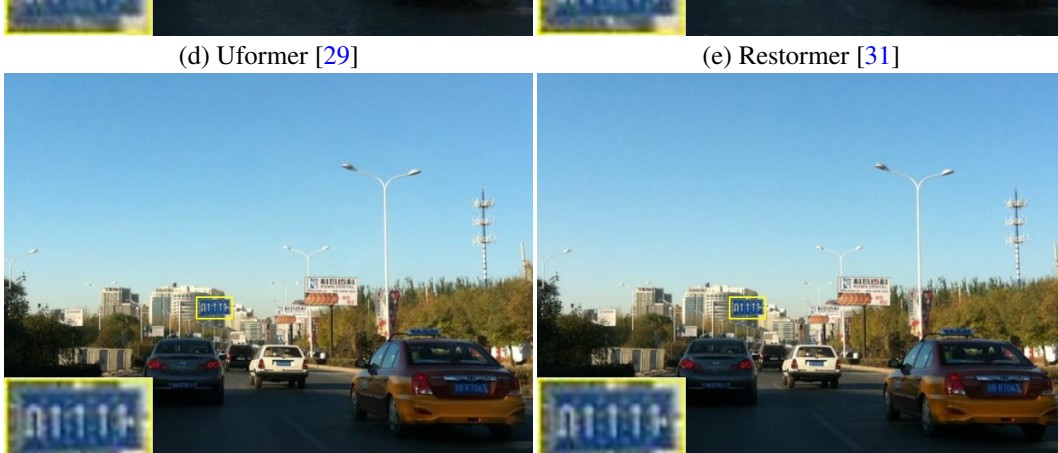

(f) PromptRestorer        (g) GT

Figure 35: **Image desnowing** example on CSD (2000) [6].

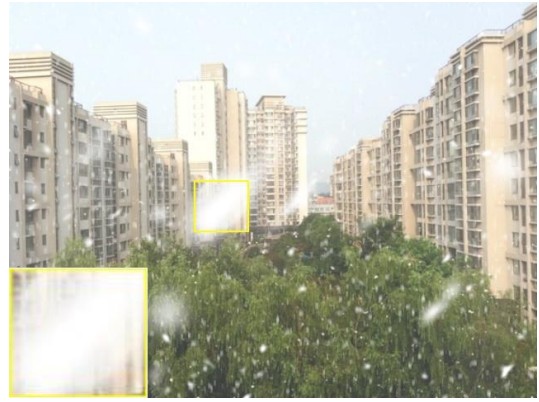

(a) Input

(b) JSTASR [5]

(c) HDCWNet [6]

(d) Uformer [29]

(e) Restormer [31]

(f) PromptRestorer

(g) GT

Figure 36: **Image desnowing** example on CSD (2000) [6].

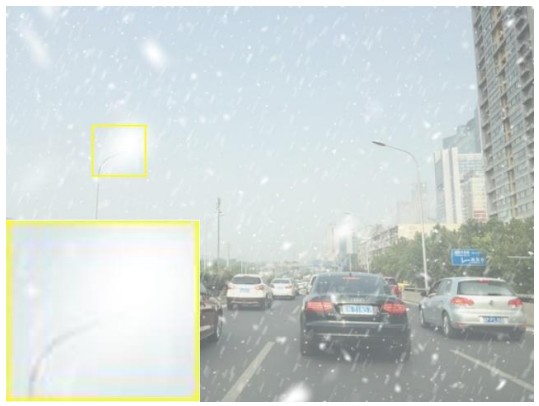

(a) Input

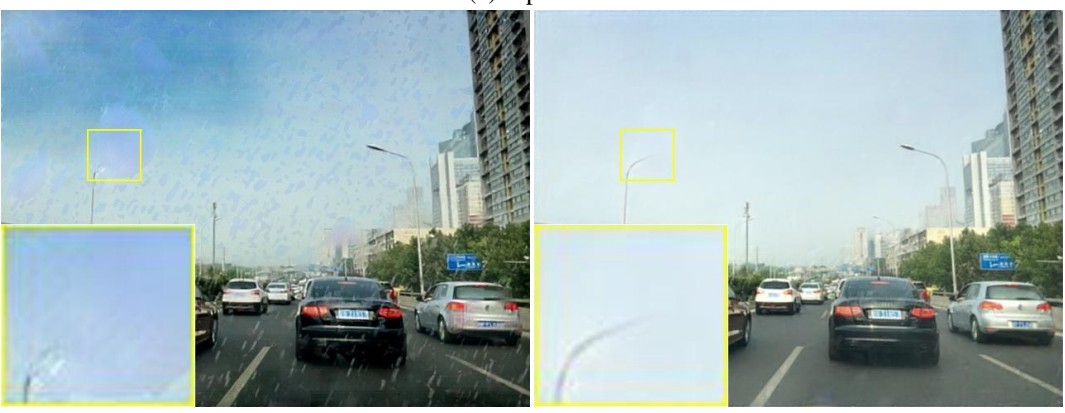

(b) JSTASR [5]              (c) HDCWNet [6]

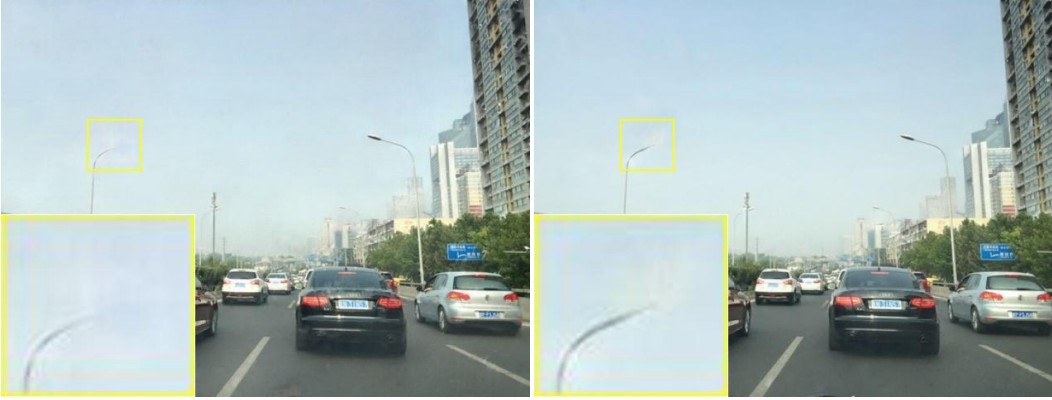

(d) Uformer [29]             (e) Restormer [31]

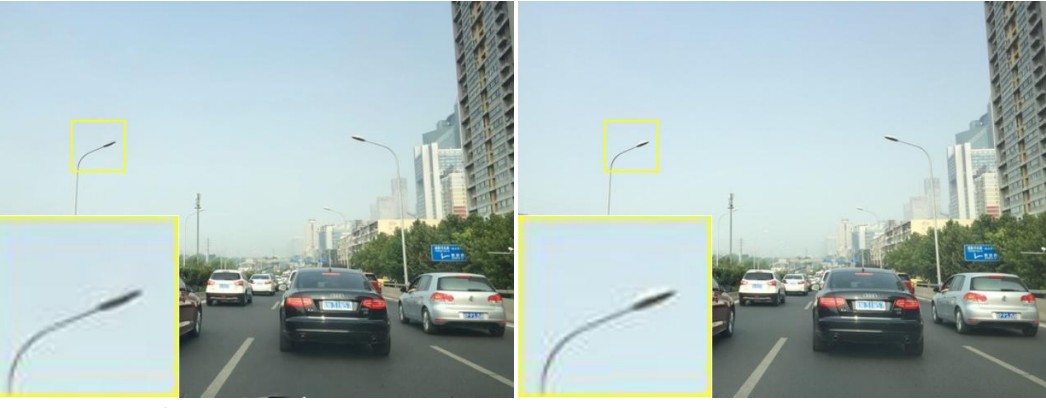

(f) PromptRestorer            (g) GT

Figure 37: **Image desnowing** example on CSD (2000) [6].

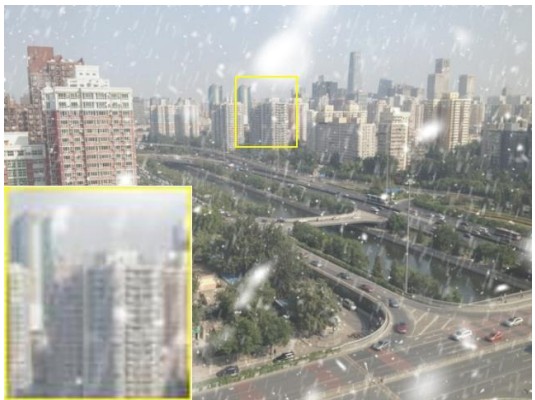

(a) Input

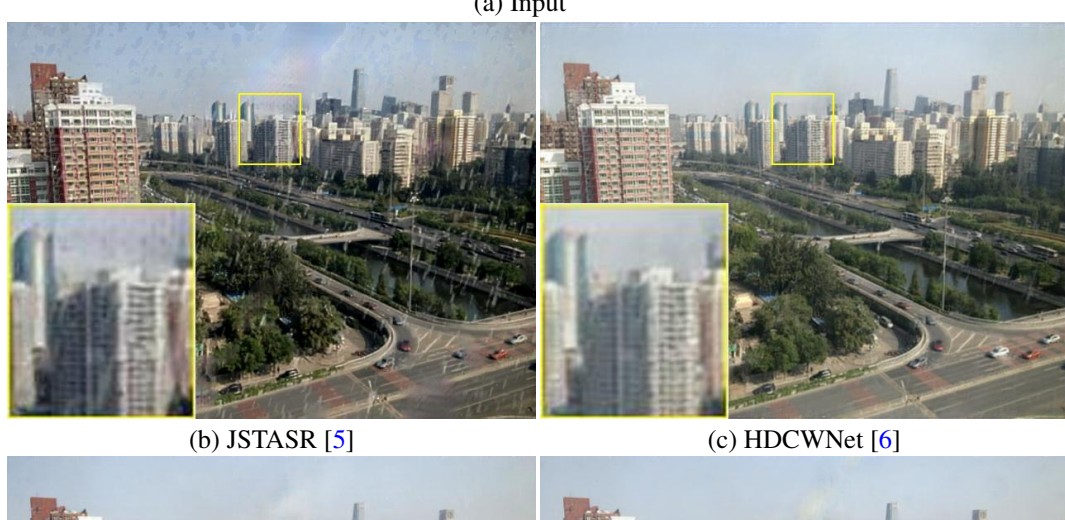

(b) JSTASR [5]             (c) HDCWNet [6]

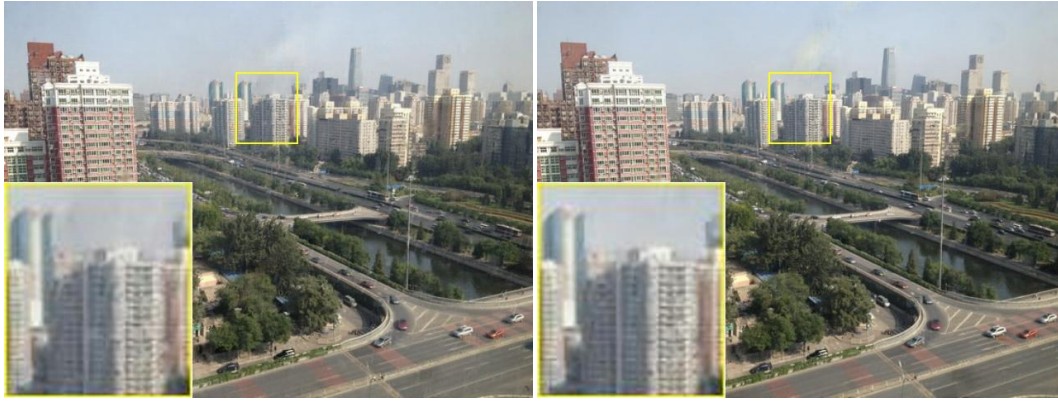

(d) Uformer [29]            (e) Restormer [31]

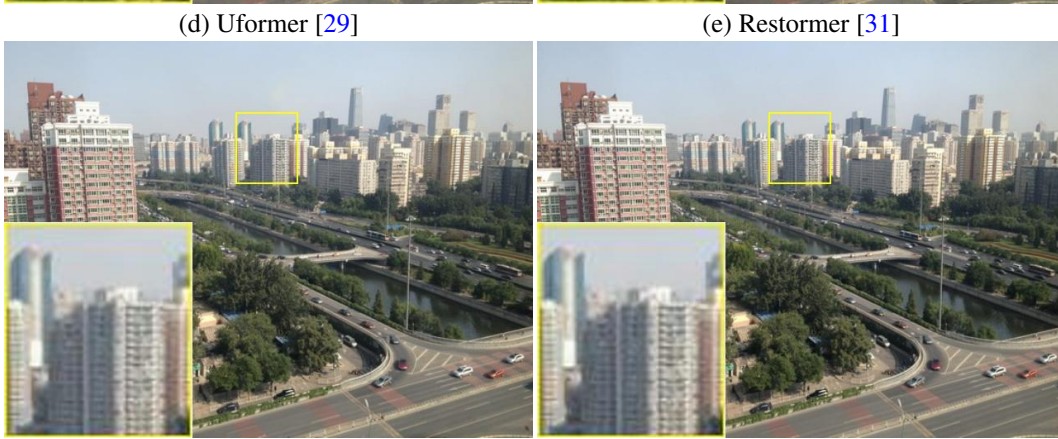

(f) PromptRestorer           (g) GT

Figure 38: **Image desnowing** example on CSD (2000) [6].