# OpenReview forum: "PromptRestorer: A Prompting Image Restoration Method with Degradation Perception"
_NeurIPS.cc/2023/Conference — NeurIPS 2023 poster_

### Official Review · Reviewer_zfG4 · 2023-06-27

**Soundness:** 3 good
**Presentation:** 2 fair
**Contribution:** 3 good
**Rating:** 6
**Confidence:** 4

**Summary:**

This paper introduces the PromptRestorer, a method for image restoration that effectively utilizes raw degradation features to guide the restoration process. PromptRestorer consists of two branches: a restoration branch and a prompting branch. The restoration branch restores the images, while the prompting branch perceives degradation priors and provides reliable content to guide the restoration process. Experimental results demonstrate that PromptRestorer achieves state-of-the-art performance on four image restoration tasks: deraining, deblurring, dehazing, and desnowing.


**Strengths:**

- The idea of utilizing intermediate layer features of pre-trained models to enhance the effectiveness of image restoration is innovative.
- The experimental validation tasks and comparison methods employed in this study are comprehensive and diverse.
- The comparative figure (Figure 1) between the proposed method and previous approaches are presented in a clear manner, highlighting the distinctive aspects of the framework.


**Weaknesses:**

- The motivation behind the introduction of the prompting degradation perception modulator is not clearly articulated in the paper. The authors fail to thoroughly analyze the existing deficiencies of previous methods and instead claim that the incorporation of prompting can lead to more accurate degradation priors or more reliable perceived content learned from the degradation priors. However, these descriptions lack quantifiable metrics and robust support from relevant literature. Despite the inclusion of visualizations depicting feature maps in the experimental section, there remains an ambiguity regarding the definition of better degradation priors.
- Similarly, I am confused regarding the motivation behind the introduction of local prompting attention and global prompting attention. It remains unclear to what extent these components address existing shortcomings in the framework. Furthermore, the results of the ablation experiments indicate relatively minor improvements resulting from the incorporation of these components.


**Questions:**

- What is a more accurate degradation priors? Can we measure it?
- What are the motivations behind the introduction of key modules and the issues they address in previous methods?
- It appears that the network parameter count and FLOPS of this method may increase significantly compared to those of Case 1 and Case 2 methods. Is there more detailed comparative data available in this regard?


**Limitations:**

- The performance improvement of this method appears to be relatively modest compared to some previous approaches, and it remains unclear what the magnitude of the associated numerical variances is.

---

> ### Author Rebuttal · Authors · 2023-08-09
>
> Q: What is a more accurate degradation prior? Can we measure it?
>
> A: The most accurate degradation priors should be the degraded image itself. In feature space, better degradation should have the property that can better represent the features of the degraded image. We note that VQGAN can reconstruct the input images and thus can accurately represent the features of the input images [1]. Hence, we use the pre-trained VQGAN to accurately extract degradation features as the degradation priors. Fig. 8 (GT features and Case 3) clearly shows that pre-trained VQGAN can better preserve the features of the input images.
>
> [1] Gu et al. VQFR: Blind Face Restoration with Vector-Quantized Dictionary and Parallel Decoder. ECCV2022.
>
> Q: What are the motivations behind the introduction of key modules and the issues they address in previous methods? I am confused regarding the motivation of local prompting attention and global prompting attention.
>
> A: Our method is motivated by that existing image restoration models usually appear to degradation vanishing, as shown in Case 1-2 of Fig. 8, as the model parameters are optimized toward producing clean content. However, the degradation content is critical for image restoration, which implies important cues for better image restoration. To solve this problem, we propose to use raw degradation features to overcome degradation vanishing so that helps the networks constantly perceive the degradation priors to facilitate better recovery. To better perceive the degradation, we propose G2P and L2P to respectively perceive the degradation from global and local perspectives, enabling to generate more useful content to guide the restoration branch. To control the propagation of the perceived content, we propose the GDP, enabling the restoration network to adaptively learn more useful features for better restoration.
>
> Our global perception attention consists of Q-InAtt and KV-InAtt. The Q-InAtt considers re-forming the query vector induced by degradation features to build a representative query to perform attention, while the KV-InAtt re-considers key and value vectors induced by other degradation counterparts to search for more similar content with the restoration query. The local perception modulator contains Deg-InBan and Res-InBan. The former is achieved by exploiting the degradation features to induce spatially useful content from restoration content to guide restoration gating fusion, while the latter utilizes the deep restoration features to induce more useful features from another degradation counterpart to form the degradation gating.
>
> Previous commonly-used fusion methods SFT usually only consider spatial feature transformation with a single direction transformation, i.e., from conditions to restoration fusion at pixel-level, while neglecting the global perspectives. We consider both local and global transformation by re-forming attention and local modulation with bidirectional interaction, i.e., from restoration to degradation, and from degradation to restoration, to guide restoration.
>
> Q: It appears that the network parameters count and FLOPs of this method may increase significantly compared to those of Case 1 and Case 2 methods. Is there more detailed comparative data available in this regard?
>
> A: The learnable parameters of our method are fewer than those of the model with learnable conditional modulation (Case 2), as shown in Tab. 5. This is because we use a pre-trained model to extract raw degradation information which keeps frozen when training the restoration models. However, our method still performs better than the model with learnable conditional modulation.
>
> For Case 1, since it does not have the conditional branch, our model (Case 3) and model with learnable conditional modulation (Case 2) naturally contain more parameters and FLOPs.
>
> Q: The improvement of this method appears to be relatively modest compared to some previous approaches.
>
> A: Our method consistently achieves better performance on $4$ image restoration tasks on various benchmarks, which demonstrates the effectiveness of our method.
>
> Q: The ablation indicates relatively minor improvements resulting from the incorporation of these components.
>
> A: Our experiments demonstrate that each component is effective for restoration. Note that our method with fewer parameters performs better than the model with learnable conditional modulation, which validates our key contribution that using raw degradation content is more effective than learnable conditions.

---

> > ### Comment · Reviewer_zfG4 · 2023-08-12
> > **Reviewer reply**
> >
> > After considering the response provided in the rebuttal and taking into account the feedback from other reviewers, I recognize the novelty of this paper. The authors highlight that utilizing raw degradation is more impactful compared to the current approach of learnable conditional modulation, offering a different perspective in this research area. So, I have raised my rating.

---

### Official Review · Reviewer_Hjzg · 2023-06-29

**Soundness:** 4 excellent
**Presentation:** 4 excellent
**Contribution:** 4 excellent
**Rating:** 7
**Confidence:** 5

**Summary:**

In this paper, the authors find that the degradation vanishes in the learning process of the image restoration problem. Which further hinders the model's capacity. To solve this problem, this paper proposes the PromptRestorer, a prompting image restoration method by exploring the raw degradation features extracted by a pre-trained model from the given degraded observations to guide the restoration process. The authors demonstrate that the proposed guidance manner is better than both the encoder-decoder model without any guidance and the model with learnable guidance. The experiments on image deraining, deblurring, dehazing, and desnowing demonstrate the superiority of the proposed algorithm.

**Strengths:**

1, The method is simple and effective and is easy to follow. The writing is professional and clear to understand.
2, The proposed Prompting Degradation Perception Modulator including global prompting perceptor and local prompting perceptor is novel and effective.
3, The proposed Gated Degradation Perception Propagation is simple and meaningful.
4, This paper clarifies a new perspective that the raw degradation features are better than the learnable features which are usually utilized in former methods.
5, The extensive experiments on various image restoration tasks demonstrate the generalization of the proposed algorithm.

**Weaknesses:**

1, Although this paper clarifies many strengths of the method, what’s the limitation? The author should present it in this paper.
2, The authors should give more detailed explanation of the proposed global prompting perceptor and local prompting perceptor to facilitate better understanding.


**Questions:**

See Weaknesses

**Limitations:**

The authors are suggested to discuss the limitations of this paper.

---

> ### Author Rebuttal · Authors · 2023-08-09
>
> Q: Although this paper clarifies many strengths of the method, what’s the limitation? The author should present it in this paper.
>
> A: Since our method needs to utilize a pre-trained model to extract degraded priors, it consumes more parameters and FLOPs compared with the model without any conditional modulation (i.e., Case 1), as illustrated in Tab. 5. However, we demonstrate that using raw features to guide restoration produces better performance than previous widely-used learnable conditional modulation (Case 2) while consuming fewer learnable parameters.
>
>
> Q: The authors should give a more detailed explanation of the proposed global prompting perceptor and local prompting perceptor to facilitate better understanding.
>
> A: The G2P fully exploits the self-attention mechanism to form the global prompting attention induced by the degraded features. The G2P contains the global perception attention followed by an improved ConvNeXt. Our global perception attention consists of 1) Query-Induced Attention (Q-InAtt) and 2) Key-Value-Induced Attention (KV-InAtt). The Q-InAtt considers re-forming the query vector induced by degradation features to build a representative query to perform attention, while the KV-InAtt re-considers key and value vectors induced by other degradation counterparts to search for more similar content with the restoration query.
>
> The L2P adequately considers the pixel-level degradation perception, enabling it to better perceive degradation from spatially neighboring pixel positions. The L2P consists of a local perception modulator followed by a separable depth-level convolution. The local perception modulator contains two core components: 1) Degradation-Induced Band (Deg-InBan) and 2) Restoration-Induced Band (Res-InBan). The former is achieved by exploiting the degradation features to induce spatially useful content from restoration content to guide restoration gating fusion, while the latter utilizes the deep restoration features to induce more useful features from another degradation counterpart to form the degradation gating.

---

> > ### Comment · Reviewer_Hjzg · 2023-08-16
> > **Reviewer reply**
> >
> > Thanks for the efforts in the response. All my concerns have been addressed.

---

### Official Review · Reviewer_T7gZ · 2023-06-30

**Soundness:** 3 good
**Presentation:** 4 excellent
**Contribution:** 4 excellent
**Rating:** 8
**Confidence:** 5

**Summary:**

This paper proposes the PromptRestorer by taking advantage of the prompting learning for image restoration. The author considers raw degradation features into restoration which consistently retains the degradation priors to facilitate better restoration. To better perceive degradation, the authors propose Global Prompting Perceptor and Local Prompting Perceptor.
To control the propagation of the perceived features, the authors propose gated degradation perception propagation, enabling the model to adaptively learn more useful features for better image restoration.


**Strengths:**

1, Good writing skills. The paper shows excellent presentation and clearly presents former disadvantages and better solves them by introducing the prompting strategy.
2, Clear motivation. To solve the problem of degradation vanishing in image restoration (Fig. 1 and 8), the authors introduce a Prompting Degradation Perception by extracting the raw degradation features to directly guide the image restoration process. The experiments (Tab. 5) show that the proposed learning strategy is better than the former methods.
3, Interesting module. The proposed Prompting Degradation Perception Modulator is interesting and Gated Degradation Perception Propagation is simple.


**Weaknesses:**

1, The authors can give a clearer understanding of the degradation priors guidance. Why does it work better than the model with a learnable conditional branch?
2, Although the Prompting Degradation Perception Modulator including the Global Prompting Perceptor and Local Prompting Perceptor is interesting, it would be better if the authors can give a more direct explanation of them as the two modules seem to look the key contributions of this work.
3, The paper exists some typos. For example, in line 89, knew->known.


**Questions:**

Please refer to the questions in Weaknesses

**Limitations:**

The authors do not present the limitations in the paper, which are suggested to add them.

---

> ### Author Rebuttal · Authors · 2023-08-09
>
> Q: The authors can give a clearer understanding of the degradation priors guidance. Why does it work better than the model with a learnable conditional branch?
>
> A: The learnable nature of the conditional network in these models does not effectively provide degradation information for the restoration network since the parameters in the learnable conditional branch are optimized in the learning process so that the features become gradually clear, which cannot effectively provide the restoration networks with degraded priors well.
>
> In contrast, we directly exploit the raw degraded features extracted by a pre-trained model from the degraded inputs to generate more reliable prompting content to guide image restoration. Raw degraded features preserve accurately degraded information, which can consistently prompt the restoration network with accurate degraded priors, enabling the restoration network to perceive the degradation for better restoration.
>
> Q: Although the Prompting Degradation Perception Modulator including the Global Prompting Perceptor (G2P) and Local Prompting Perceptor (L2P) is interesting, it would be better if the authors can give a more direct explanation of them as the two modules seem to look the key contributions of this work.
>
> A: The G2P fully exploits the self-attention mechanism to form the global prompting attention induced by the degraded features. The G2P contains the global perception attention followed by an improved ConvNeXt. Our global perception attention consists of 1) Query-Induced Attention (Q-InAtt) and 2) Key-Value-Induced Attention (KV-InAtt). The Q-InAtt considers re-forming the query vector induced by degradation features to build a representative query to perform attention, while the KV-InAtt re-considers key and value vectors induced by other degradation counterparts to search for more similar content with the restoration query.
>
>
> The L2P adequately considers the pixel-level degradation perception, enabling it to better perceive degradation from spatially neighboring pixel positions. The L2P consists of a local perception modulator followed by a separable depth-level convolution. The local perception modulator contains two core components: 1) Degradation-Induced Band (Deg-InBan) and 2) Restoration-Induced Band (Res-InBan). The former is achieved by exploiting the degradation features to induce spatially useful content from restoration content to guide restoration gating fusion, while the latter utilizes the deep restoration features to induce more useful features from another degradation counterpart to form the degradation gating.
>
> Q: The paper exists some typos. For example, in line 89.
>
> A: Thanks for your careful reading. We will fix them in the revised paper.

---

> ### Comment · Reviewer_T7gZ · 2023-08-18
> **Good paper, suggest to accept.**
>
> After thoroughly reviewing the authors' feedback, I find that most of my questions have been adequately addressed. Thank you for your efforts.

---

### Official Review · Reviewer_24Ac · 2023-07-05

**Soundness:** 4 excellent
**Presentation:** 3 good
**Contribution:** 4 excellent
**Rating:** 7
**Confidence:** 5

**Summary:**

This paper proposes a prompting image restoration method with degradation perception. The authors show that raw degradation features can effectively guide deep restoration models, providing accurate degradation priors to facilitate better restoration. To perceive the degradation, the authors propose prompting degradation perception modulator. To control the propagation of the perceived content for restoration branch, the authors propose gated degradation perception propagation.



**Strengths:**

+ This paper is professionally wrote and presents a better organization.
+ The proposed prompting image restoration method is interesting, which clearly demonstrate that the raw degradation features are better guidance than the learnable parameters used in previous methods.
+ The proposed prompting degradation perception modulator is novel. It has potential applications to latter features fusion manner.
+ The gated degradation perception propagation seems to be useful.


**Weaknesses:**

1, The author claim that the pre-trained model is the same with the learnable conditional network. Hence, the FLOPs in Tab. 5 should be same.

2, The authors would better to explain why the raw degradation features are better than features from learnable parameters network since more learnable parameters should have better results within the similar network framework.
3, SFT is a commonly used feature fusion module, is the proposed PromptDPM than that? What is the result if replacing the PromptDPM with SFT?


**Questions:**

see weakness

**Limitations:**

see weakness

---

> ### Author Rebuttal · Authors · 2023-08-09
>
> Q: The author claims that the pre-trained model is the same as the learnable conditional network. Hence, the FLOPs in Tab. 5 should be the same.
>
> A: Thanks for your reminder. The reported FLOPs of Case 3 in Tab. 5 are the restoration network. We will correct it in the revised paper.
>
> Q: The authors would better explain why the raw degradation features are better than features from the learnable parameters network since more learnable parameters should have better results within a similar network framework.
>
> A: The learnable nature of the conditional network in these models does not effectively provide degradation information for the restoration network. As parameters are optimized in the learning process, features become gradually clear, which cannot effectively guide the restoration networks.
>
> In contrast, we directly exploit the raw degraded features extracted by a pre-trained model from the degraded inputs to generate more reliable prompting content to guide image restoration. Raw degraded features preserve accurately degraded information, which can consistently prompt the restoration network with accurate degraded priors, enabling the restoration network to perceive the degradation for better restoration. Tab. 5 clearly shows using the raw degraded features to guide restoration produces 0.369dB PSNR gains than the model with a learnable conditional branch.
>
> Q: SFT is a commonly used feature fusion module, is the proposed PromptDPM better than that? What is the result if we replace the PromptDPM with SFT?
>
> A: We compare the SFT with our PromptDPM by replacing the PromptDPM with SFT in our model. The PSNR results are summarised as follows:
>
> SFT: 30.699; Ours: 31.015.
>
> Our PromptDPM improves by 0.316dB PSNR compared with the commonly used SFT module, which further demonstrates the effectiveness of the proposed PromptDPM.

---

> > ### Comment · Reviewer_24Ac · 2023-08-17
> >
> > After reading the rebuttal, I would like to keep my rating.

---

### Official Review · Reviewer_NMEt · 2023-07-06

**Soundness:** 2 fair
**Presentation:** 3 good
**Contribution:** 2 fair
**Rating:** 4
**Confidence:** 5

**Summary:**

The paper proposes PromptRestorer for image restoration, addressing the issue of degradation vanishing in existing methods. Unlike previous approaches that do not explicitly consider degradation information or fail to effectively model degradation priors, PromptRestorer leverages the raw degraded features extracted from pre-trained models to generate reliable prompting content. The contributions of the work include PromptRestorer, the first approach to leverage prompting learning for general image restoration by considering raw degradation features, overcoming degradation vanishing while consistently retaining degradation priors. Additionally, the proposed prompting degradation perception modulator and gated degradation perception propagation enhance the restoration process by providing more reliable perceived content and controlling feature propagation. Extensive experiments show the superiority of the proposed method.

**Strengths:**

- The paper introduces a new approach called PromptRestorer for image restoration, which addresses the issue of degradation vanishing. The integral designs take advantage of prompting learning and explicitly consider degradation information, making it a unique and innovative contribution to the field.

- The proposed method effectively perceives and retains degradation priors by leveraging raw degraded features and designing a prompting degradation perception modulator, resulting in improved performance compared to existing methods.

- This comprehensive framework ensures that all aspects of restoration, including generating reliable prompting content and controlling feature propagation.

- The paper provides a detailed analysis of the proposed method and the experimental results demonstrate the effectiveness of the PromptRestorer and achieve better restoration outcomes.


**Weaknesses:**

- The main problem **degradation vanishing** is not well defined and discussed in this paper, leading to the confusing and distracting presentation that what problem the authors are trying to solve. In Section 4.3 the authors claim sharper details in features fail to provide sufficient degraded information. Why this is the case? Is there any evidence that can support sharper features of the conditional branch would deteriorate the restoration performance? Overall the analysis of degradation vanishing is insufficient.

- The core idea of this paper seems like another attempt to utilize prior information from pre-trained large models (*e.g.*, StyleGAN, DDPM, CLIP) to facilitate restoration, which has long been explored [1,2,3,4] in the image restoration community, though the designs and technical details are rather different. Could the authors further elaborate on the main differences between previous works?

- As claimed that PromptRestorer can consistently retain the degradation priors (L62), the reviewer would expect a better adaptation of this model to various degradations. Hence, it would be better if the authors can show how well this algorithm can perform on different degradations (*e.g.*, various sizes of blur kernels, various levels of noise).

[1] Chan, Kelvin CK, et al. "Glean: Generative latent bank for large-factor image super-resolution." Proceedings of the IEEE/CVF conference on computer vision and pattern recognition. 2021.

[2] Lugmayr, Andreas, et al. "Repaint: Inpainting using denoising diffusion probabilistic models." Proceedings of the IEEE/CVF Conference on Computer Vision and Pattern Recognition. 2022.

[3] Kim, Geonung, et al. "Bigcolor: colorization using a generative color prior for natural images." European Conference on Computer Vision. Cham: Springer Nature Switzerland, 2022.

[4] Chen, Chaofeng, et al. "Real-world blind super-resolution via feature matching with implicit high-resolution priors." Proceedings of the 30th ACM International Conference on Multimedia. 2022.


**Questions:**

See weaknesses section.

**Limitations:**

Limitations and broader societal impacts are not discussed in this paper.

---

> ### Author Rebuttal · Authors · 2023-08-09
>
> Q: The main problem ''degradation vanishing'' is not well defined and discussed in this paper, leading to a confusing and distracting presentation.
>
> A: ''Degradation vanishing'' means the degradation features gradually vanish to become sharper features in the learning process of restoration networks. It is a straightforward phenomenon in most restoration networks as the model parameters are optimized toward producing clearer content, as shown in Case 1-2 of Fig. 8.
>
> We discuss that using previously widely-used learnable conditional modulation to guide restoration models is worse than our solution which directly provides raw degradation information into deep restoration models (see Fig. 1, Tab. 5, and Case 2-3 in Fig. 8).
>
> Hence, we introduce and show the degradation vanishing by visualization in Fig. 8, solve it by providing raw degradation features to restoration networks, discuss and compare it with previous related works in Tab. 5, and show that the proposed solution is simpler and more effective.
>
> Q: In Sec. 4.3 the authors claim sharper details in features fail to provide sufficient degraded information. Why this is the case? Is there any evidence that can support sharper features of the conditional branch would deteriorate the restoration performance?
>
> A: Sharper details contain less degraded content as the model parameters are optimized in the learning process of restoration networks, while degraded features can provide the restoration networks with accurate degraded priors so that the restoration networks can perceive degraded content to guide the restoration process for better recovery.
>
> The visualization of Case 2 in Fig. 8 reveals that the learnable conditional branch learns sharper features, which thus cannot effectively provide restoration networks with effective degradation priors. Tab. 5 shows that the learnable conditional manner deteriorates the restoration performance, where our method produces 0.369dB gains.
>
> Moreover, Fig. 1(b) shows that the Case 2 has better performance in early iterations (about 25K iterations) but deteriorates the restoration performance after latter iterations compared with our method since Case 2 exhibits sharper results in later iterations while our method can constantly provide the restoration branch with raw degradation information, which would help better recovery.
>
> Our visualization and quantitative results support that using a learnable conditional branch to produce sharper features is not more effective than our method that using raw degradation to guide restoration networks.
>
> Q: The core idea seems like another attempt to utilize prior information from pre-trained large models to facilitate restoration, which has long been explored [1,2,3,4]. Could the authors further elaborate on the main differences between previous works?
>
> A: GLEAN utilizes pre-trained GANs to generate a latent bank, which produces the ''generative priors'' to embed into the encoder-decoder framework for image SR. Repaint employs a pre-trained unconditional DDPM as the ''generative prior'' to condition the generation process by sampling from the given pixels during the reverse diffusion iterations. BigColor also adopts the ''generative color prior'' to guide the image colorization. FeMaSR utilizes the VQGAN to generate the codebook of clean images to supervise the networks.
>
> Different from all previous methods that utilize the generative priors or codebook of clean images, our method directly utilizes the ''degradation information'' of the degraded image itself to guide restoration.
>
> We note that there is NOT any literature to explore the original degraded features to guide image restoration up to now. We give the validation that raw degradation content extracted from the degraded images itself is more effective than previous commonly-used methods that usually use a learnable conditional branch to guide the image restoration. We make it possible that NOT learnable condition is better than learnable condition methods.
>
> Q: As claimed that PromptRestorer can consistently retain the degradation priors (L62), the reviewer would expect a better adaptation of this model to various degradations. Hence, it would be better if the authors can show how well this algorithm can perform on different degradations (e.g., various sizes of blur kernels, various levels of noise).
>
> A: First, blur kernels or various levels of noise are not available in practical images. Second, estimating blur kernels or various levels of noise may be not an easy task, which will introduce additional estimating errors that may subsequently affect the restoration quality.
>
> Our PromptRestorer does not require any additional estimation modules to estimate the gap between the degraded images and clean ones. In contrast, we directly utilize the degradation features extracted from the degraded image itself by a pre-trained model to provide the restoration networks with more precise degraded information. Hence, our PromptRestorer is simpler, more general, and more suitable for general image restoration tasks.

---

> > ### Comment · Reviewer_NMEt · 2023-08-16
> >
> > Thanks for the efforts in the response. I still have some concerns after reading the rebuttal.
> > - How is the learnable net in Case 2 initialized?
> > - The claim "sharper details in features fail to provide sufficient degraded information" is still weak and unconvincing. Simply visualizing features of *ONE* image and relating it to an overall PSNR gain is weak evidence to make strong conclusion. For example, Case 2 exhibits sharper features in its later iterations (500K) but achieves better performance (~30.5 dB in Fig (b)), which contradicts the claim and explanation in the rebuttal. The authors should dig deep into this phenomenon and do comprehensive analysis before making strong claims. Is it possible to fix the well-trained restoration branch and only fine-tune conditional branch? In this case, we can rule out the optimization of restoration branch and explore the relation between conditional branch and final performance.
> > - The so-called *generative priors* in previous works, more or less, are different levels of features of the pre-trained models. Some prefer to freeze the pre-trained models, while others adopt joint training. Considering this, the core idea of using "original degraded features to guide image restoration" is, in its nature, not that novel in the community. The authors should avoid overclaiming the contribution.
> > - In the last question, the authors are not asked to add an additional estimation module. If "PromptRestorer can consistently retain the degradation priors", it should be able to handle images with various degradations in a *BLIND* setting. This claim, however, is not supported by any evidence.

---

> > > ### Author Response · Authors · 2023-08-18
> > > **Response to Reviewer NMEt (1/2)**
> > >
> > > Q: How is the learnable net in Case 2 initialized?
> > >
> > > A: In Case 2, the learnable conditional branch has similar learning manners to existing conditional modulation methods [1,2,3,4,5,6]. It is directly trained together with the restoration network without initialization, which aims to adaptively learn useful conditional content to guide the restoration networks. We find that the learnable manner in Case 2 is not better than our method (Case 3) which directly uses raw degradation features of the degraded image itself without learning.
> > >
> > > The learnable conditional branch tends to forget the degradation content of the input images with more iterations (Case 2 in Fig. 8), while our NOT learnable method can provide the restoration branch with better degradation priors (Case 3 in Fig. 8) so that it can better restore images. Note that the learnable conditional branch in Case 2 has the same network structure as our Case 3 for fair comparisons.
> > >
> > > [1] Interactive multi-dimension modulation for image restoration. TPAMI, 2022.
> > >
> > > [2] Interactive multi-dimension modulation with dynamic controllable residual learning for image restoration. In ECCV, 2020.
> > >
> > > [3] Conditional sequential modulation for efficient global image retouching. In ECCV, 2020.
> > >
> > > [4] Hdrunet: Single image hdr reconstruction with denoising and dequantization. In CVPR Workshops, 2021.
> > >
> > > [5] A new journey from sdrtv to hdrtv. In ICCV, 2021.
> > >
> > > [6] Toward interactive modulation for photo-realistic image restoration. In CVPR Workshops, 2021.
> > >
> > > Q: The claim "sharper details in features fail to provide sufficient degraded information" is still weak and unconvincing. For example, Case 2 exhibits sharper features in its later iterations (500K) but achieves better performance (30.5 dB in Fig. 1 (b)).
> > >
> > > A: In Fig. 1 (b), our method (Case 3) shows better performance than Case 1-2 at later iterations (500K).
> > >
> > > |   Case   | PSNR (dB) |
> > > |:--------:|:---------:|
> > > |     1    |   30.138  |
> > > |     2    |   30.646  |
> > > | 3 (Ours) |   31.015  |
> > >
> > > And our method uses a not learnable network to extract the raw degradation features while Case 2 uses a learnable conditional network which consumes more learnable parameters.
> > >
> > > Notably, both Cases 1-2 exhibit sharper results in later iterations compared to earlier ones, which fail to provide the restoration branch with sufficient degraded information. In contrast, as the restoration branch needs to adapt perceived features from the PromptDPM which is to perceive the raw degradation features from the degradation image itself, our model (Case 3) initially exhibits inferior performance (around 20K iterations) as shown in Fig. 1(b). However, with better adaptation to the degradation information after more iterations, the prompting branch can better prompt the restoration branch with more reliable perceived content learned from the raw degradation, enabling our restoration branch to overcome degradation vanishing to achieve better restoration quality.
> > >
> > > Q: Is it possible to fix the well-trained restoration branch and only fine-tune the conditional branch?
> > >
> > > A: Thanks for your good suggestions. We will discuss the case in the revised paper as the time does not allow us to finish the training of the model. Moreover, we would like to further express that fixing the well-trained restoration branch and only fine-tuning the conditional branch seems to learn suitable conditional content for a specific restoration network. Furthermore, this case is more complicated since we need to individually train each restoration network for the specific restoration task and then individually fine-tune each conditional branch. In contrast, our method is simpler that only needs to pre-train the conditional network once and then can be applied to all restoration networks to serve as the accurate degradation feature extractor. Our approach greatly simplifies the method mentioned by the reviewer.
> > >
> > > Moreover, our method that uses raw degradation features to guide the restoration network may have better insights since we explore the accurate degradation prior in deep feature space for degraded image restoration. For the case that the reviewer suggests, it seems to discard the degradation prior but is a good suggestion that will be discussed in the revised paper.

---

> > > > ### Author Response · Authors · 2023-08-18
> > > > **Response to Reviewer NMEt (2/2)**
> > > >
> > > > Q: The so-called generative priors in previous works are different levels of features of the pre-trained models.
> > > >
> > > > A: Yes. The generative priors also provide the features from pre-trained models to restoration networks. However, we note that these methods with generative priors first learn the generative prior knowledge from large-scale additional data and then exploit it to guide restoration. In such cases, these methods learn some **additional knowledge** from **other data prior distribution** to guide deep restoration models. In contrast, our method **does not learn other knowledge**, whereas the pre-trained model only plays the role of the accurate feature extractor to extract the degradation features of the **degraded image itself**.
> > > >
> > > > Furthermore, we observe and demonstrate that previous widely-used methods which use a learnable conditional branch to guide restoration are not better than the model with a NOT learnable branch, which is not explored yet.
> > > >
> > > > We thank you for your good suggestions. We will discuss the relationship between the generative priors and our degradation priors in the revised paper.
> > > >
> > > > Q: The authors are not asked to add an additional estimation module. If "PromptRestorer can consistently retain the degradation priors", it should be able to handle images with various degradations in a BLIND setting.
> > > >
> > > > A:  Adding an additional estimation module to estimate the degradations may be not an easy task. In addition, if degradations are not accurately estimated by this module, errors may subsequently affect the restoration quality. Moreover, in such cases, different restoration tasks also require estimating different degradations, which may introduce a more complex model and be not suitable for unified image restoration networks. Our method does not require additional estimation modules to estimate the gap between the degraded images and clean ones. We directly utilize the degradation features extracted from the degraded image itself by a pre-trained model to provide the restoration networks with more precise degraded information. Hence, our method, to some extent, is simpler for general image restoration tasks.
> > > >
> > > > For the blind setting, on the image deblurring task, our method is trained on the GoPro and then directly applied to HIDE, RealBlur-R, and RealBlur-J. Compared with the recent state-of-the-art method NAFNet, our method consistently achieves better generalization to unseen scenes on HIDE, RealBlur-R, and RealBlur-J while consuming about 1/3 parameters. The results show that our method is able to handle the images with blind settings to some extent.
> > > >
> > > > |  Datasets  |       NAF    |     Ours   |
> > > > |:----------:|:-----------:|:-----------:|
> > > > |    GoPro   |  33.71/0.967 | 33.06/0.962 |
> > > > |    HIDE    |  31.31/0.943 | 31.36/0.944 |
> > > > | RealBlur-R |  35.97/0.951 | 36.06/0.954 |
> > > > | RealBlur-J |  28.31/0.856 | 28.82/0.873 |
> > > > |  Para (M)  |      67.9    |     24.4    |
> > > >
> > > >
> > > > For the image deraining, our method is trained on Rain13K and then directly applied to Test100, Rain100H, Rain100L, Rain1200, and Rain2800 testing sets. Our method achieves the best PSNR and SSIM on average. The results also illustrate that our method is able to handle blind image restoration to some extent. The training samples of Rain13K and testing sets are summarised in the Supplementary Materials.

---

> > > > > ### Comment · Reviewer_NMEt · 2023-08-21
> > > > >
> > > > > Thanks for the clarification. I appreciate the integral design of PromptDMP and find it effective to utilize priors from pre-trained models.
> > > > > - Why the proposed method does **NOT** learn other knowledge, since it uses *pre-trained* VQGAN, which intrinsically learns the data distribution from large-scale datasets? This statement only holds true when the prompting branch is **randomly weighted**. In this aspect, the experiments between Case 2 and Case 3 are unfair. Case 2 is initialized with **random** weights, while Case 3 uses **pre-trained** weights. A correct experiment design is to initialize both cases with pre-trained weights and get the network in Case 2 *learnable* and Case 3 *fixed*.
> > > > > - The experiments on image deblurring tasks show some indirect results on the generalizability of the proposed model. However, it would be much clearer and more controllable to evaluate models'  adaptation and generalizability with controllable experiments. For example, the authors can train the PromptRestorer and directly apply it to datasets generated with different levels of noise (*e.g.*, Table 9 in [1]). Note that this does not involve any explicit degradation estimation.
> > > > >
> > > > > [1] Potlapalli, Vaishnav, et al. "PromptIR: Prompting for All-in-One Blind Image Restoration." arXiv preprint arXiv:2306.13090 (2023).

---

> > > > > > ### Author Response · Authors · 2023-08-21
> > > > > > **Response to Reviewer NMEt (1/2)**
> > > > > >
> > > > > > Q: Why the proposed method does NOT learn other knowledge, since it uses pre-trained VQGAN, which intrinsically learns the data distribution from large-scale datasets? This statement only holds true when the prompting branch is randomly weighted. Case 2 is initialized with random weights, while Case 3 uses pre-trained weights. A correct experiment design is to initialize both cases with pre-trained weights and get the network in Case 2 learnable and Case 3 fixed.
> > > > > >
> > > > > > A: The pre-trained VQGAN is only used to extract the raw features of the degraded image itself as the VQGAN has shown that it can reconstruct the input images well in some VQGAN-based image restoration methods, e.g., [1,2,3].
> > > > > >
> > > > > > Note that we only use the encoder of VQGAN instead of the features after vector quantization as vector quantization tends to damage image structure [1,2,3]. Furthermore, as we do not use the learned codebook of the VQGAN, the encoder of the pre-trained VQGAN does not involve the knowledge transfer from other data.
> > > > > >
> > > > > > Moreover, the GT features and degraded features of Case 3 in Fig. 8 show that the encoder of VQGAN can better extract the features of the input image itself.
> > > > > >
> > > > > > We thank the very good suggestions from the reviewer NMEt and we will initialize Case 2 with pre-trained weights in the revised paper. As the deadline is approaching, it does not allow us to finish the training but we sincerely guarantee that we will adopt the suggestions by the reviewer in the revised paper.
> > > > > >
> > > > > > Case 2 is learnable, which aims to adaptively learn useful conditional content to guide the restoration networks. We follow existing conditional modulation-based restoration networks [4,5,6,7,8,9] that set Case 2 with random initialization weights as the conditional networks play the role of adaptively learning the most useful content for the restoration networks in different restoration tasks.
> > > > > >
> > > > > > [1] Gu et al. VQFR: Blind Face Restoration with Vector-Quantized Dictionary and Parallel Decoder. ECCV2022.
> > > > > >
> > > > > > [2] S. Zhou, K.C.K. Chan, C. Li, and C.C. Loy. Towards robust blind face restoration with codebook look-up transformer. InNeurIPS, 2022.
> > > > > >
> > > > > > [3] C. Chen, X. Shi, Y. Qin, X. Li, X. Han, T. Yang, and S. Guo. Real-world blind super-resolution via feature matching with implicit high-resolution priors. In ACM MM, pages 1329–1338,2022.
> > > > > >
> > > > > > [4] Interactive multi-dimension modulation for image restoration. TPAMI, 2022.
> > > > > >
> > > > > > [5] Interactive multi-dimension modulation with dynamic controllable residual learning for image restoration. In ECCV, 2020.
> > > > > >
> > > > > > [6] Conditional sequential modulation for efficient global image retouching. In ECCV, 2020.
> > > > > >
> > > > > > [7] Hdrunet: Single image hdr reconstruction with denoising and dequantization. In CVPR Workshops, 2021.
> > > > > >
> > > > > > [8] A new journey from sdrtv to hdrtv. In ICCV, 2021.
> > > > > >
> > > > > > [9] Toward interactive modulation for photo-realistic image restoration. In CVPR Workshops, 2021.

---

> > > > > > > ### Author Response · Authors · 2023-08-21
> > > > > > > **Response to Reviewer NMEt (2/2)**
> > > > > > >
> > > > > > > Q: The authors can train the PromptRestorer and directly apply it to datasets generated with different levels of noise. Note that this does not involve any explicit degradation estimation.
> > > > > > >
> > > > > > > A: We will add the results on blind image denoising including color/grayscale Gaussian denoising which is trained on single-level noises and then test on a variety level of noisy images in the revised paper since the time does not allow us to finish the complete training of the model (It takes over one week to finish the training of a single restoration model). We sincerely guarantee that we will add the blind image denoising results in the revised paper.
> > > > > > >
> > > > > > > Moreover, we would like to clarify that the restoration performance mainly depends on the restoration networks. Although our prompting method can provide the restoration networks with reliable degradation features, it would not get promising results if the restoration networks were lightweight. As the restoration network needs to learn and process the prompted degradation features, it is natural that the restoration performance depends on the restoration networks.
> > > > > > >
> > > > > > > Furthermore, one may think that the learnable networks would be better than our Case 3 which uses the Not learnable conditional model since the learnable conditional networks can adaptively learn the most useful content for the restoration networks. However, we make the first attempt and find that a Not learnable conditional network can maximize the feature-level priors to make the restoration networks perceive the reliable feature-level degradation of the degraded image itself to facilitate the restoration networks to more effectively recover.
> > > > > > >
> > > > > > > The NOT learnable conditional network has some advantages:
> > > > > > >
> > > > > > > On the one hand, compared with the learnable conditional networks that provide gradually clear features for the restoration network which may be more useless for later learning with more iterations, our NOT learnable conditional network consistently provides the restoration network with more reliable degradation features so that the restoration network can perceive the degradation features to help the restoration network more effectively learn.
> > > > > > >
> > > > > > > On the other hand, our NOT learnable conditional network is more explainable as it can better provide the degraded image restoration network with more precise degraded information from the degraded image itself. Hence, our network may be more general for degraded image restoration tasks.

---

> > > > > > > ### Comment · Reviewer_NMEt · 2023-08-21
> > > > > > >
> > > > > > > Thanks for the swift response.
> > > > > > >
> > > > > > > I cannot agree with the argument that *"the encoder of the pre-trained VQGAN does not involve the knowledge transfer from other data"*. When employing pre-trained models, whether in their entirety or just specific parts, they inherently encompass the prior knowledge embedded within them through their training data. Again, it is important to note that this argument holds true only if the prompting branch is *randomly weighted*. There do exist studies that reveal that the network structure, rather than the pre-trained weights, can adequately capture the dependencies between variable statistics [2,3]. However, this assertion cannot be expanded to the current scenario without concrete observations and evidence.
> > > > > > >
> > > > > > > In addition, it is essential to establish the same initialization in both Case 2 and Case 3, regardless of whether *random* or *pre-trained* weights are used. This precondition is necessary to derive conclusions that the **fixed** prompting branch is better/comparable/worse than the **learnable** one. This consideration is distinct and has nothing to do with common practice in existing conditional modulation-based restoration networks. Without such comparative experiments, it is not prudent to draw definitive conclusions.
> > > > > > >
> > > > > > >
> > > > > > > [2] Ulyanov, Dmitry, et al. "Deep image prior." Proceedings of the IEEE conference on computer vision and pattern recognition. 2018.
> > > > > > >
> > > > > > > [3] Liu, Yifan, et al. "Generic perceptual loss for modeling structured output dependencies." Proceedings of the IEEE/CVF Conference on Computer Vision and Pattern Recognition. 2021.

---

> > > > > > > > ### Author Response · Authors · 2023-08-21
> > > > > > > > **Response to Reviewer NMEt**
> > > > > > > >
> > > > > > > > We thank the reviewer for the swift response and such constructive suggestions that significantly improve the quality of this paper.
> > > > > > > >
> > > > > > > > We sincerely accept the suggestions from the reviewer that we should initialize Case 2 with pre-training weights and then derive conclusions that the fixed prompting branch is better/comparable/worse than the learnable one. We will add additional experiments on Case 2 with pre-trained initialization in the revised paper.
> > > > > > > >
> > > > > > > > Since the DDL is approaching, could we show the results in the revised paper after we finish the model training? Many thanks for your valuable reviews again!

---

### Official Review · Reviewer_TXGv · 2023-07-16

**Soundness:** 2 fair
**Presentation:** 3 good
**Contribution:** 2 fair
**Rating:** 3
**Confidence:** 4

**Summary:**

This paper proposes a Prompt image Restorer, which contains a restoration branch and a prompting branch. A pre-trained model is utilized to extract features in the prompting branch and a prompting modulator is proposed to better perceive features from global and local perspectives in the restoration branch. Extensive experimental results of multiple restoration tasks show the effectiveness of their method.

**Strengths:**

1. The results in some experiments have shown improvement compared to previous methods.
2. The paper is well-written and easy to understand.

**Weaknesses:**

1. The prompting way proposed in this paper is similar to the approach taken in some other papers, such as "Take a Prior from Other Tasks for Severe Blur Removal", where features are added as conditions. PromptRestorer doesn't have a significant difference, which makes me feel like it is more of a repackaging of prompting learning.
2. As a model for image generation, why does the pre-trained model provide raw degradation information instead of image content information? What advantages does this approach have compared to some other methods of degradation representation? For example, "Learning Degradation Representations for Image Deblurring", and "Learning Disentangled Feature Representation for Hybrid-distorted Image Restoration".
3. The comparisons with the latest methods in Table 2, such as Restormer and NAFNet, are missing.
4. The article lacks comparisons with previous methods in terms of computational complexity and the number of parameters.
5. The method in this paper employs a pre-trained  VQGAN, which introduces additional data. Is this experimental comparison fair? Did the authors account for the impact of this aspect?
6. From Figure 8, it can be observed that your method visually resembles the features of the ground truth (GT). However, GT does not contain any degradation information, which contradicts the claims of raw degradation information made in your paper. This makes the motivation and claims of this paper unconvincing.

**Questions:**

Please refer to Weaknesses.

---

> ### Author Rebuttal · Authors · 2023-08-09
>
> Q: The prompting way proposed in this paper is similar to the approach taken in some other papers, such as "Take a Prior from Other Tasks for Severe Blur Removal", where features are added as conditions. PromptRestorer doesn't have a significant difference.
>
> A: The abovementioned method uses a semantic/classification prior to guide the restoration branch. We note that the condition networks in this method are learnable and optimized by the features of GT. This manner is similar to the model of Case 2 in our paper. However, our method is different from the abovementioned method, where we use a pre-trained model to extract the raw degraded features and the pre-trained model keeps freezing when optimizing the restoration branch. Such a design provides the restoration branch with original degradation features instead of optimized features. Our method is more direct, simpler, and more effective than the ones that usually use a learnable branch to guide the restoration network.
>
> Moreover, different from existing methods, we also show that using the pre-trained model to extract raw degraded features to guide the restoration network overcomes the problem of ''degradation vanishing'' (as shown in Fig. 8), which provides the restoration networks with precise degraded information, enabling the restoration network to perceive the degradation priors for better restoration.
>
> To the best of our knowledge, we are the first to use the raw degraded features to guide image restoration and show that raw degraded features are more effective than learnable conditions.
>
> Q: As a model for image generation, why does the pre-trained model provide raw degradation information instead of image content information?
>
> A: Since the pre-trained VQGAN can reconstruct the input images as illustrated in VQFR [1], the pre-trained VQGAN model represents the features of the input images in feature space. While the input is the degraded image, the pre-trained VQGAN model generates degradation features. Hence, naming the raw degradation information is more precise than image content information.
>
> [1] Gu et al. VQFR: Blind Face Restoration with Vector-Quantized Dictionary and Parallel Decoder. ECCV2022.
>
> Q: What advantages does this approach have compared to some other methods of degradation representation? e.g., "Learning Degradation Representations for Image Deblurring" (Ref-A), and "Learning Disentangled Feature Representation for Hybrid-distorted Image Restoration" (Ref-B).
>
> A: We note that the method in Ref-A first jointly trains the conditional encoder that extracts the degraded information with the deblurring networks within 200K iterations in the first training phase, then freezes the conditional encoder and trains the deblurring networks in the second training phase.
>
> The first training phase will make the encoder cannot effectively preserve the degraded information as the encoder is optimized toward sharper features. In contrast, our pre-trained model which extracts the degraded information keeps freezing when training the restoration networks and thus can more precisely generate the degraded information. Hence, our method is a pure method that provides degraded information for restoration networks.
>
> The method in Ref-B uses the learnable disentangled features in the restoration networks instead of using degraded information to guide the deep networks.
>
> Hence, our method is different from the above methods.
>
> Q: Comparisons with Restormer and NAF.
>
> A: We re-implement our method and compare it with the Restormer and NAF and the results are summarised in the following Table. Compared with Restormer, our method consumes fewer parameters while improving the PSNR by 0.14dB and producing better generalization on the HIDE dataset. On RealBlur benchmarks, our PromptRestorer is competitive. Compared with the NAF, our method only consumes about 1/3 parameters of NAF but achieves consistent improvement on HIDE, RealBlur-R, and RealBlur-J. These results demonstrate our PromptRestorer is competitive.
>
> |  Datasets  |  Restormer  |     NAF     |     Ours    |
> |:----------:|:-----------:|:-----------:|:-----------:|
> |    GoPro   | 32.92/0.961 | 33.71/0.967 | 33.06/0.962 |
> |    HIDE    | 31.22/0.942 | 31.31/0.943 | 31.36/0.944 |
> | RealBlur-R | 36.19/0.957 | 35.97/0.951 | 36.06/0.954 |
> | RealBlur-J | 28.96/0.879 | 28.31/0.856 | 28.82/0.873 |
> |  Para (M)  |     26.1    |     67.9    |     24.4    |
> |  FLOPs (G) |    140.99   |    63.33    |    186.3    |
>
> We set the number of channels as 32, 64, and 128, the number of CGT is [2, 3, 5] from level-1 to level-3, and the expanding channel capacity factor is 3. The iterations are 600K and the batch size is 15.
>
> Q: The method in this paper employs a pre-trained VQGAN, which introduces additional data. Is this comparison fair? Did the authors account for the impact?
>
> A: The comparison is fair. We compare our method with the model with learnable conditional modulation, where the learnable conditional modulation has the same network architecture as the pre-trained VQGAN encoder used in our method for fair comparisons (footnote on Page 8). Tab. 5 shows that our method performs better.
>
> Q: From Fig. 8, it can be observed that your method visually resembles the features of the GT. However, GT does not contain any degradation information, which contradicts the claims of raw degradation information made in your paper.
>
> A: As GT features and the raw degraded features are extracted from the same pre-trained VQGAN, their colors are similar but the GT features are sharper and the raw degraded features are blurry. As other features are shown from their individual learning networks, both of them have different patterns (colors). By viewing on a high-resolution display, the degraded features are more blurry while other features become sharper. The GT feature put in this paper only plays a reference. Hence, our conclusion is consistent with our motivation.

---

> > ### Comment · Reviewer_TXGv · 2023-08-16
> >
> > Your approach involves a pre-trained prompting branch with freezing, while the learnable ways include a randomly initialized prompting branch with fine-tuning, and a pre-trained prompting branch with fine-tuning. From the context provided in the text, it seems that Case 2 refers to the randomly initialized prompting branch with fine-tuning. Therefore, in your Case 2 experiments, how was the prompting branch initialized?

---

> > > ### Author Response · Authors · 2023-08-18
> > > **Response to Reviewer TXGv**
> > >
> > > Q: Your approach involves a pre-trained prompting branch with freezing, while the learnable ways include a randomly initialized prompting branch with fine-tuning, and a pre-trained prompting branch with fine-tuning. From the context provided in the text, it seems that Case 2 refers to the randomly initialized prompting branch with fine-tuning. Therefore, in your Case 2 experiments, how was the prompting branch initialized?
> > >
> > > A: In Case 2, the learnable conditional branch has similar learning manners to existing conditional modulation methods [1,2,3,4,5,6]. It is directly trained together with the restoration network without initialization, which aims to adaptively learn useful conditional content to guide the restoration networks. We find that the learnable manner in Case 2 is not better than our method (Case 3) which directly uses raw degradation features of the degraded image itself without learning.
> > >
> > > The learnable conditional branch tends to forget the degradation content of the input images with more iterations (Case 2 in Fig. 8), while our NOT learnable method can provide the restoration branch with better degradation priors (Case 3 in Fig. 8) so that it can better restore images. Note that the learnable conditional branch in Case 2 has the same network structure as our Case 3 for fair comparisons.
> > >
> > > [1] Interactive multi-dimension modulation for image restoration. TPAMI, 2022.
> > >
> > > [2] Interactive multi-dimension modulation with dynamic controllable residual learning for image restoration. In ECCV, 2020.
> > >
> > > [3] Conditional sequential modulation for efficient global image retouching. In ECCV, 2020.
> > >
> > > [4] Hdrunet: Single image hdr reconstruction with denoising and dequantization. In CVPR Workshops, 2021.
> > >
> > > [5] A new journey from sdrtv to hdrtv. In ICCV, 2021.
> > >
> > > [6] Toward interactive modulation for photo-realistic image restoration. In CVPR Workshops, 2021.

---

> > > > ### Comment · Reviewer_TXGv · 2023-08-18
> > > >
> > > > Your comparative experiment is incomplete and exhibits significant flaws. You have not accounted for the influence of additional data and initialization, yet you claim that the "not learnable" approach is superior to the "learnable conditional branch" method. This not only lacks rigor but also contradicts the effectiveness of widely used fine-tuning. Without further comprehensive experimental evidence, I tend to believe that this paper is not suitable for publication, as it could be misleading to the advancement of the image restoration field.

---

> > > > > ### Author Response · Authors · 2023-08-21
> > > > > **Response to Reviewer TXGv**
> > > > >
> > > > > Q: Your comparative experiment is incomplete and exhibits significant flaws. You have not accounted for the influence of additional data and initialization, yet you claim that the "not learnable" approach is superior to the "learnable conditional branch" method. This not only lacks rigor but also contradicts the effectiveness of widely used fine-tuning. Without further comprehensive experimental evidence, I tend to believe that this paper is not suitable for publication, as it could be misleading to the advancement of the image restoration field.
> > > > >
> > > > > A: We respectfully disagree with the statement of the reviewer. Most conditional modulation-based methods do not use specific initialization, e.g., [1,2,3,4,5,6,7,8].
> > > > >
> > > > > Moreover, we will initialize Case 2 with pre-trained weights in the revised paper. As the deadline is approaching, it does not allow us to finish the training but we sincerely guarantee that we will add the results in the revised paper.
> > > > >
> > > > > [1] Interactive multi-dimension modulation for image restoration. TPAMI, 2022.
> > > > >
> > > > > [2] Interactive multi-dimension modulation with dynamic controllable residual learning for image restoration. In ECCV, 2020.
> > > > >
> > > > > [3] Conditional sequential modulation for efficient global image retouching. In ECCV, 2020.
> > > > >
> > > > > [4] Hdrunet: Single image hdr reconstruction with denoising and dequantization. In CVPR Workshops, 2021.
> > > > >
> > > > > [5] A new journey from sdrtv to hdrtv. In ICCV, 2021.
> > > > >
> > > > > [6] Toward interactive modulation for photo-realistic image restoration. In CVPR Workshops, 2021.
> > > > >
> > > > > [7] Blind Super-Resolution With Iterative Kernel Correction, CVPR19.
> > > > >
> > > > > [8] Recovering Realistic Texture in Image Super-Resolution by Deep Spatial Feature Transform, CVPR18.

---

> > > > > > ### Comment · Reviewer_TXGv · 2023-08-22
> > > > > >
> > > > > > After reviewing the author's response, I believe that the author may still not realize that the problem lies not only in the lack of crucial experiments but also in the flawed claims and motivations of the article based on incomplete experimental results. Due to this reason and considering the content of the author's reply, I am still giving a final rating of "reject" and increasing my confidence level to "very confident."

---

### Comment · Area_Chair_tWDv · 2023-08-16
**Discussion**

Dear Reviewers,

I appreciate your efforts thus far. Please read the author's rebuttals and other reviews attentively and respond to at least acknowledge that you've seen them. If your evaluation of the paper changes, please update your score and briefly explain the difference.

The paper received diverging initial reviews. Please consider discussing whether we can reach a consensus with the authors or other reviewers.

Thank you,
AC

---

### Author Response · Authors · 2023-08-18
**Further clarify the contributions of this paper as well as the difference from previous works**

Dear all reviewers:

We would like to clarify our contributions as well as the difference from previous works to facilitate you to make final decisions.

1, We find and demonstrate that using raw degradation features extracted from a pre-trained model is more effective than previous widely-used learnable conditional modulation to guide image restoration. Raw degradation features would make restoration networks better perceive more reliable degradation priors to facilitate better image restoration. The motivation of this paper is simpler, clearer, and more general for image restoration, which is that raw degradation features can provide the restoration network with effective degradation priors from the degraded image itself instead of other additional priors learned from additional data distribution (e.g., generative prior, semantic prior, etc). This finding is not explored yet and we hope that our work can offer a new perspective for future conditional modulation-based image restoration.

2, We propose a new module, the prompting degradation perception modulator (PromptDMP), that is used to perceive degradation from global and local perspectives (G2P and L2P), which is able to provide the restoration network with more reliable perceived content learned from the degradation priors, enabling it to better guide the restoration process.

3, We propose the gated degradation perception propagation (GDP) that exploits a gating mechanism to control the propagation of the perceived features, enabling the model to adaptively learn more useful features for better image restoration.

We sincerely hope that our rebuttal can address your concerns and look forward to your response.

Best wishes,

The authors

---

### Comment · Area_Chair_tWDv · 2023-08-19
**Look forward to further feedback**

Dear Reviewers,

The open discussion phase of the paper is nearing its end, and the authors have given more detailed explanations in the rebuttal phase, mainly focusing on the contributions, rationale, and limitations of the proposed algorithms. In order to ensure the smooth running of the conference, we would like to receive your responses to the authors' rebuttals as soon as possible. Therefore, we kindly ask you to submit your feedback as early as possible, if possible. Once again, we thank you for your time and look forward to your valuable comments.

Thank you,
AC

---

### Decision · Program_Chairs · 2023-09-21

**Decision:**

Accept (poster)

**Comment:**

This paper points out the problem of degradation vanishing in image recovery and puts forward the idea that accurate original degradation features can guide the deep recovery model. Based on this, this paper designs a PromptRestorer network containing recovery branches and prompting branches, which cues the recovery branches for better-guiding image restoration through reliable perceptual content prompting with the designed PromptDMP. Almost all reviewers recognized the technical novelty as well as the contribution of the method. The authors' response successfully addressed most of the reviewers' concerns and provided additional (supportive) experimental results. Thus, the AC recommends accepting this paper. However, some reviewers still expressed partial concerns about the experiments in this paper. I suggest that the authors should further add to the experiments and organize the open source in the final version.